# Is Reinforcement Learning (Not) for Natural Language Processing: Benchmarks, Baselines, and Building Blocks for Natural Language Policy Optimization

**Rajkumar Ramamurthy**[*♠]  **Prithviraj Ammanabrolu**[*♡]  **Kianté Brantley**[♣]  **Jack Hessel**[♡]
**Rafet Sifa**[♠]  **Christian Bauckhage**[♠]  **Hannaneh Hajishirzi**[◇♡]  **Yejin Choi**[◇♡]
[♠]Fraunhofer IAIS  [♡]Allen Institute for Artificial Intelligence  [♣]Cornell University
[◇]Paul G. Allen School of Computer Science, University of Washington
`rajkumar.ramamurthy@iais.fraunhofer.de`
`{raja,jackh}@allenai.org; kdb82@cornell.edu`

## Abstract

We tackle the problem of aligning pre-trained large language models (LMs) with human preferences. If we view text generation as a sequential decision-making problem, reinforcement learning (RL) appears to be a natural conceptual framework. However, using RL for LM-based generation faces empirical challenges, including training instability due to the combinatorial action space, as well as a lack of open-source libraries and benchmarks customized for LM alignment. Thus, a question rises in the research community: *is RL a practical paradigm for NLP?*

To help answer this, we first introduce an open-source modular library, **RL4LMs**[1],[2] for optimizing language generators with RL. The library consists of on-policy RL algorithms that can be used to train any encoder or encoder-decoder LM in the HuggingFace library (Wolf et al., 2020) with an arbitrary reward function. Next, we present the **GRUE (General Reinforced-language Understanding Evaluation)** benchmark, a set of 6 language generation tasks which are supervised not by target strings, but by reward functions which capture automated measures of human preference. GRUE is the first leaderboard-style evaluation of RL algorithms for NLP tasks. Finally, we introduce an easy-to-use, performant RL algorithm, NLPO **(Natural Language Policy Optimization)** that learns to effectively reduce the combinatorial action space in language generation. We show 1) that RL techniques are generally better than supervised methods at aligning LMs to human preferences; and 2) that NLPO exhibits greater stability and performance than previous policy gradient methods (e.g., PPO (Schulman et al., 2017)), based on both automatic and human evaluations.

## 1 Introduction

The ultimate aim of language technology is to interact with humans. However, most language models are trained without direct signals of human preference, with supervised target strings serving as (a sometimes crude) proxy. One option to incorporate user feedback is via human-in-the-loop, i.e., a user would be expected to provide feedback for each sample online as the model trains, but this degree of dense supervision is often prohibitive and inefficient. Automated metrics offer a promising compromise: models of human preference like pairwise learned preference models (Ouyang et al., 2022), BERTScore (Zhang et al., 2019), BLEURT (Sellam et al., 2020) have significantly improved correlation with human judgment compared to earlier metrics (BLEU, METEOR, etc.), and are cheap to evaluate. But — these functions are usually not per-token differentiable: like humans, metrics

---

[*]Denotes Equal Contribution

[1]Code: `https://github.com/allenai/RL4LMs`

[2]Project Website: `https://rl4lms.apps.allenai.org/`

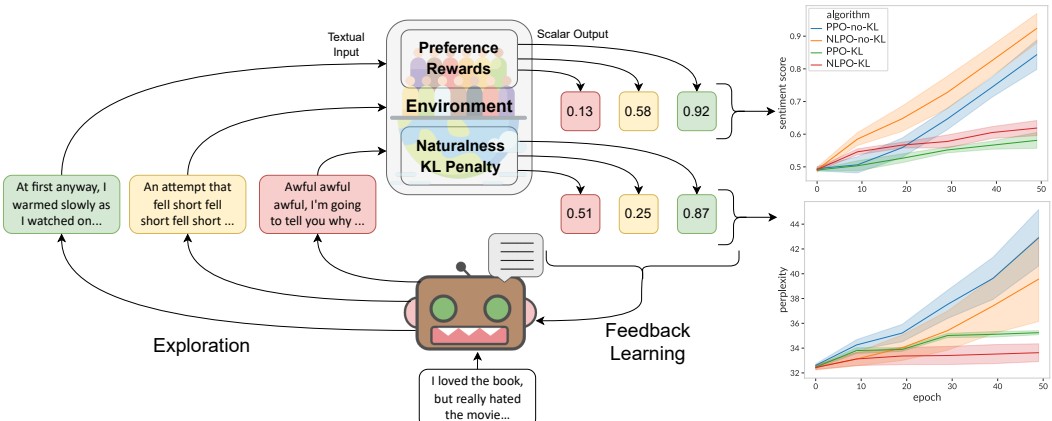

Figure 1: **Natural Language Policy Optimization (NLPO)** in the case of sentiment-guided continuation. Here, the LM (i.e., the policy) needs to produce a positive sentiment continuation given a review prompt (we cover other models of human preference in Sec. 3.2). Two objectives are balanced: 1) an automated proxy of human preference that serves as a reward (here: a sentiment classifier); and 2) "naturalness" as measured by a KL divergence from an LM not trained with explicit human feedback. The plots show validation learning curves comparing our NLPO to the popular policy gradient method PPO. (Top plot:) RL methods can easily achieve high reward if the KL penalty is removed, (Bottom:) but at the cost of higher perplexity. NLPO+KL, our proposed approach, succeeds in balancing reward and naturalness more effectively than prior work.

can only offer quality estimates for full generations. Reinforcement Learning (RL) offers a natural path forward for optimizing non-differentiable, scalar objectives for LM-based generation when it is cast as a sequential decision-making problem. However, Goodhart's Law[3] looms: particularly in the case of imperfect metrics that use neural networks, it is easy to find nonsense samples that achieve high-quality estimates. Recent works have shown promising results in aligning LMs to human preferences via RL by constraining preference-based rewards to incorporate notions of fluency (Wu et al., 2021a; Ouyang et al., 2022) but progress in this line of work is heavily hindered by a lack of open-source benchmarks and algorithmic implementations—resulting in perception that RL is a challenging paradigm for NLP (Choshen et al., 2020; Kreutzer et al., 2021).

To facilitate research in building RL algorithms to better align LMs, we release a library, a benchmark, and an algorithm. First, we release the **RL4LMs library**, which enables generative HuggingFace models (e.g., GPT-2 or T5) to be trained using a variety of existing RL methods like PPO/A2C/etc. Next, we apply models trained using RL4LMs to the new **GRUE (General Reinforced-language Understanding Evaluation)** benchmark: GRUE is a collection of 7 contemporary NLP tasks (see Table1 for details); in contrast to other benchmarks, instead of supervised training, we pair each task with reward function(s). GRUE challenges models to optimize these reward functions while remaining fluent language generators. We train language models via RL—both with and without task specific supervised pre-training—to optimize rewards. Finally, beyond existing RL methods, we introduce a novel on-policy RL algorithm called **NLPO (Natural Language Policy Optimization)**, that dynamically learns task-specific constraints over the distribution of language at a token level.

Experiments on GRUE and human evaluations show that NLPO better balances learning preference rewards while maintaining language fluency compared to alternatives, including PPO (Figure 1). We find that using RL to learn from scalar reward feedback can be more: (1) data efficient than using additional expert demonstrations via supervised learning (though a combination of both is best)—a learned reward function enables greater performance when used as a signal for an RL method than a supervised method trained with 5 times more data, and (2) parameter efficient—enabling a 220 million parameter model trained with a combination of supervision and NLPO to outperform a 3 billion supervised model. We hope that the benchmarks, baselines, and building blocks we release serve to drive forward research in aligning LMs to human preferences.

---

[3]Strathern (1997) paraphrases: *When a measure becomes a target, it ceases to be a good measure.*

## 2 RELATED WORK

**Imitation learning for NLP.** Algorithms such as Schedule Sampling (SS) (Bengio et al., 2015), Parallel SS (Duckworth et al., 2019), SS for Transformers (Mihaylova & Martins, 2019), Diffential SS (Goyal et al., 2017), LOLS (Lampouras & Vlachos, 2016; Chang et al., 2015), TextGAIL (Wu et al., 2021b), and SEARNN (Leblond et al., 2017), have been inspired by DAGGER (Ross et al., 2011) and SEARN (Daumé et al., 2009). However, these algorithms are known to suffer from exposure bias in generation (Chiang & Chen, 2021; Arora et al., 2022) and the cliff MDP problem (Huszár, 2015; Agarwal et al., 2019; Swamy et al., 2021).

**RL for Large Action Spaces.** MIXER (Ranzato et al., 2016) combined ideas from schedule sampling and REINFORCE (Williams, 1992). Bahdanau et al. (2016) proposed an actor-critic algorithm to address the variance/large action space problems when using REINFORCE for language generation; follow-up works such as KG-A2C (Ammanabrolu & Hausknecht, 2020), TrufLL (Martin et al., 2022), AE-DQN (Zahavy et al., 2018), and GALAD (Ammanabrolu et al., 2022) addressed similar issues by attempting to eliminate and reduce the action space during exploration.

**RL for NLP.** RL, often in the form of bandit learning, has been used to improve models in machine translation (Wu et al., 2016; Nguyen et al., 2017; Kiegeland & Kreutzer, 2021), summarization (Stiennon et al., 2020; Paulus et al., 2017), dialogue (Li et al., 2016; Zhou et al., 2017; Jaques et al., 2020), image captioning (Rennie et al., 2017), question generation (Pang & He, 2021), text-games (Narasimhan et al., 2015; Hausknecht et al., 2020), and more (Ranzato et al., 2016; Snell et al., 2022). Lu et al. (2022) adapt reward-conditioned transformers (Chen et al., 2021) for several language generation tasks. RL has been the focus of efforts to align LMs with human preferences (Stiennon et al., 2020; Wu et al., 2021a; Nakano et al., 2021; Ziegler et al., 2019), e.g., Ouyang et al. (2022) fine-tuned large language model with PPO Schulman et al. (2017) to align with models of human preference, but their non-public dataset doesn't enable comparison. Though RL has been successful in some of the use cases described above, it has simultaneously been critiqued for being significantly less stable than supervised LM training (Choshen et al., 2020). As a result, there is relatively little consensus if RL is a worthwhile consideration for training LMs compared to, say, collecting additional supervised data.

## 3 RL4LMS: A LIBRARY FOR TRAINING LMS WITH RL

We introduce RL4LMs, an open-source library with building blocks for fine-tuning and evaluating RL algorithms on LM-based generation. The library is built on HuggingFace (Wolf et al., 2020) and stable-baselines-3 (Raffin et al., 2021), combining important components from their interfaces. RL4LMs can be used to train any decoder only or encoder-decoder transformer models from HuggingFace with any on-policy RL algorithm from stable-baselines-3. Furthermore, we provide reliable implementations of popular on-policy RL algorithms that are tailored for LM fine-tuning such as PPO (Schulman et al., 2017), TRPO (Schulman et al., 2015a), A2C (Mnih et al., 2016), and our own NLPO (§4). The library is modular, which enables users to plug-in customized environments, reward functions, metrics, and algorithms. In the initial release, we provide support for 6 different NLP tasks, 16 evaluation metrics and rewards, and 4 RL algorithms.

### 3.1 ENVIRONMENTS: GENERATION AS A TOKEN-LEVEL MDP

Each environment is an NLP task: we are given a supervised dataset $\mathcal{D} = \{(\boldsymbol{x}^i, \boldsymbol{y}^i)\}_{i=1}^N$ of $N$ examples, where $\boldsymbol{x} \in \mathcal{X}$ is an language input and $\boldsymbol{y} \in \mathcal{Y}$ is the target string. Generation can be viewed as a Markov Decision Process (MDP) $\langle \mathcal{S}, \mathcal{A}, \mathcal{R}, P, \gamma, T \rangle$ using a finite vocabulary $\mathcal{V}$. Each episode in the MDP begins by sampling a datapoint $(\boldsymbol{x}, \boldsymbol{y})$ from our dataset and ends when the current time step $t$ exceeds the horizon $T$ or an end of sentence (EOS) token is generated. The input $\boldsymbol{x} = (x_0, \cdots, x_m)$ is a task-specific prompt that is used as our initial state $\boldsymbol{s}_0 = (x_0, \cdots, x_m)$, where $\boldsymbol{s}_0 \in \mathcal{S}$ and $\mathcal{S}$ is the state space with $x_m \in \mathcal{V}$. An action in the environment $a_t \in \mathcal{A}$ consists of a token from our vocabulary $\mathcal{V}$. The transition function $P : \mathcal{S} \times \mathcal{A} \to \Delta(\mathcal{S})$ deterministically appends an action $a_t$ to the end of the state $\boldsymbol{s}_{t-1} = (x_0, \cdots, x_m, a_0, \cdots, a_{t-1})$. This continues until the end of the horizon $t \leq T$ and we obtain a state $\boldsymbol{s}_T = (x_0, \cdots, x_m, a_0, \cdots, a_T)$. At the end of an episode a reward $\mathcal{R} : \mathcal{S} \times \mathcal{A} \times \mathcal{Y} \to \mathbb{R}^1$ that depends on the $(\boldsymbol{s}_T, \boldsymbol{y})$ (e.g., an automated metric like PARENT Dhingra et al. (2019)) is emitted. RL4LMs provides an OpenAI gym (Brockman et al., 2016) style

API for an RL environment that simulates this LM-Based MDP formulation. This abstraction allows for new tasks to be added quickly with compatibility across all implemented algorithms.

## 3.2 REWARD FUNCTIONS AND EVALUATION METRICS

Because RL4LMs provides a generic interface for per-token or per-sequence generation rewards, it is possible to quickly apply a wide array of RL algorithms to a similarly diverse range of textual metrics-as-rewards. Specifically, we provide interfaces to 1) **n-gram overlap metrics** metrics such as ROUGE (Lin, 2004), BLEU (Papineni et al., 2002), SacreBLEU (Post, 2018), METEOR (Banerjee & Lavie, 2005); (2) **model-based semantic metrics** such as BertScore (Zhang et al., 2019) and BLEURT (Sellam et al., 2020) which generally provide higher correlation with human judgment; 3) **task-specific metrics** such as CIDER (Vedantam et al., 2015), SPICE (Anderson et al., 2016) (for captioning/commonsense generation), PARENT (Dhingra et al., 2019) (for data-to-text) and SummaCZS (Laban et al., 2022) (for factuality of summarization); 4) **diversity/fluency/naturalness metrics** such as perplexity, Mean Segmented Type Token Ratio (MSSTR) (Johnson, 1944), Shannon entropy over unigrams and bigrams (Shannon, 1948), the ratio of distinct n-grams over the total number of n-grams (Distinct-1, Distinct-2) and count of n-grams that appear only once in the entire generated text (Li et al., 2015); 5) **task-specific, model-based human preference metrics** such as classifiers trained on human preference data collected in the methodology of Ouyang et al. (2022).

## 3.3 ON-POLICY ACTOR-CRITIC ALGORITHMS

RL4LMs supports fine-tuning and training LMs from scratch via on-policy actor-critic algorithms on language environments. Formally, this class of algorithms allows us to train a parameterized control policy defined as $\pi_\theta : \mathcal{S} \to \Delta(\mathcal{A})$, a function that attempts to select an action in a given state so as to maximize long term discounted rewards over a trajectory $\mathbb{E}_\pi[\sum_{t=0}^{T} \gamma^t \mathcal{R}(\boldsymbol{s}_t, a_t)]$. Our benchmark experiments focus on fine-tuning a pre-trained LM denoted as $\pi_0$ from which we initial our agent's policy $\pi_\theta = \pi_0$. Similarly, the value network $V_\phi$ used to estimate the value function is also initialized from $\pi_0$ except for the final layer which is randomly initialized to output a single scalar value. As with other deep RL actor-critic algorithms, we define our value and Q-value functions as $V_t^\pi = \mathbb{E}_{a_t \sim \pi}[\sum_{\tau=t}^{T} \gamma R(\boldsymbol{s}_\tau, a_\tau, \boldsymbol{y})], Q_t^\pi(\boldsymbol{s}_t, a_t) = R(\boldsymbol{s}_t, a_t, \boldsymbol{y}) + \gamma \mathbb{E}_{s_{t+1} \sim P}[V_{t+1}^\pi(\boldsymbol{s}_{t+1})]$ leading to a definition of our advantage function as $A_t^\pi(\boldsymbol{s}, a) = Q_t^\pi(\boldsymbol{s}, a) - V_t^\pi$. To increase training stability, advantage is appoximated using Generalized Advantage Estimation (Schulman et al., 2015b).

Given an input-output pair $(\boldsymbol{x}, \boldsymbol{y})$ and generation predictions from our agent; because the environment rewards are sequence-level and sparse, following Wu et al. (2021a) we regularize the reward function using a token-level KL penalty for all on-policy algorithms, to prevent the model from deviating too far from the initialized LM $\pi_0$. Formally, the regularized reward function is:

$$\hat{R}(\boldsymbol{s}_t, a_t, \boldsymbol{y}) = R(\boldsymbol{s}_t, a_t, \boldsymbol{y}) - \beta \text{KL}\left(\pi_\theta(a_t|\boldsymbol{s}_t)||\pi_0(a_t|\boldsymbol{s}_t)\right) \tag{1}$$

where $\hat{R}$ is the regularized KL reward, $\boldsymbol{y}$ is gold-truth predictions, $\text{KL}\left(\pi_\theta(a_t|\boldsymbol{s}_t)||\pi_0(a_t|\boldsymbol{s}_t)\right) = \left(\log \pi_0(a_t|\boldsymbol{s}_t) - \log \pi_\theta(a_t|\boldsymbol{s}_t)\right)$ and the KL coefficient $\beta$ is dynamically adapted (Ziegler et al., 2019). Further details on actor-critic methods can be found in Appendix A.

## 4 NLPO: NATURAL LANGUAGE POLICY OPTIMIZATION

Language generation action spaces are orders of magnitude larger than what most discrete action space RL algorithms are designed for (Ranzato et al., 2016; Ammanabrolu, 2021), e.g., GPT-2/3 and T5 have a vocabulary size of 50K and 32K respectively. We hypothesize that the size of the action space is a core cause of instability when training LMs with existing RL methods. To address this issue, we introduce NLPO (Natural Language Policy Optimization), which is inspired by work on action elimination/invalid-action masking (Zahavy et al., 2018; Huang & Ontañón, 2020; Ammanabrolu & Hausknecht, 2020). NLPO, a parameterized-masked extension of PPO, learns to mask out less relevant tokens in-context as it trains. NLPO accomplishes this via top-$p$ sampling, which restricts tokens to the smallest possible set whose cumulative probability is greater than the probability parameter $p$ (Holtzman et al., 2018).

Specifically, NLPO maintains a *masking policy* $\pi_\psi$: the masking policy is a copy of the current policy ($\pi_\theta$), but is updated only every $\mu$ steps. A parameterized-invalid-mask is created from $\pi_\psi$ by first

selecting the top-$p$ tokens from the vocabulary, [4] and then applying an invalid-mask to the remaining tokens—i.e. setting their probabilities to zero when sampling actions from $\pi_\theta$ during training; this periodic updating policy $\pi_\psi$ is inspired by off-policy Q-learning algorithms (Andrychowicz et al., 2017), providing the policy $\pi_\theta$ with an additional constraint that balances between the benefits of containing more task relevant information than the KL penalty derived from $\pi_0$ and the risk of reward hacking. We provide pseudocode in Algorithm 1 (green portions highlight the differences with PPO).

---

**Algorithm 1** NLPO - Natural Language Policy Optimization

---

**Input:** Dataset $\mathcal{D} = \{(\boldsymbol{x}^i, \boldsymbol{y}^i)\}_{i=1}^N$ of size $N$
**Input:** initial policy parameters $\pi_{\theta_0}$
**Input:** initial LM $\pi_0$
**Input:** initial value function parameters $V_{\phi_0}$
**Input:** initialize parameterized masked policy $\pi_{\psi_0}(\cdot|\cdot, \pi_{\theta_0})$ with parameterized top-$p$ policy $\pi_{\theta_0}$
**Input:** policy update frequency $\mu$
**repeat**
    Sample mini-batch $\mathcal{D}_m = \{(\boldsymbol{x}^m, \boldsymbol{y}^m)\}_{m=1}^M$ from $\mathcal{D}$
    Collect trajectories $\mathcal{T}_m = \{\tau_i\}$ by running policy $\pi_{\psi_n}$ in for batch $\mathcal{D}_m$ in env.     ▷ Eq.6
    Compute Preference and KL penalty rewards $\hat{R}_t$     ▷ Eq. 1
    Compute the advantage estimate $\hat{A}_t$     ▷ Sec. 3.3
    Update the policy by maximizing the PPO-Clip objective:

$$\pi_{\theta_{m+1}} = \text{argmax}_\theta \frac{1}{|\mathcal{D}_m|T} \sum_{\tau \in \mathcal{D}_m} \sum_{\tau=0}^T \min\left(r_t(\theta)A^{\pi_{\theta_m}}, \text{clip}(r_t(\theta), 1-\epsilon, 1+\epsilon)A^{\pi_{\theta_m}}\right)$$

    where $r_t(\theta) = \frac{\pi_\theta(a_t|s_t)}{\pi_{\theta_m}(a_t|s_t)}$.
    Update the value function:

$$V_{\phi_{m+1}} = \text{argmin}_\phi \frac{1}{|\mathcal{D}_m|T} \sum_{\tau \in \mathcal{D}_m} \sum_{t=0}^T \left(V_\phi(s_t) - \hat{R}_t\right)^2$$

    Update the parameterized masked policy every $\mu$ iterations:

$$\pi_{\psi_{n+1}}(\cdot|\cdot, \pi_{\theta_{m+1}})$$

**until** convergence and **return** $\pi_\theta$

---

## 5   GRUE (GENERAL REINFORCED-LANGUAGE UNDERSTANDING EVAL)

**GRUE** is a collection of 7 generative NLP tasks. To combat reward hacking for any single metric, each task is evaluated at test time according to a task-specific mix of metrics, detailed in Table 1. The metrics span two categories. **Task preference metrics** capture how well the models produce generations that satisfy the desiderata of the specific generation task, e.g., for Commongen, if the generations contain all the required words, or for IMDB, how positive the generated completions are. **Naturalness metrics** capture fluency, readability, etc. and provide perspective on factors beyond semantics. At training time, there are no special restrictions: models are free to use the supervised data, compute metrics on intermediate generations, etc. Train/val/test splits follow the original works. All results are averaged over multiple seeds, with exact counts being found in Appendix B.

**Experimental Setup.** We use RL4LMs to test a large range of algorithms on the GRUE benchmark. Specifically: We compare 3 algorithms for direct fine-tuning — Supervised, PPO,[5] and NLPO. In

---

[4]$\pi_\psi$ could be trained with alternate sampling techniques like top-$k$ or beam search (or even hard-coded via rules by domain experts), though we find top-$p$ sampling to be most effective in practice.

[5]We consider PPO representative of the present state-of-the-art — in particular, we do not consider the popular REINFORCE (Willianms, 1988; Williams, 1992), as recent works have shown PPO to be strictly superior to REINFORCE in multiple domains (Schulman et al., 2017)

| Dataset | Task | Input | Output | Task Preference Metrics(s) | Naturalness Metrics(s) |
|---|---|---|---|---|---|
| IMDB (Maas et al., 2011) | Text Continuation | Partial Movie Review | A positive completion of the movie review. | Learned Sentiment Classifier | Perplexity (GPT-2) |
| CommonGEN (Lin et al., 2020) | Generative Commonsense | Concept Set | A sentence coherently using all input concepts. | CIDER; ROUGE-2,L; BLEU-3,4; METEOR; Coverage | SPICE |
| CNN Daily Mail (Hermann et al., 2015) | Summarization | News Article | Summarized article. | SummaCZS; ROUGE-1, 2, L, LSum; METEOR; BLEU | BertScore |
| ToTTo (Parikh et al., 2020) | Data to Text | Highlighted Wiki Table | Factually accurate text describing the information. | SacreBLEU; PARENT | BLEURT |
| WMT-16 (en-de) (Bojar et al., 2016) | Machine Translation | Text (English) | Translated text (German). | TER; cHRF; ROUGE-1, 2, L, LSum, METEOR; SacreBLEU, BLEU | BertScore |
| NarrativeQA (Kočiskỳ et al., 2018) | Question Answering | Question Context (a Story) | Abstractive answer to the question. | ROUGE-1, 2, L, LSum, LMax; METEOR; BLEU; SacreBLEU | BertScore |
| DailyDialog (Li et al., 2017) | Chitchat Dialogue | Dialogue History | A conversational response | METEOR; Learned Intent Classifier | BertScore |

Table 1: **GRUE Benchmark using RL4LMs** showing the various tasks, input and output types, and the metrics used. We note that we test RL algorithms on these tasks for a wider range of possible rewards than just the task specific ones shown here. Unless specified, datasets are in English.

addition, we consider a hybrid approach of supervised learning and our RL methods by applying PPO and NLPO on checkpoints that have been fine-tuned in a supervised fashion—we call these Supervised+PPO, Supervised+NLPO. As an additional baseline, we additionally run zero-shot evaluations where we design prompts which aim to elicit task-specific generations, but with no training data or parameter updates.

For each task, to isolate the effect of training method, we select a single pre-trained LM backbone. For IMDB text continuation we use GPT-2 (117m parameters), and for the rest of the tasks we use T5-base (220m parameters). For our RL models (PPO, NLPO, Supervised+PPO, Supervised+NLPO), for a thorough investigation of how reward-hacking might interplay with GRUE, we run a separate set of experiments optimizing multiple task rewards for each task independently, e.g., for Commongen which has 6 task rewards (CIDER, ROUGE-2, ROUGE-L, BLEU-3, BLEU-4, METEOR) we run 6 different experiments optimizing each metric independently and report all possible metrics seen in Table 1 regardless of which individual metric was being optimized for.

**Human Participant Study.** We gather human judgments for five of the tasks in GRUE. In doing so, our goals are 1) to validate that the automated metrics we selected for GRUE correlate with human judgments with respect to relative ranking between models; and 2) to provide additional empirical comparisons regarding NLPO vs. PPO, ablations to study the effects of the KL naturalness penalty, etc. We specifically consider IMDB, Commongen, ToTTo, DailyDialog, and CNN Daily Mail. For each individual sample in a task, we ask 3 unique human raters to provide Likert judgments of 1) quality, i.e., for the specific task, how correct/appropriate is the generation, given the context, and 2) fluency, i.e., how well-written is the generation. We used Amazon Mechanical Turk, and paid crowdworkers a minimum of $15/hr. More details, including qualification information, interface screenshots, instructions, etc. are given in the corresponding Appendicies.

### 5.1 RESULTS ON GRUE: WHICH ALGORITHM SHOULD BE USED TO LEARN PREFERENCES?

Figures 2(a), 2(b) present the results on GRUE, split into task metrics and naturalness metrics, and Tables 2, 3 highlight key results via ablation studies. Full results are available in Appendix B. For text continuation and summarization, with non-trivial zero-shot performance, RL tends to perform better than supervised training, but for tasks like Commongen and ToTTo, which have very low zero-shot performance, supervised training performs best—with both approaches outperforming zero-shot.

However, **using RL+Supervised learning in conjunction works best;** NLPO+supervised and PPO+supervised usually always outperforms NLPO/PPO (or supervised in isolation) across both task metrics and naturalness metrics. Supervised warm-starting is particularly effective for Commongen and ToTTo, which our results suggest are more prone to reward hacking. The one exception to this trend is DailyDialog where the RL models outperform warm-started Supervised+RL models likely due to the low performance of the Supervised models. We note that Supervised+NLPO using a

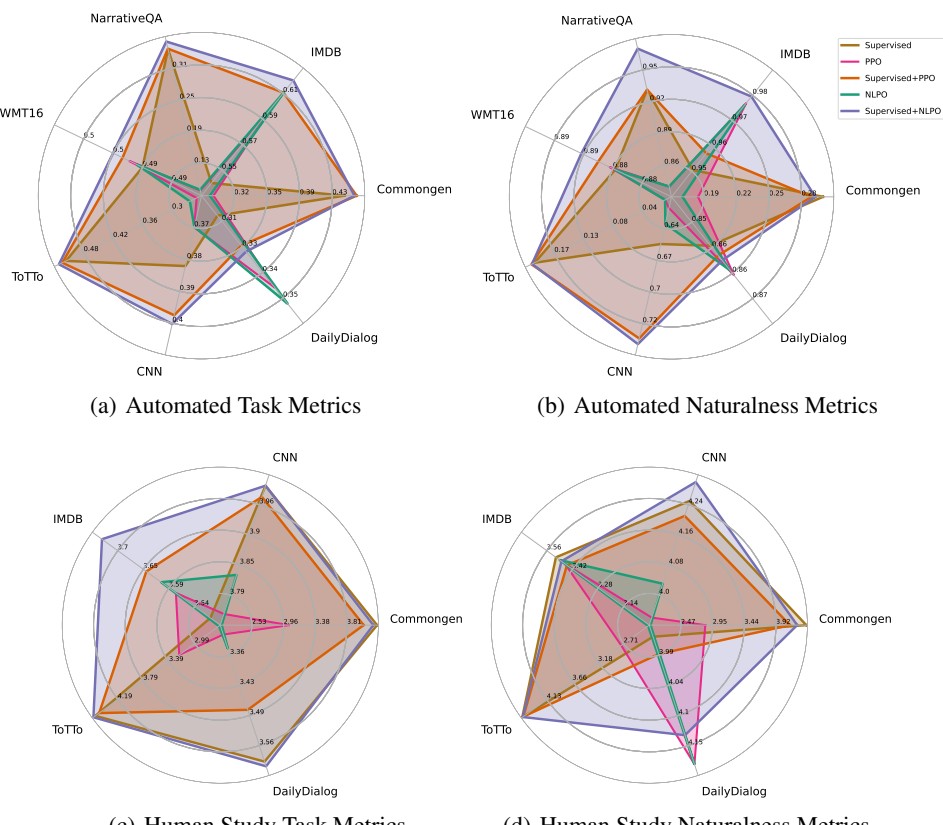

Figure 2: Summarized results via automated metrics across all 7 GRUE tasks for each of the 5 algorithms we consider, and human participant studies for the 5 tasks suitable for human studies. Test results are averaged over all the respective metrics seen in Table 1.

Table 2: **Key questions answered using GRUE + RL4LMs:** This table summarizes the results found in the ablations and Fig. 2 and provides an overview of the questions we ask in Section 5: which tasks require warm starts or are easily reward hackable; when to use RL over Supervised, when to use both; and when to use NLPO over PPO. All conclusions drawn are the result of statistical analysis as discussed in the experimental setup.

| Questions | Tasks | | | | | | |
|---|---|---|---|---|---|---|---|
| | IMDB | CommonGen | CNN/DM | ToTTo | WMT16 | NarQA | Dialog |
| Needs Warm Start | ✗ | ✓ | ✗ | ✓ | ✗ | ✓ | ✗ |
| Easily reward hackable? | ✓ | ✗ | ✗ | ✗ | ✗ | ✗ | ✓ |
| RL > Sup (auto)? | ✓ | ✗ | ✗ | ✗ | ✗ | ✗ | ✓ |
| RL > Sup (human)? | ✓ | ✗ | ✗ | ✗ | - | - | ✓ |
| Sup+RL > Sup (auto)? | ✓ | ✓ | ✓ | ✓ | ✓ | ✓ | ✗ |
| Sup+RL > Sup (human)? | ✓ | ✗ | ✓ | ✓ | - | - | ✗ |
| Sup+NLPO > Sup+PPO (auto)? | ✓ | ✓ | ✓ | ✓ | ✓ | ✓ | ✓ |
| Sup+NLPO > Sup+PPO (human)? | ✓ | ✓ | ✓ | ✓ | - | - | ✓ |

| Ablation | Sentiment | Perplexity |
|---|---|---|
| Zero Shot | 0.489 | 32.171 |
| Supervised | 0.539 | 35.472 |
| PPO | 0.602 | 33.816 |
| NLPO | 0.611 | 33.832 |
| Warm Starting (Sec. 5.1) | | |
| PPO+Supervised | 0.626 | 35.049 |
| NLPO+Supervised | 0.620 | 34.816 |
| Data Budget (Reward trained on 10% of data, Sec. 5.3) | | |
| PPO | 0.598 | 35.929 |
| NLPO | 0.599 | 33.536 |
| Removing NLPO Top-$p$ Constraints (Sec. 5.2) ($p = 1$ is equivalent to PPO, $p = 0.9$ is NLPO) | | |
| NLPO $p = 0.1$ | 0.579 | 32.451 |
| NLPO $p = 0.5$ | 0.588 | 32.447 |
| Removing KL Constraints (Sec. 5.2) | | |
| PPO-no-KL | 0.838 | 41.897 |
| NLPO-no-KL | 0.858 | 41.429 |
| Discount Ablations ($\gamma = 1$) (Sec. 5.4) | | |
| PPO | 0.651 | 41.035 |
| NLPO | 0.624 | 43.720 |

Table 3: IMDB Ablation Results.

T5-base (220m parameter) LM currently outperforms all the models on the ToTTo leaderboard, many of which have $\geq$ 3b parameter supervised models—suggesting that RL is parameter efficient as well. In these cases, it is critical that the initial policy already contain (some) signal for the task due to it being used as a KL constraint and masking constraint in NLPO. If the mask contains no initial priors about task specific language, it will be eliminating the wrong actions—a better initial policy leads to better RL performance downstream.

**Human agreement with automated metrics.** As human judgments can be noisy, we run additional statistical analysis such as measuring inter-annotator agreement, via Krippendorf's alpha score, and using a one-way ANOVA followed by a post-hoc Tukey HSD test to measure if differences in means of average scores between pairs of models are significant. We find that trends in our human evaluations generally match those seen in the automated metrics for both task and naturalness metrics (see Figures 2(c), 2(d) which summarize Appendix Tables 10,15,21,26, 35—Supervised+NLPO > Supervised ≥ Supervised+PPO > NLPO ≥ PPO > Zero-shot—with the exception of Supervised outperforming Supervised+PPO on 2 out of 5 tasks when automated metrics would indicate that Supervised+PPO outperforms Supervised on all of the tasks. We draw two conclusions from this: (1) if the generated text is above a certain threshold of naturalness, the automated metrics *usually* correlate with human judgements; (2) usually but not always as seen in the relative performance of Supervised and Supervised+PPO, potentially indicating reward hacking behaviors undetected by automated metrics but caught by human preference feedback.

## 5.2 PREFERENCE REWARD LEARNING, SELECTION, AND HACKING

While the GRUE benchmark's metric for each task is an average over several measures, the RL models we trained optimized only a single metric independently. Thus, we can empirically investigate which metric for which GRUE produces the best results. We observe that many possible single metric rewards provide task performance gains over supervised methods (results shown in Fig. 3(a), 2(c) are averaged across these reward functions) with the condition that the text is also coherent and natural.

**Which constraints best prevent reward hacking?** The reward function in Equation 1 balances a task-specific reward with a KL constraint — models are penalized from straying too far from a base LM in their pursuit of high reward (Table 3 and Appendix Table 5) clearly show that if KL constraints are removed entirely, models reward hack). But which model works best as a base regularizing LM? When the initial policy (i.e., the raw, pretrained model) has low performance on the task, the KL penalty pushes the policy towards nonsense, e.g. on Commongen and ToTTo the trained policy learns to simply repeat portions of the input (as seen in Tables B.4.5, B.6.4). This behavior is mitigated if the base regularizing LM is the supervised model—the reward encourages the policy to balance the task-specific reward and a more reasonable regularization term. Deriving KL penalties from warm-started initial policies is critical for performance on such tasks.

**PPO vs. NLPO.** Figure 2 shows that NLPO generally outperforms PPO and supervised, especially when applied after supervised training. We hypothesize that the primary reason for NLPO's improved performance and stability is because the masking policy provides an additional constraint for the current policy. This constraint is not based on the initial untuned policy like the KL penalty but of the policy from $\mu$ iterations ago and likely contains more task-relevant information learned during RL training. Table 3 (and Appendix Table 8) shows how performance increases up to a point and then decreases as $p$ in top-$p$ sampling is increased for the masking policy, relaxing the constraint by eliminating less tokens at each step, implying that there is a balance to be found in how much the model should be constrained during RL training.

**Human Preference Reward Learning.** To this point, our experiments have largely focused on optimizing evaluation metrics that correlate with human judgments, e.g., METEOR. Here: we additionally test how well preferences can be learned from direct human feedback. For this, we focus on Commongen — a GRUE dataset well-suited for displaying differences due to human preferences. First, we randomly select prompts from the Commongen train dataset and sample a single completion from both the Supervised and Supervised+NLPO models. We then present the prompt and the two completion candidates to 3 unique crowdworkers and ask them to select which one they prefer with respect to commonsense/fluency for 417 unique pairs (Krippendorf $\alpha = .28$). We use this data to train a reward model, T5-11B Raffel et al. (2020), on the balanced binary classification task of predicting which of the pair was preferred by a majority of 3 annotators, conditioned on the prompt and completion. The resulting model achieved 69.5 test ROC AUC suggesting it indeed captures average human preferences. Additional details on this process are found in Appendix B.4.4. We train Supervised+RL with a METEOR-only reward as a baseline, and compare it to a reward function that uses the fine-tuned T5-11B model. Finally, we rerun the same pairwise preference collection procedure—this time sampling from Commongen test—with human participants to compare the generations from a preference optimized RL policy to the previously best Supervised+NLPO policy. Comparing the METEOR-only to the preference model, the generations produced by the human

feedback model are preferred in 682 cases, compared to the METEOR-only model which is preferred in 587 cases ($p < 0.01$ the models are equally preferred). This implies that this pipeline of collecting preferences, training a reward, and further tuning the policy improves alignment to human preferences.

## 5.3 DATA BUDGET: IMPROVE YOUR REWARD OR GATHER MORE DEMONSTRATION?

Given a fixed data collection budget, is it more efficient to gather feedback to improve a learned reward function or to gather more expert demonstrations? We use the IMDB text continuation task as a case study. In the IMDB task, a model is given a partial movie review as a prompt, and is asked to continue it as positively as possible (even if the prompt was negative). The original dataset consists of movie reviews and sentiment labels of positive, negative, or neutral. A DistilBERT (Sanh et al., 2019) classifier is trained on these labels and used to provide sentiment scores on how positive a given piece of text is, which serves as the task reward. The trade-off is between gathering more: 1) sentiment labels (improving the reward); or 2) positive sentiment reviews (improving supervised training).

We train a classifier on varying amounts of training data and evaluate on the held out test dataset—finding as expected that more training data improves test accuracy and so results in a higher quality reward. We then use each of these rewards of varying quality during RL training, and evaluate using the same metric as GRUE (i.e., a classifier trained with the entire training set). As seen in Table 3, we find that improving the reward quality improves LM performance as well. Further, we trained a supervised model with at least as many samples used to train each of these reward classifiers. We find that **a learned reward function enables greater performance when used as a signal for an RL method than a supervised method trained with 5 times more data.** This implies that improving reward models can be more data efficient than collection expert demonstrations for a task—and that's not accounting for the fact that assigning sentiment labels is likely a simpler task than writing full demonstrations. Further details on this ablation are found in Appendix Table 7.

## 5.4 PRACTICAL CONSIDERATIONS: WHICH IMPLEMENTATION DETAILS MATTER MOST?

**Generation as a token-level MDP, not a bandit environment.** Most recent works that tune LMs using RL do so by calculating a reward for all the tokens in the sentence (Wu et al., 2021a; Ouyang et al., 2022; Lu et al., 2022). This setting is equivalent to a bandit feedback environment where the action space is the space of all possible generations for the task (Sutton & Barto, 2018). This type of environment can be simulated within our RL formulation by setting the discount factor $\gamma = 1$. Table 3 (and Appendix Table 6) shows that this causes instability in training with respect to naturalness in both PPO and NLPO for IMDB. Our standard setting is $\gamma = 0.95$ when calculating discounted rewards-to-go in the token-level MDP formulation, which reduces the magnitude of the reward that is applied to tokens selected at the beginning. The sentiment scores are approximately the same between both settings but the naturalness of language in the bandit setting is significantly less—indicating that discounting rewards with $\gamma < 1$ via a token-level MDP formulation is at least sometimes more effective for language generation.

**Dropout and Sampling.** We found two other implementation details to be critical for stability of RL training. The first is dropout, which in its standard form was found to cause instability in policy gradient methods in continuous control settings by Hausknecht & Wagener (2022). We find a similar effect when using dropout when RL training LMs as well, with training loss often diverging for dropout $> 0$ in training. The second important detail, particularly affecting the machine translation task, is sampling methods. We find that using the same sampling methods during exploration and inference is critical to translating training performance to test performance–else the model exhibits high train rewards but low test metrics.

## 6 CONCLUSIONS

We're hopeful that the GRUE benchmark and the RL4LMs library can push progress in aligning language models to human preferences via RL methods by providing the community with a standard means of comparing methods. Furthermore, we're optimistic that, as the stability and consistency of training improves, our methods provide a path towards iterative improvement of language technologies, with deployment, user feedback collection, and re-optimization enabling better user experiences when interacting with generative models.

## 7 ACKNOWLEDGEMENTS

We'd like to acknowledge the support of DARPA MCS program through NIWC Pacific (N66001-19-2-4031), Google Cloud Compute, and the ReViz team at the Allen Institute for AI. KB is supported by NSF under grant No. 2127309 to the Computing Research Association for the CIFellows Project. Rajkumar is funded by the Federal Ministry of Education and Research of Germany and the state of North-Rhine Westphalia as part of the Lamarr-Institute for Machine Learning and Artificial Intelligence.

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

CONTENTS

## A  ON-POLICY ALGORITHM IMPLEMENTATION DETAILS

### A.1  PPO DETAILS

Given discussion and equations in Section 3.3, we further note that we follow (Ziegler et al., 2019) and dynamically adapt the KL coefficient $\beta$ during training where,

$$e_t = \text{clip}\left(\frac{\text{KL}\left(\pi(a_t|s_t)||\pi_0(a_t|s_t)\right) - \text{KL}_{\text{target}}}{\text{KL}_{\text{target}}}, -0.2, 0.2\right) \tag{2}$$

$$\beta_{t+1} = \beta_t(1 + \text{K}_\beta e_t) \tag{3}$$

where $\text{KL}_{\text{target}}$ is user-specified KL divergence between initial model $h$ and current policy $\pi$ and $\text{K}_\beta$ is rate of update which we generally set to $0.2$ in our experiments.

To increase stability during training, we further use Generalized Advantage Estimation (GAE) (Schulman et al., 2015b) and define the advantage estimator $\hat{A}(s_n, a_n)$ based on the Temporal Difference residual as:

$$\delta_t = r(s_t, a_t) + V_\phi(s_{t+1}) - V_\phi(s_t). \tag{4}$$

$$\hat{A}(s_n, a_n) = \sum_{t=0}^{\infty} \lambda^t \delta_{n+t}, \tag{5}$$

where $\lambda$ provides the trade-off between bias and variance.

### A.2  NLPO DETAILS

NLPO learns to mask irrelevant language by maintaining a *masking policy* $\pi_\psi$: the masking policy is a copy of the current policy ($\pi_\theta$), but is updated only every $\mu$ steps. Given $Z(\pi_\theta) = \sum_{a \in \mathcal{V}} \pi_{\theta_0}(a|s)$ the normalization value of the sum of probabilities of all action $a \in \mathcal{A}$ given a particular State $s \in \mathcal{S}$, let the parameterized top-$p$ vocabulary $\mathcal{V}_{\pi_\theta}^p \subset \mathcal{V}$ be the subset of the vocab, consisting of the top-$p$ highest probability vocabulary tokens with respect to $\pi_\theta$. Formally, let $Z^p$ be the normalization value for the parameterized top-$p$ vocabulary, can be defined as the subset of tokens that maximizes $Z^p(\pi_\theta) = \sum_{a \in \mathcal{V}_{\pi_\theta}^k} \pi_\theta(a|s)$. Then optimizing a policy according to the parameterized top-$p$ vocabulary can be defined as:

$$\pi_\psi(\cdot|s, \pi_\theta) = \begin{cases} \pi_\theta(\cdot|s)/Z^p(\pi_\theta) & \text{if } a \in V_{\pi_\theta}^p \text{ and } Z(\pi_\theta) \\ 0 & \text{otherwise.} \end{cases} \tag{6}$$

# B    EXPERIMENTAL DETAILS

## B.1    CROWDWORKING DETAILS

**Qualification round**    We ran a qualification round using the IMDB task. We opened the qualification around to users from {AU, CA, NZ, GB, US} with 5K prior approved HITs and a minimum acceptance rate of 97% on their previous HITs. We gathered judgments over 600 generations from 3 annotators per generation. One of the authors of this paper also completed 17 random HITs to serve as a proxy for "ground truth." After gathering these annotations, we selected workers who: 1) didn't significantly disagree with other annotators on the same instance more than 20% of the time; 2) who completed at least 5 HITs; 3) who didn't disagree with the author annotator on the 17 HITs by more than 1 point; and 4) (likely) spent a reasonable amount of time reading the instructions/examples provided. In the end, 56 annotators were qualified. Additional per-task details are provided in the per-task sections of the Appendix.

**Compensation details**    As per Amazon Mechanical Turk policy, annotators were compensated on a per-HIT basis. In addition, we used a timing script to estimate hourly wages to ensure our target of $15/hr was met. In cases where this minimum hourly rate was not met, we manually assigned bonuses.

## B.2    GRUE EXPERIMENT SETUP

We benchmark 5 training algorithms on 6 tasks (see Table 1) using either an encoder model (eg. GPT-2) or encoder-decoder model (eg. T5). We train policies using PPO, NLPO with variations of whether supervised pre-training is applied before RL fine-tuning and compare against supervised policy. The choice of LM is based on the type of task. For IMDB text continuation, we use GPT-2 and T5 for rest of the tasks. We use two separate LM models as actor and critics networks (i.e. no shared layers) in which the critic network has an additional linear layer mapping last token's hidden representation to a scalar value. We use AdamW optimizer Loshchilov & Hutter (2017) with fixed learning rate and no scheduling.

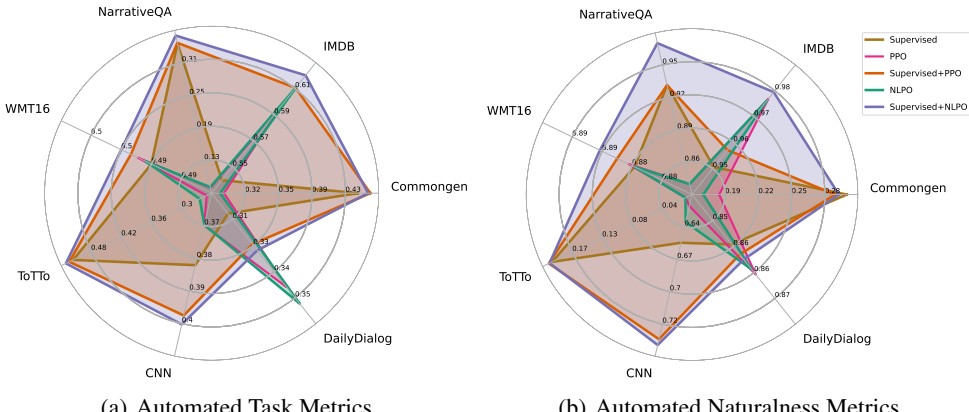

(a) Automated Task Metrics          (b) Automated Naturalness Metrics

Figure 3: Summarized results via automated metrics across all 7 GRUE tasks for each of the 5 algorithms we consider, and human participant studies for the 5 tasks suitable for human studies. We break up the metrics into task-specific, e.g. average positive sentiment for IMDB task, and naturalness metrics, such as perplexity and human perceived coherence for the human rated metrics. This plot differs from Figure 2 as this one averages over over multiple reward functions per each task.

| Model Params | value |
|---|---|
| supervised | batch size: 64
epochs: 10
learning rate: 0.00001 |
| ppo | steps per update: 1280
total number of steps: 64000
batch size: 64
epochs per update: 5
learning rate: 0.000001
discount factor: 0.99
gae lambda: 0.95
clip ratio: 0.2
value function coeff: 0.5 |
| nlpo | steps per update: 1280
total number of steps: 64000
batch size: 64
epochs per update: 5
learning rate: 0.000001
discount factor: 0.99
gae lambda: 0.95
clip ratio: 0.2
top mask ratio: 0.9
target update iterations: 5 |
| decoding | sampling: true
top k: 50
min length: 48
max new tokens: 48 |
| tokenizer | padding side: left
truncation side: left
max length: 64 |

Table 4: **IMDB Hyperparams**: Table shows a list of all hyper-parameters and their settings

## B.3   IMDB

### B.3.1   SETUP

We consider IMDB dataset for the task of generating text with positive sentiment. The dataset consists of 25k training, 5k validation and 5k test examples of movie review text with sentiment labels of positive and negative. The input to the model is a partial movie review text (upto 64 tokens) that needs to be completed (generating 48 tokens) by the model with a positive sentiment while retaining fluency. For RL methods, we use a sentiment classifier Sanh et al. (2019) that is trained on pairs of text and labels as a reward model which provides sentiment scores indicating how positive a given piece of text is. For supervised Seq2Seq baselines, we consider only the examples with positive labels. We chose GPT-2 as LM for this task as it is more suited for text continuation than encoder-decoder LMs (eg. T5). We use top-k sampling with $K = 50$ as the decoding method and for fair comparison, we keep this setting for all methods. For PPO and NLPO models, we train for $64k$ steps in total and update policy and value networks every $1280$ steps with a mini-batch size of $64$ and epochs of $5$ per update. We apply adaptive KL controllers with different target KLs of $0.02, 0.05, 0.1, \inf$ with an initial KL co-efficient of $\beta = 0.1$. Table 4 provides an in-depth summary of all hyperparameters and other implementation details.

### B.3.2   RESULTS AND DISCUSSION

**Target KL ablation**   Fig 4 shows learning curves for PPO and NLPO in terms of episodic training reward, corpus level sentiment scores and perplexity scores on validation set averaged for 5 random seeds. It is seen that higher target KL of 0.1 is desired to achieve higher rewards but results in drifting

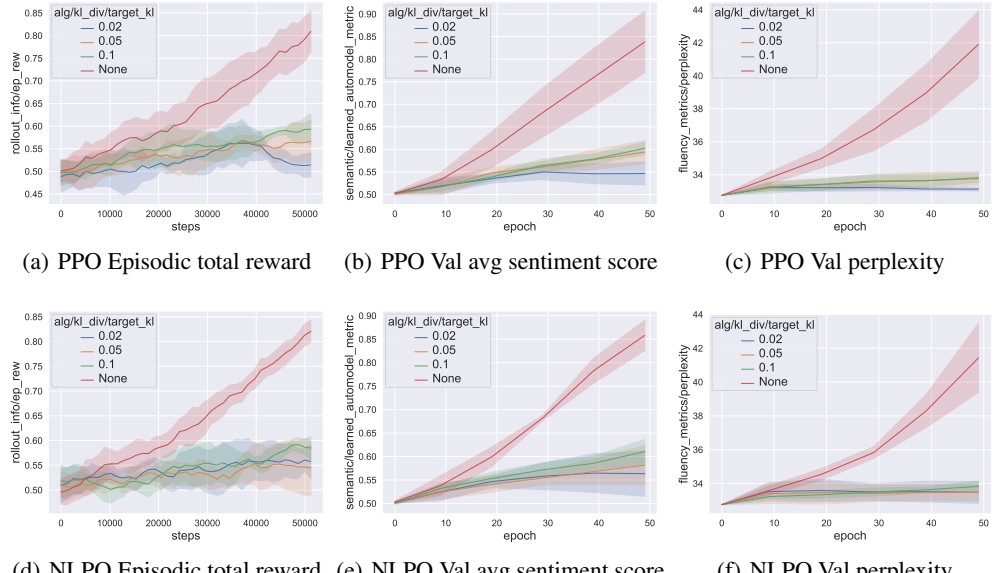

(a) PPO Episodic total reward (b) PPO Val avg sentiment score (c) PPO Val perplexity

(d) NLPO Episodic total reward (e) NLPO Val avg sentiment score (f) NLPO Val perplexity

Figure 4: **Learning Curves**: Averaged learning curves over 5 different runs by varying target KL, shaded regions indicate one standard deviation. (a) shows the rollout episodic total reward during training (b) shows evolution of sentiment scores on the validation split (c) shows evolution of perplexity on the validation split. From (a) and (b), it is seen that higher target KL (0.1) is desired to achieve higher rewards. However, this setting drifts away from the original LM too much and loses fluency. Therefore a lower target KL (0.02 or 0.05) is required to keep the model closer to original LM. Similar trends hold for NLPO but when compared to PPO, it retains lower perplexities and is more stable even with higher KL targets

| Target-KL | Semantic and Fluency Metrics | | Diversity Metrics | | | | | | |
|---|---|---|---|---|---|---|---|---|---|
| | Sentiment Score ↑ | Perplexity ↓ | MSTTR | Distinct$_1$ | Distinct$_2$ | H$_1$ | H$_2$ | Unique$_1$ | Unique$_2$ |
| Zero-Shot | $0.489 \pm 0.006$ | $32.171 \pm 0.137$ | $0.682 \pm 0.001$ | $0.042 \pm 0.001$ | $0.294 \pm 0.001$ | $8.656 \pm 0.004$ | $13.716 \pm 0.003$ | $5063 \pm 14.832$ | $47620 \pm 238$ |
| Supervised | $0.539 \pm 0.004$ | $35.472 \pm 0.074$ | $0.682 \pm 0.001$ | $0.047 \pm 0.001$ | $0.312 \pm 0.002$ | $8.755 \pm 0.012$ | $13.806 \pm 0.016$ | $5601 \pm 57$ | $51151 \pm 345$ |
| **PPO** | | | | | | | | | |
| 0.02 | $0.546 \pm 0.022$ | $33.127 \pm 0.092$ | $0.680 \pm 0.003$ | $0.044 \pm 0.001$ | $0.297 \pm 0.004$ | $8.665 \pm 0.029$ | $13.685 \pm 0.076$ | $5332 \pm 184$ | $48380 \pm 733$ |
| 0.05 | $0.594 \pm 0.022$ | $33.765 \pm 0.367$ | $0.671 \pm 0.005$ | $0.043 \pm 0.001$ | $0.286 \pm 0.009$ | $8.588 \pm 0.066$ | $13.519 \pm 0.103$ | $5171 \pm 190$ | $46336 \pm 1872$ |
| 0.1 | $0.602 \pm 0.012$ | $33.816 \pm 0.233$ | $0.664 \pm 0.007$ | $0.042 \pm 0.001$ | $0.278 \pm 0.005$ | $8.529 \pm 0.037$ | $13.366 \pm 0.119$ | $5108 \pm 204$ | $45158 \pm 961$ |
| inf | $0.838 \pm 0.061$ | $41.897 \pm 1.806$ | $0.577 \pm 0.059$ | $0.034 \pm 0.003$ | $0.197 \pm 0.036$ | $7.737 \pm 0.514$ | $11.866 \pm 0.993$ | $4214 \pm 260$ | $31181 \pm 5524$ |
| **PPO+supervised** | | | | | | | | | |
| 0.1 | $0.626 \pm 0.014$ | $35.049 \pm 0.347$ | $0.668 \pm 0.004$ | $0.048 \pm 0.002$ | $0.307 \pm 0.008$ | $8.704 \pm 0.053$ | $13.656 \pm 0.066$ | $5757 \pm 324$ | $50522 \pm 1514$ |
| inf | $0.796 \pm 0.004$ | $42.916 \pm 1.716$ | $0.617 \pm 0.017$ | $0.038 \pm 0.003$ | $0.233 \pm 0.017$ | $8.149 \pm 0.183$ | $12.733 \pm 0.316$ | $4563 \pm 327$ | $37040 \pm 2507$ |
| **NLPO** | | | | | | | | | |
| 0.02 | $0.564 \pm 0.043$ | $33.477 \pm 0.578$ | $0.679 \pm 0.002$ | $0.043 \pm 0.001$ | $0.294 \pm 0.001$ | $8.649 \pm 0.007$ | $13.688 \pm 0.04$ | $5232 \pm 96$ | $47732 \pm 184$ |
| 0.05 | $0.582 \pm 0.037$ | $33.470 \pm 0.453$ | $0.675 \pm 0.003$ | $0.043 \pm 0.001$ | $0.293 \pm 0.004$ | $8.63 \pm 0.033$ | $13.656 \pm 0.085$ | $5200 \pm 101$ | $47484 \pm 822$ |
| 0.1 | $0.611 \pm 0.023$ | $33.832 \pm 0.283$ | $0.670 \pm 0.002$ | $0.043 \pm 0.002$ | $0.286 \pm 0.006$ | $8.602 \pm 0.049$ | $13.53 \pm 0.076$ | $5179 \pm 196$ | $46294 \pm 1072$ |
| inf | $0.858 \pm 0.029$ | $41.429 \pm 1.825$ | $0.575 \pm 0.048$ | $0.035 \pm 0.005$ | $0.201 \pm 0.028$ | $7.755 \pm 0.379$ | $11.862 \pm 0.808$ | $4389 \pm 609$ | $31714 \pm 4500$ |
| **NLPO+supervised** | | | | | | | | | |
| 0.1 | $0.620 \pm 0.014$ | $34.816 \pm 0.340$ | $0.672 \pm 0.006$ | $0.048 \pm 0.002$ | $0.31 \pm 0.012$ | $8.725 \pm 0.09$ | $13.709 \pm 0.174$ | $5589 \pm 140$ | $50734 \pm 1903$ |
| inf | $0.777 \pm 0.042$ | $41.035 \pm 0.601$ | $0.636 \pm 0.023$ | $0.043 \pm 0.005$ | $0.265 \pm 0.034$ | $8.373 \pm 0.269$ | $12.947 \pm 0.359$ | $5173 \pm 589$ | $43342 \pm 6828$ |

Table 5: **Target KL Ablations**: Mean and standard deviations over 5 random seeds is reported for sentiment scores along with fluency and diversity metrics on validation set. It is seen from perplexity scores that a lower target KL constraint is desired to keep the model closer to the original model. On the otherhand, a higher target KL yields higher sentiment scores at the cost of fluency. inf KL penalty (target KL of inf), model simply learns to generate positive phrases (eg: "I highly recommend this movie to all!", "worth watching") regardless of the context. NLPO achieves better sentiment and perplexity scores than PPO.

away from pre-trained LM and loses fluency. Therefore, a lower target KL (0.02 or 0.05) is required to keep the LM closer to original LM. This is also seen in Table 5 where we presented a comparative analysis of final performance of all models.

**Training data size ablation** We vary the amount of data used to train the reward classifier and the supervised baseline model to understand whether it is more efficient to gather data to improve reward model or to gather expert demonstrations for supervised learning. As observed in Table 7, improving the quality of reward function increases the performance on the overall task better than training with more data for supervised training, indicating that improving reward models is efficient than collect expert demonstrations for supervised training from a data efficiency perspective.

**Discount factor ablation** To understand the effect of discounted vs undiscounted (bandit) environments, we report sentiment and perplexity scores for different values of discount factor (0.5, 0.95 and 1.0) in Table 6 and observe that using a bandit environment (discount factor of 1.0) results in performance loss in the case of NLPO and reward hacking in the case of PPO, indicating that discounted setting (with 0.95) is desired.

**NLPO params** Table. 8 shows ablation on different hyperparameters in NLPO algorithm.

| Gamma | Semantic and Fluency Metrics | | Diversity Metrics | | | | | | |
|---|---|---|---|---|---|---|---|---|---|
| | Sentiment Score ↑ | Perplexity ↓ | MSTTR | Distinct$_1$ | Distinct$_2$ | H$_1$ | H$_2$ | Unique$_1$ | Unique$_2$ |
| Zero-Shot | 0.489 ± 0.006 | 32.371 ± 0.137 | 0.682 ± 0.001 | 0.042 ± 0.001 | 0.294 ± 0.001 | 8.656 ± 0.004 | 13.716 ± 0.003 | 5063 ± 14.832 | 47620 ± 238 |
| PPO | | | | | | | | | |
| 0.5 | 0.511 ± 0.023 | 35.945 ± 0.92 | 0.69 ± 0.001 | 0.044 ± 0.002 | 0.304 ± 0.007 | 8.726 ± 0.041 | 13.793 ± 0.055 | 5304 ± 285 | 49668± 1496 |
| 0.95 | 0.605 ± 0.023 | 33.497 ± 0.447 | 0.666 ± 0.013 | 0.043 ± 0.002 | 0.287 ± 0.008 | 8.575 ± 0.073 | 13.484 ± 0.244 | 5230 ± 363 | 46483 ± 1318 |
| 1.0 | 0.651 ± 0.05 | 41.035 ± 2.885 | 0.691 ± 0.017 | 0.042 ± 0.004 | 0.295 ± 0.031 | 8.697 ± 0.237 | 13.563 ± 0.396 | 5127 ± 460 | 48319 ± 5650 |
| NLPO | | | | | | | | | |
| 0.5 | 0.49 ± 0.01 | 37.279 ± 5.137 | 0.688 ± 0.01 | 0.045 ± 0.002 | 0.312 ± 0.016 | 8.746 ± 0.113 | 13.873 ± 0.25 | 5395 ± 192 | 50828 ± 2506 |
| 0.95 | 0.637 ± 0.013 | 32.667 ± 0.631 | 0.677 ± 0.014 | 0.044 ± 0.002 | 0.288 ± 0.010 | 8.588 ± 0.100 | 13.484 ± 0.236 | 5205 ± 189 | 46344 ± 2688 |
| 1.0 | 0.624 ± 0.039 | 43.72 ± 2.475 | 0.662 ± 0.019 | 0.05 ± 0.007 | 0.3 ± 0.038 | 8.624 ± 0.277 | 13.360 ± 0.537 | 6337 ± 921 | 49441 ± 6520 |

Table 6: **Evaluation of GPT2 with different algorithms on IMDB sentiment text continuation task, discount factor ablations**: Mean and standard deviations over 5 random seeds is reported for sentiment scores along with fluency and diversity metrics. This table measures performance differences for the discount factor. We note that most NLP approaches using RL follow the style of Li et al. (2016); Wu et al. (2021a) and use a discount factor of 1. This is equivalent to reducing the generation MDP to a bandit feedback environment and causes performance loss (in the case of NLPO) and reward hacking and training instability (in the case of PPO).

| Perc Data (size) | Semantic and Fluency Metrics | | Diversity Metrics | | | | | | |
|---|---|---|---|---|---|---|---|---|---|
| | Sentiment Score ↑ | Perplexity ↓ | MSTTR | Distinct$_1$ | Distinct$_2$ | H$_1$ | H$_2$ | Unique$_1$ | Unique$_2$ |
| Zero-Shot | 0.489 ± 0.006 | 32.371 ± 0.137 | 0.682 ± 0.001 | 0.042 ± 0.001 | 0.294 ± 0.001 | 8.656 ± 0.004 | 13.716 ± 0.003 | 5063 ± 14.832 | 47620 ± 238 |
| Supervised | | | | | | | | | |
| 0.0 (0k) | 0.489 ± 0.006 | 32.371 ± 0.137 | 0.682 ± 0.001 | 0.042 ± 0.001 | 0.294 ± 0.001 | 8.656 ± 0.004 | 13.716 ± 0.003 | 5063 ± 14 | 47620 ± 238 |
| 0.1 (1k) | 0.531 ± 0.005 | 34.846 ± 0.123 | 0.685 ± 0.001 | 0.045 ± 0.001 | 0.313 ± 0.004 | 8.775 ± 0.023 | 13.854 ± 0.032 | 5215 ± 62 | 51125 ± 685 |
| 0.5 (5k) | 0.536 ± 0.006 | 35.008 ± 0.229 | 0.684 ± 0.001 | 0.047 ± 0.000 | 0.314 ± 0.002 | 8.764 ± 0.010 | 13.837 ± 0.0178 | 5489 ± 44 | 51284 ± 576 |
| 1.0 (10k) | 0.539 ± 0.004 | 35.472 ± 0.074 | 0.682 ± 0.001 | 0.047 ± 0.001 | 0.312 ± 0.002 | 8.755 ± 0.012 | 13.806 ± 0.016 | 5601 ± 57 | 51151 ± 345 |
| PPO | | | | | | | | | |
| 0.0 (0k) | 0.492 ± 0.01 | 33.57 ± 0.323 | 0.69 ± 0.02 | 0.047 ± 0.001 | 0.321 ± 0.015 | 8.816 ± 0.149 | 13.866 ± 0.36 | 5629 ± 240 | 52911 ± 1786 |
| 0.1 (2k) | 0.598 ± 0.017 | 35.929 ± 1.397 | 0.698 ± 0.009 | 0.051 ± 0.003 | 0.339 ± 0.012 | 8.968 ± 0.083 | 14.013 ± 0.158 | 6173 ± 360 | 55918 ± 2641 |
| 0.5 (10k) | 0.593 ± 0.026 | 35.95 ± 2.177 | 0.666 ± 0.073 | 0.049 ± 0.003 | 0.314 ± 0.046 | 8.635 ± 0.634 | 13.432 ± 1.173 | 5882 ± 356 | 51403 ± 9297 |
| 1.0 (20k) | 0.605 ± 0.023 | 33.497 ± 0.447 | 0.666 ± 0.013 | 0.043 ± 0.002 | 0.287 ± 0.008 | 8.575 ± 0.073 | 13.484 ± 0.244 | 5230 ± 363 | 46483 ± 1318 |
| NLPO | | | | | | | | | |
| 0.0 (0k) | 0.487 ± 0.01 | 32.572 ± 0.165 | 0.685 ± 0.003 | 0.043 ± 0.001 | 0.299 ± 0.003 | 8.691 ± 0.023 | 13.787 ± 0.034 | 5126 ± 177 | 48475 ± 491 |
| 0.1 (2k) | 0.599 ± 0.007 | 33.536 ± 0.378 | 0.67 ± 0.01 | 0.043 ± 0.001 | 0.289 ± 0.009 | 8.608 ± 0.061 | 13.576 ± 0.192 | 5125 ± 220 | 46755 ± 1449 |
| 0.5 (10k) | 0.617 ± 0.021 | 33.409 ± 0.354 | 0.668 ± 0.005 | 0.041 ± 0.001 | 0.281 ± 0.006 | 8.552 ± 0.044 | 13.533 ± 0.091 | 4926 ± 183 | 45256 ± 1022 |
| 1.0 (20k) | 0.637 ± 0.013 | 32.667 ± 0.631 | 0.677 ± 0.014 | 0.044 ± 0.002 | 0.288 ± 0.010 | 8.588 ± 0.100 | 13.484 ± 0.236 | 5205 ± 189 | 46344 ± 2688 |

Table 7: **Evaluation of GPT2 with different algorithms on IMDB sentiment text continuation task, data budget ablations**: Mean and standard deviations over 5 random seeds is reported for sentiment scores along with fluency and diversity metrics. This table measures performance differences as a function of the fraction of the dataset that has been used. In the case of the RL approaches, this measures how much data is used to train the reward classifier, and for the supervised method it directly measures fraction of positive reviews used for training. We note that using even a small fraction of data to train a reward classifier proves to be effective in terms of downstream task performance while this is not true for supervised approaches. This lends evidence to the hypothesis that adding expending data budget on a reward classifier is more effective than adding more gold label expert demonstrations.

| Hyperparams | Semantic and Fluency Metrics | | Diversity Metrics | | | | | | |
|---|---|---|---|---|---|---|---|---|---|
| | Sentiment Score ↑ | Perplexity ↓ | MSTTR | Distinct$_1$ | Distinct$_2$ | H$_1$ | H$_2$ | Unique$_1$ | Unique$_2$ |
| Target Update Iterations $\mu$ | | | | | | | | | |
| 1 | $0.594 \pm 0.018$ | $32.671 \pm 0.201$ | $0.669 \pm 0.008$ | $0.042 \pm 0.002$ | $0.284 \pm 0.007$ | $8.575 \pm 0.064$ | $13.503 \pm 0.181$ | $4986 \pm 265$ | $45916 \pm 1168$ |
| 10 | $0.622 \pm 0.014$ | $32.729 \pm 0.567$ | $0.659 \pm 0.019$ | $0.042 \pm 0.002$ | $0.274 \pm 0.007$ | $8.489 \pm 0.106$ | $13.31 \pm 0.272$ | $5138 \pm 385$ | $43989 \pm 1120$ |
| 20 | $0.637 \pm 0.013$ | $32.667 \pm 0.631$ | $0.677 \pm 0.014$ | $0.044 \pm 0.002$ | $0.288 \pm 0.010$ | $8.588 \pm 0.100$ | $13.484 \pm 0.236$ | $5205 \pm 189$ | $46344 \pm 2688$ |
| 50 | $0.603 \pm 0.015$ | $33.397 \pm 0.325$ | $0.67 \pm 0.006$ | $0.043 \pm 0.001$ | $0.287 \pm 0.004$ | $8.605 \pm 0.041$ | $13.54 \pm 0.116$ | $5228 \pm 113$ | $46418 \pm 685$ |
| Top-p mask | | | | | | | | | |
| 0.1 | $0.579 \pm 0.021$ | $32.451 \pm 0.243$ | $0.67 \pm 0.008$ | $0.042 \pm 0.001$ | $0.283 \pm 0.01$ | $8.569 \pm 0.084$ | $13.515 \pm 0.195$ | $5018 \pm 47$ | $45760 \pm 1579$ |
| 0.3 | $0.588 \pm 0.019$ | $32.451 \pm 0.303$ | $0.666 \pm 0.007$ | $0.043 \pm 0.001$ | $0.285 \pm 0.004$ | $8.568 \pm 0.032$ | $13.482 \pm 0.172$ | $5201 \pm 247$ | $46357 \pm 539$ |
| 0.5 | $0.588 \pm 0.01$ | $32.447 \pm 0.393$ | $0.669 \pm 0.001$ | $0.044 \pm 0.003$ | $0.291 \pm 0.008$ | $8.614 \pm 0.053$ | $13.535 \pm 0.06$ | $5305 \pm 384$ | $47251 \pm 1226$ |
| 0.7 | $0.619 \pm 0.013$ | $32.373 \pm 0.329$ | $0.663 \pm 0.008$ | $0.043 \pm 0.001$ | $0.28 \pm 0.006$ | $8.533 \pm 0.043$ | $13.366 \pm 0.129$ | $5186 \pm 216$ | $45149 \pm 1452$ |
| 0.9 | $0.637 \pm 0.013$ | $32.667 \pm 0.631$ | $0.677 \pm 0.014$ | $0.044 \pm 0.002$ | $0.288 \pm 0.010$ | $8.588 \pm 0.100$ | $13.484 \pm 0.236$ | $5205 \pm 189$ | $46344 \pm 2688$ |

Table 8: **Evaluation of GPT2 with different algorithms on IMDB sentiment text continuation task, NLPO hyperparameter ablations**: Mean and standard deviations over 5 random seeds is reported for sentiment scores along with fluency and diversity metrics. This table shows results of NLPO's stability to the unique hyperparameters introduced in the algorithm - all other parameters held constant from the best PPO model. The number of iterations after which the masking model syncs with the policy and the top-p nucleus percentage for the mask model itself. We see that in general, the higher the top-p mask percentage, the better the performance. For target update iterations, performance is low if the mask model is not updated often enough or if it updated too often.

| Algorithm | Unique N | Coherence | | | Sentiment | | |
|---|---|---|---|---|---|---|---|
| | | Value | Alpha | Skew | Value | Alpha | Skew |
| NLPO with KL | 27 | 3.49 | 0.196 | 3.497 | 3.61 | 0.2 | 3.601 |
| NLPO without KL | 29 | 3.16 | 0.21 | 3.158 | 4.41 | 0.158 | 4.403 |
| PPO without KL | 27 | 3.16 | 0.17 | 3.163 | 4.36 | 0.196 | 4.363 |
| PPO with KL | 29 | 3.46 | 0.124 | 3.462 | 3.58 | 0.116 | 3.575 |
| Zero Shot | 28 | 3.6 | 0.162 | 3.591 | 3.1 | 0.13 | 3.097 |
| Supervised | 29 | 3.51 | 0.192 | 3.512 | 3.43 | 0.2 | 3.428 |
| Human | 27 | 4.13 | 0.159 | 4.128 | 3.01 | 0.31 | 3.017 |
| Supervised+PPO | 22 | 3.45 | 0.211 | 3.147 | 3.64 | 0.21 | 3.161 |
| Supervised+NLPO | 22 | 3.48 | 0.181 | 3.226 | 3.73 | 0.22 | 3.047 |

Table 9: Results of the human subject study showing the number of participants N, average Likert scale value for coherence and sentiment, Krippendorf's alpha showing inter-annotator agreement, and Skew. For each model a total of 100 samples were drawn randomly from the test set and rated by 3 annotators each, resulting in 300 data points per algorithm.

### B.3.3 HUMAN PARTICIPANT STUDY

Figure 5 shows the IMDB instructions, example, and interface used both for the qualification round, and then later, for the human evaluation experiments. Tables 9, 10 show averaged results, annotator agreement, and the results of statistical significance tests to determine which models output better generations when rated by humans.

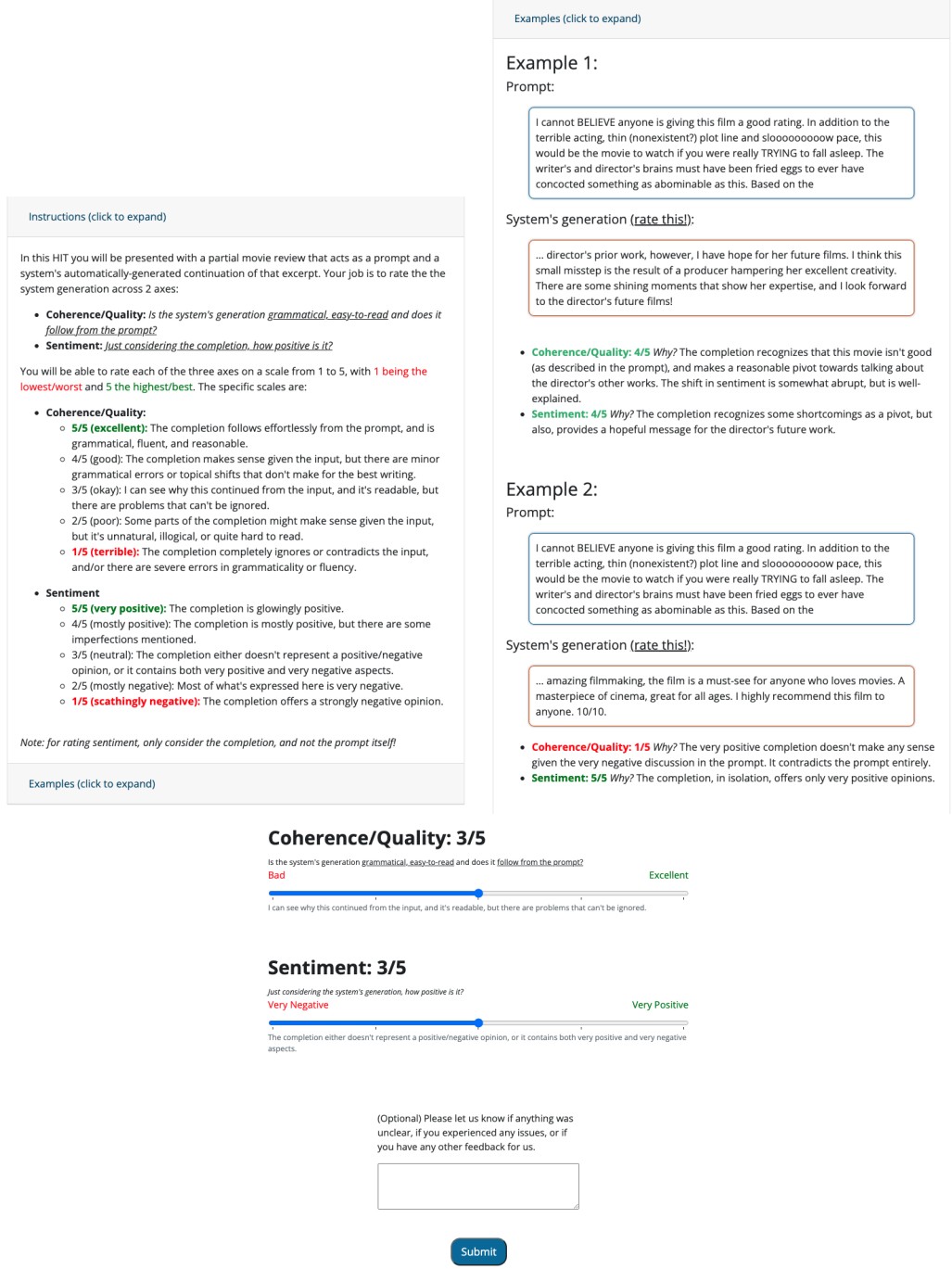

Figure 5: Instructions, example, and interface for the IMDB sentiment completion task.

| Group 1 | Group 2 | Coherence | | Sentiment | |
|---|---|---|---|---|---|
| | | Diff (G2-G1) | *p-values* | Diff (G2-G1) | *p-values* |
| PPO with KL | PPO without KL | **-0.3** | **0.035** | **0.783** | **0.001** |
| PPO with KL | NLPO with KL | 0.03 | 0.9 | 0.027 | 0.9 |
| PPO with KL | NLPO without KL | **-0.3** | **0.035** | **0.827** | **0.001** |
| PPO with KL | Supervised | 0.05 | 0.9 | -0.15 | 0.591 |
| PPO with KL | Human | **0.667** | **0.001** | **-0.567** | **0.001** |
| PPO with KL | Zero Shot | 0.137 | 0.776 | **-0.483** | **0.001** |
| PPO without KL | NLPO with KL | **0.33** | **0.013** | **-0.757** | **0.001** |
| PPO without KL | NLPO without KL | 0.001 | 0.9 | 0.043 | 0.9 |
| PPO without KL | Supervised | **0.35** | **0.006** | **-0.933** | **0.001** |
| PPO without KL | Human | **0.967** | **0.009** | **-1.35** | **0.001** |
| PPO without KL | Zero Shot | **0.437** | **0.001** | **-1.267** | **0.001** |
| NLPO with KL | NLPO without KL | **-0.33** | **0.013** | **0.8** | **0.001** |
| NLPO with KL | Supervised | 0.02 | 0.9 | -0.177 | 0.404 |
| NLPO with KL | Human | **0.637** | **0.001** | **-0.593** | **0.001** |
| NLPO with KL | Zero Shot | 0.107 | 0.9 | **-0.51** | **0.001** |
| NLPO without KL | Supervised | **0.35** | **0.006** | **-0.977** | **0.001** |
| NLPO without KL | Human | **0.967** | **0.001** | **-1.393** | **0.001** |
| NLPO without KL | Zero Shot | **0.437** | **0.001** | **-1.31** | **0.001** |
| Supervised | Human | **0.617** | **0.001** | **-0.417** | **0.001** |
| Supervised | Zero Shot | 0.087 | 0.9 | **-0.333** | **0.0027** |
| Human | Zero Shot | **-0.53** | **0.001** | 0.083 | 0.9 |
| Supervised+PPO | Supervised+NLPO | 0.03 | 0.9 | **0.09** | **0.035** |
| Supervised+PPO | NLPO with KL | 0.04 | 0.9 | -0.03 | 0.9 |
| Supervised+PPO | NLPO without KL | **-0.29** | **0.001** | **0.77** | **0.001** |
| Supervised+PPO | PPO without KL | **-0.29** | **0.006** | **0.72** | **0.001** |
| Supervised+PPO | PPO with KL | 0.01 | 0.9 | **-0.06** | **0.001** |
| Supervised+PPO | Zero Shot | **0.15** | **0.035** | **-0.54** | **0.001** |
| Supervised+PPO | Supervised | **0.06** | **0.001** | **-0.21** | **0.001** |
| Supervised+PPO | Human | **0.68** | **0.001** | **-0.63** | **0.001** |
| Supervised+NLPO | NLPO with KL | 0.01 | 0.9 | **-0.12** | **0.001** |
| Supervised+NLPO | NLPO without KL | **-0.32** | **0.001** | **0.68** | **0.001** |
| Supervised+NLPO | PPO without KL | **-0.32** | **0.035** | **0.63** | **0.001** |
| Supervised+NLPO | PPO with KL | -0.02 | 0.9 | **-0.15** | **0.006** |
| Supervised+NLPO | Zero Shot | **-0.12** | **0.001** | **-0.63** | **0.001** |
| Supervised+NLPO | Supervised | 0.03 | 0.9 | **-0.3** | **0.001** |
| Supervised+NLPO | Human | **0.65** | **0.001** | **-0.72** | **0.006** |

Table 10: Results of an post-hoc Tukey HSD Test for difference in means between pairs of algorithms (Group 2 - Group 1) and corresponding $p$-values. Individually statistically significant results are bolded and are used to discuss results in the analysis. Overall $p$-values showing that there is a significant difference in means between the models via a one-way ANOVA test are significant with $p \ll 0.05$ for both coherence and sentiment.

### B.3.4 QUALITATIVE RESULTS

We show sample generations from each of the algorithms for three randomly picked prompts below.

**Sample** 1
**Prompt:** A man and his wife are not getting along because of the wife's jealousy and drinking problems. When the wife disappears, her sister begins a vigorous search involving the
**Zero-Shot:** husband's sister, which leads several men to a remote cabin and then a boat. The wife and her husband then start to see the same things. Finally, a strange woman is driving out to her cabin and discovers the husband's disappearance but
**References:** local police, state government, and a television crime show. What she finds out stuns the entire community. Good drama with lots of fine performances; based on a true story
**PPO:** police and the local public. At the end, the woman is killed and the wife is found naked and with a red head tattooed on her back. This is part of a special feature called"On the Road, Hidden" which was
**PPO-no-KL:** police and other local governments. Written by G. Scott Fitzgerald, who also stars in this production, "A Very Merry Christmas" is a wonderful family movie with a wonderful soundtrack and wonderful characters. It brings home the true beauty of small-
**NLPO:** mother while the brother goes under. After all, his sister, who was apparently kidnapped but who turns out to be a man, is no longer in her place. If a film that was made in the 1950s
**NLPO-no-KL:** friends of the wife, and also the friend of the wife of the woman that is missing, and eventually he begins to find that she has gone to Germany and he has made a very good friend. There is a good scene between John and
**Supervised:** amily tree. The widower, now widowed, returns, the daughter, daughter is not pleased when his wife is mistaken for one of her sons, and the patriarch, then a wealthy man, falls victim to a jealous husband. The

**Sample** 2
**Prompt:** A very credible and unsettling movie portraying the effects of war on the general public, Shame gives us a chilling understanding oh Ingmar Bergman's beliefs. The absence of color and sound (as in a soundtrack) helps to give the film a more realistic feeling. There is no soft or hard lighting or a variety of camera angles to give the story a charming or dramatic effect. Everything is very simply portrayed as it would be in reality. At times the storyline was confusing, but perhaps in order to give a sense of the characters' frame of mind how they too do now know what is happening or why. The simplicity of the
**Zero-Shot:** camera work is at once apparent and fascinating in the manner in which it affects the way things are portrayed. In other films these two have been referred to from time to time in a form and manner more suited to the screen. The film shows
**References:** film made it somewhat boring, but I could understand how the director is trying not to glamorize the story and points. He depicted the harsh and unsettling events and effects of war. In the beginning the characters were introduced and shown as content with the lighter settings
**PPO:** characters relationships is so engrossing and the great and unique story presented by each pair of brothers and sisters creates a truly gripping and enjoyable experience. To put it simply, I strongly recommend to everyone who plays the role to buy this DVD!!!
**PPO-no-KL:** movie helps it to flow along in a believable way. Rating 4 out of 5 Great Movie! I highly recommend these movies! I highly recommend these movies! Rating 5 out of 5 Great Movie! I highly recommend these movies! Rating 10 out of"
**NLPO:** novel makes it one of the first books I have read by a British actor. It is a fun way for someone who likes their books, like me, to sit through a novel. If you like to read novels, check it out.
**NLPO-no-KL:** setting makes the final product very enjoyable to watch. Highly recommended!! ~ Michael Bowerman, SPSA Film Festival Host, SPSA, WA (December 17, 2006), 8/17/2006 Great photography, good acting,
**Supervised:** "Dakota Clan" and his sister, the two become friends with Ted Masters who soon finds the chance to fly back to their home land and go back to save his brother's life and his family. The

**Sample** 3
**Prompt:** Shakespearean language is wild and rambling, saturated in multiple meanings, word play and metaphor. To be understood it must be wrangled and tamed by an actor with the strength and knowledge to do so. When an actor fails, the words pour forth in a torrent of incomprehensible words, but when he succeeds, the English language springs to life with an immediacy and vibrancy that takes your breath away. Al Pacino is one such actor, and here displays an incredible level of clarity and control that, were there any justice,

**Zero-Shot:** might still be seen in the world today. This movie, like "Vacation", is an effort at telling the tale of two families from different races. One is African but never grew up the other. One of them, a man
**References:** would sweep every award in the offering. He meets the challenge of presenting Shylock head on, and delivers an extraordinarily subtle and nuanced performance. It would be a crime if we never got the opportunity to see what he does with King Lear.

The supporting cast
**PPO:** would have made him an instant-classic. This movie captures the spirit of childhood and has everyone in the world watching it. It's a love story and it's really heartbreaking. And at the same time, it's a classic. It

**PPO**-no-**KL:** will bring victory to the fight against hatred and extremism. Al Pacino is a very beautiful, well executed actor who brings an absolute intensity and sensibility to his portrayal of love and loyalty. A must see! Recommended! Also, an Excellent

**NLPO:** I would watch and enjoy. 7/10, Oscar Wilde is always a pleasure to see. A must see. Highly recommend. Highly recommend! One of my top ten films of all time. A must see!!

**NLPO**-no-**KL:** the whole film would have fallen to the same fate, just as it did just a couple hundred years ago. Don't miss it. It's a real classic. Highly Recommended. * outta five stars for it!

**Supervised:** his performance (so far) would seem mere shadow. He is truly in the middle of a movie, and this film is one of those films where he can be convincing in it (and his trademark acting, as you can see in the

## B.4 COMMONGEN

### B.4.1 SETUP

CommonGen (Lin et al., 2020) deals with task of generating coherent sentences describing an input set of concepts (eg. "a man is throwing a frisbee"). For training RL methods, we consider 3 traditional lexical rewards namely Rouge-1, Rouge-avg (which is an average of Rouge-1, 2 and L) and meteor. Additionally, we also train with task-specific rewards such as CIDEr (Vedantam et al., 2015), SPICE (Anderson et al., 2016) and SPiDer (Liu et al., 2017) which is a just a linear combination of both with equal weights. We chose T5-base as the base LM since it is well-suited for structure to text tasks. We additionally note that concept set inputs are prefixed with "generate a sentence with:" to encourage exploration.

During our initial experiments when fine-tuning directly on LM, we observed that policy learns to repeat the prompted concepts in order to maximize rewards resulting in a well-known problem of *reward hacking*. To mitigate this, we add a penalty score of $-1$ to final task reward if the n-grams of prompt text overlaps with generated text. In contrast, when initialized with a supervised policy, this problem is not seen and hence penalty score is not applied. We use beam search as the decoding method during evaluation whereas for rollouts, we use top k sampling to favor exploration over exploitation. Table 11 provides an in-depth summary of setting of hyperparameter values along with other implementation details.

| Model Params | value |
|---|---|
| supervised | batch size: 8
epochs: 4
learning rate: 0.00001
learning rate scheduler: cosine
weight decay: 0.01 |
| ppo/ nlpo | steps per update: 1280
total number of steps: 256000
batch size: 64
epochs per update: 5
learning rate: 0.000002
entropy coefficient: 0.01
initial kl coeff: 0.001
target kl: 2.0
discount factor: 0.99
gae lambda: 0.95
clip ratio: 0.2
value function coeff: 0.5
top mask ratio: 0.9
target update iterations: 20 |
| supervised+ ppo (or nlpo) | steps per update: 1280
total number of steps: 128000
batch size: 64
epochs per update: 5
learning rate: 0.000002
entropy coefficient: 0.01
initial kl coeff: 0.01
target kl: 1.0
discount factor: 0.99
gae lambda: 0.95
clip ratio: 0.2
value function coeff: 0.5
top mask ratio: 0.9
target update iterations: 20 |
| decoding | num beams: 5
min length: 5
max new tokens: 20 |
| tokenizer | padding side: left
max length: 20 |

Table 11: **CommonGen Hyperparams**: Table shows a list of all hyper-parameters and their settings

### B.4.2 RESULTS AND DISCUSSION

Tables 13, 12 presents our benchmarking results with 6 reward functions along with supervised baseline performances on dev and test sets respectively. Our main finding is that warm-started initial policies are crucial for learning to generate coherent sentences with common sense. Without warm-start, policies suffer from reward hacking despite application of repetition penalty and task-specific metrics such as CIDer etc. Further, we find that RL fine-tuned models obtain very high concept coverage which is also seen in Table B.4.5. Supervised models often tend to miss few concepts in its generation compared to RL methods.

| Tasks | Alg | LM | Reward function | Lexical and Semantic Metrics | | | | | | | |
| | | | | Rouge-2 | Rouge-L | Bleu (n=3) | Bleu (n=4) | Meteor | CIDEr | SPICE | Coverage |
| --- | --- | --- | --- | --- | --- | --- | --- | --- | --- | --- | --- |
| | Zero-Shot | T5 | | 0.016 | 0.264 | 0.029 | 0.006 | 0.203 | 6.200 | 0.115 | 91.070 |
| | PPO | T5 | Rouge-1 | $0.085 \pm 0.008$ | $0.354 \pm 0.004$ | $0.161 \pm 0.011$ | $0.087 \pm 0.009$ | $0.235 \pm 0.002$ | $8.673 \pm 0.234$ | $0.157 \pm 0.001$ | $88.544 \pm 2.36$ |
| | | T5 | Rouge-Avg | $0.093 \pm 0.005$ | $0.351 \pm 0.001$ | $0.169 \pm 0.032$ | $0.097 \pm 0.017$ | $0.224 \pm 0.012$ | $8.212 \pm 1.329$ | $0.159 \pm 0.011$ | $82.584 \pm 2.569$ |
| | | T5 | Meteor | $0.091 \pm 0.008$ | $0.308 \pm 0.007$ | $0.166 \pm 0.016$ | $0.088 \pm 0.013$ | $0.220 \pm 0.006$ | $7.251 \pm 0.453$ | $0.161 \pm 0.007$ | $79.718 \pm 2.267$ |
| | | T5 | SPice | $0.065 \pm 0.003$ | $0.302 \pm 0.002$ | $0.115 \pm 0.063$ | $0.067 \pm 0.041$ | $0.193 \pm 0.014$ | $6.571 \pm 1.312$ | $0.175 \pm 0.011$ | $69.340 \pm 3.617$ |
| | | T5 | CiDer | $0.066 \pm 0.003$ | $0.304 \pm 0.002$ | $0.132 \pm 0.057$ | $0.074 \pm 0.036$ | $0.211 \pm 0.009$ | $6.877 \pm 1.218$ | $0.143 \pm 0.017$ | $80.114 \pm 4.852$ |
| | | T5 | SPider | $0.117 \pm 0.005$ | $0.352 \pm 0.007$ | $0.224 \pm 0.014$ | $0.137 \pm 0.011$ | $0.226 \pm 0.01$ | $9.162 \pm 0.539$ | $0.186 \pm 0.006$ | $73.374 \pm 6.073$ |
| | NLPO | T5 | Rouge-1 | $0.087 \pm 0.002$ | $0.339 \pm 0.009$ | $0.127 \pm 0.048$ | $0.069 \pm 0.035$ | $0.213 \pm 0.002$ | $6.962 \pm 0.883$ | $0.145 \pm 0.022$ | $80.89 \pm 9.544$ |
| | | T5 | Rouge-Avg | $0.095 \pm 0.001$ | $0.338 \pm 0.002$ | $0.159 \pm 0.02$ | $0.093 \pm 0.013$ | $0.216 \pm 0.009$ | $7.55 \pm 0.688$ | $0.153 \pm 0.008$ | $77.944 \pm 2.770$ |
| CommonGen | | T5 | Meteor | $0.110 \pm 0.005$ | $0.332 \pm 0.003$ | $0.214 \pm 0.007$ | $0.124 \pm 0.007$ | $0.235 \pm 0.004$ | $8.669 \pm 0.164$ | $0.173 \pm 0.002$ | $82.007 \pm 1.012$ |
| | | T5 | SPice | $0.014 \pm 0.006$ | $0.242 \pm 0.001$ | $0.037 \pm 0.011$ | $0.018 \pm 0.007$ | $0.156 \pm 0.007$ | $4.685 \pm 0.283$ | $0.168 \pm 0.008$ | $56.998 \pm 3.548$ |
| | | T5 | CiDer | $0.046 \pm 0.001$ | $0.241 \pm 0.003$ | $0.078 \pm 0.028$ | $0.043 \pm 0.016$ | $0.151 \pm 0.022$ | $3.964 \pm 0.792$ | $0.103 \pm 0.012$ | $49.606 \pm 7.971$ |
| | | T5 | SPider | $0.060 \pm 0.006$ | $0.258 \pm 0.001$ | $0.090 \pm 0.008$ | $0.056 \pm 0.005$ | $0.151 \pm 0.022$ | $4.411 \pm 0.837$ | $0.123 \pm 0.022$ | $49.230 \pm 10.468$ |
| | Supervised | T5 | | $0.215 \pm 0.001$ | $0.438 \pm 0.001$ | $0.444 \pm 0.001$ | $0.329 \pm 0.001$ | $0.321 \pm 0.001$ | $16.385 \pm 0.046$ | $\mathbf{0.299} \pm 0.001$ | $94.476 \pm 0.172$ |
| | Supervised + PPO | T5 | Rouge-1 | $0.232 \pm 0.002$ | $\mathbf{0.453} \pm 0.002$ | $0.454 \pm 0.006$ | $\mathbf{0.338} \pm 0.006$ | $0.320 \pm 0.002$ | $16.233 \pm 0.159$ | $0.288 \pm 0.004$ | $96.412 \pm 0.424$ |
| | | T5 | Rouge-Avg | $0.230 \pm 0.001$ | $0.450 \pm 0.001$ | $0.448 \pm 0.005$ | $0.334 \pm 0.005$ | $0.319 \pm 0.001$ | $16.069 \pm 0.167$ | $0.287 \pm 0.003$ | $96.116 \pm 0.679$ |
| | | T5 | Meteor | $\mathbf{0.234} \pm 0.002$ | $0.450 \pm 0.003$ | $\mathbf{0.462} \pm 0.007$ | $0.342 \pm 0.007$ | $\mathbf{0.327} \pm 0.001$ | $\mathbf{16.797} \pm 0.152$ | $0.295 \pm 0.001$ | $\mathbf{97.690} \pm 0.371$ |
| | | T5 | SPice | $0.227 \pm 0.004$ | $0.447 \pm 0.003$ | $0.450 \pm 0.007$ | $0.336 \pm 0.008$ | $0.319 \pm 0.002$ | $16.208 \pm 0.249$ | $0.288 \pm 0.003$ | $96.492 \pm 0.29$ |
| | | T5 | CiDer | $0.224 \pm 0.003$ | $0.446 \pm 0.003$ | $0.427 \pm 0.012$ | $0.309 \pm 0.01$ | $0.316 \pm 0.004$ | $15.497 \pm 0.428$ | $0.283 \pm 0.004$ | $96.344 \pm 0.547$ |
| | | T5 | SPider | $0.226 \pm 0.003$ | $0.448 \pm 0.002$ | $0.436 \pm 0.005$ | $0.319 \pm 0.004$ | $0.317 \pm 0.003$ | $15.678 \pm 0.192$ | $0.281 \pm 0.003$ | $96.154 \pm 0.426$ |
| | Supervised + NLPO | T5 | Rouge-1 | $0.229 \pm 0.002$ | $0.450 \pm 0.001$ | $0.454 \pm 0.005$ | $0.338 \pm 0.004$ | $0.320 \pm 0.003$ | $16.206 \pm 0.175$ | $0.289 \pm 0.002$ | $96.342 \pm 0.572$ |
| | | T5 | Rouge-Avg | $0.232 \pm 0.003$ | $0.451 \pm 0.002$ | $0.458 \pm 0.01$ | $0.342 \pm 0.009$ | $0.321 \pm 0.003$ | $16.351 \pm 0.335$ | $0.290 \pm 0.005$ | $95.998 \pm 0.496$ |
| | | T5 | Meteor | $0.231 \pm 0.003$ | $0.449 \pm 0.002$ | $0.454 \pm 0.007$ | $0.334 \pm 0.008$ | $0.326 \pm 0.002$ | $16.574 \pm 0.269$ | $0.292 \pm 0.003$ | $97.374 \pm 0.457$ |
| | | T5 | SPice | $0.223 \pm 0.002$ | $0.442 \pm 0.001$ | $0.435 \pm 0.011$ | $0.321 \pm 0.010$ | $0.315 \pm 0.004$ | $15.747 \pm 0.401$ | $0.283 \pm 0.005$ | $96.25 \pm 0.313$ |
| | | T5 | CiDer | $0.226 \pm 0.002$ | $0.447 \pm 0.004$ | $0.433 \pm 0.007$ | $0.315 \pm 0.008$ | $0.318 \pm 0.003$ | $15.741 \pm 0.170$ | $0.285 \pm 0.001$ | $96.354 \pm 0.971$ |
| | | T5 | SPider | $0.226 \pm 0.004$ | $0.447 \pm 0.003$ | $0.434 \pm 0.006$ | $0.316 \pm 0.006$ | $0.319 \pm 0.002$ | $15.739 \pm 0.311$ | $0.284 \pm 0.003$ | $96.333 \pm 0.644$ |

Table 12: **CommonGen test evaluation** Table shows official scores obtained from CommonGen hold-out evaluation. The most important result is that RL fine-tuning on a supervised model yields better performance across most metrics especially Coverage which indicates the ratio of concepts covered in generated texts

| Tasks | Alg | Reward Function | Top k | LM | Rouge-1 | Rouge-2 | Rouge-L | Rouge-LSum | Meteor | BLEU | BertScore | Cider | Spice | MSTTR | Distinct$_1$ | Distinct$_2$ | H$_1$ | H$_2$ | Unique$_1$ | Unique$_2$ | Mean Output Length |
|---|---|---|---|---|---|---|---|---|---|---|---|---|---|---|---|---|---|---|---|---|---|
| | | | | | | | | | **Lexical and Semantic Metrics** | | | | | | | | **Diversity Metrics** | | | | |
| | Zero-Shot | | | T5 | 0.415 | 0.016 | 0.270 | 0.270 | 0.179 | 0.0 | 0.854 | 0.640 | 0.231 | 0.430 | 0.090 | 0.335 | 5.998 | 7.957 | 345 | 1964 | 8.797 |
| | PPO | Rouge-1 | 50 | T5 | 0.537 ± 0.004 | 0.093 ± 0.012 | 0.380 ± 0.006 | 0.380 ± 0.006 | 0.235 ± 0.005 | 0.016 ± 0.002 | 0.896 ± 0.001 | 0.950 ± 0.015 | 0.318 ± 0.016 | 0.526 ± 0.020 | 0.128 ± 0.005 | 0.518 ± 0.036 | 6.679 ± 0.132 | 10.572 ± 0.234 | 437.4 ± 42.017 | 2418.8 ± 167.947 | 7.214 ± 0.374 |
| | | Rouge-Avg | 50 | T5 | 0.519 ± 0.0185 | 0.102 ± 0.007 | 0.377 ± 0.013 | 0.376 ± 0.014 | 0.225 ± 0.024 | 0.020 ± 0.002 | 0.897 ± 0.005 | 0.921 ± 0.102 | 0.328 ± 0.009 | 0.536 ± 0.069 | 0.141 ± 0.022 | 0.510 ± 0.056 | 6.777 ± 0.539 | 10.348 ± 0.134 | 458.6 ± 19.734 | 2244.4 ± 162.855 | 6.887 ± 1.006 |
| | | Meteor | 50 | T5 | 0.411 ± 0.009 | 0.090 ± 0.008 | 0.304 ± 0.006 | 0.304 ± 0.006 | 0.210 ± 0.005 | 0.029 ± 0.004 | 0.875 ± 0.007 | 0.638 ± 0.048 | 0.259 ± 0.017 | 0.547 ± 0.012 | 0.147 ± 0.003 | 0.529 ± 0.014 | 7.62 ± 0.127 | 11.464 ± 0.151 | 1039.4 ± 63.276 | 5197.2 ± 280.004 | 13.660 ± 0.324 |
| | | SPice | 50 | T5 | 0.439 ± 0.035 | 0.079 ± 0.045 | 0.323 ± 0.036 | 0.323 ± 0.036 | 0.183 ± 0.022 | 0.012 ± 0.009 | 0.891 ± 0.005 | 0.777 ± 0.140 | 0.400 ± 0.012 | 0.546 ± 0.054 | 0.149 ± 0.019 | 0.545 ± 0.072 | 6.721 ± 0.441 | 10.492 ± 0.330 | 409.2 ± 41.605 | 1878.4 ± 167.492 | 5.706 ± 0.678 |
| | | CiDer | 50 | T5 | 0.453 ± 0.038 | 0.081 ± 0.037 | 0.326 ± 0.033 | 0.326 ± 0.033 | 0.203 ± 0.022 | 0.017 ± 0.009 | 0.885 ± 0.008 | 0.770 ± 0.134 | 0.291 ± 0.036 | 0.597 ± 0.081 | 0.195 ± 0.040 | 0.639 ± 0.106 | 7.732 ± 0.682 | 11.131 ± 0.502 | 777.0 ± 144.676 | 3350.8 ± 503.419 | 7.393 ± 0.572 |
| | | SPider | 50 | T5 | 0.512 ± 0.008 | 0.141 ± 0.007 | 0.388 ± 0.002 | 0.388 ± 0.003 | 0.242 ± 0.007 | 0.032 ± 0.003 | 0.902 ± 0.001 | 1.045 ± 0.034 | 0.380 ± 0.006 | 0.482 ± 0.015 | 0.133 ± 0.003 | 0.472 ± 0.021 | 6.372 ± 0.221 | 10.303 ± 0.228 | 502.6 ± 33.422 | 2281.4 ± 252.471 | 7.489 ± 0.358 |
| | NLPO | Rouge-1 | 50 | T5 | 0.499 ± 0.012 | 0.089 ± 0.003 | 0.328 ± 0.007 | 0.328 ± 0.007 | 0.198 ± 0.002 | 0.021 ± 0.001 | 0.872 ± 0.005 | 0.815 ± 0.009 | 0.305 ± 0.008 | 0.559 ± 0.01 | 0.148 ± 0.003 | 0.555 ± 0.012 | 7.059 ± 0.067 | 10.657 ± 0.105 | 457.9 ± 11.108 | 2349.6 ± 60.345 | 6.586 ± 0.094 |
| | | Rouge-Avg | 50 | T5 | 0.47 ± 0.01 | 0.096 ± 0.004 | 0.312 ± 0.006 | 0.312 ± 0.006 | 0.202 ± 0.008 | 0.025 ± 0.002 | 0.843 ± 0.013 | 0.816 ± 0.026 | 0.299 ± 0.007 | 0.512 ± 0.019 | 0.146 ± 0.011 | 0.513 ± 0.012 | 6.781 ± 0.15 | 10.424 ± 0.156 | 484.18 ± 17.303 | 2357.54 ± 152.113 | 7.131 ± 0.487 |
| | | Meteor | 50 | T5 | 0.389 ± 0.013 | 0.1 ± 0.004 | 0.293 ± 0.008 | 0.293 ± 0.008 | 0.226 ± 0.024 | 0.035 ± 0.004 | 0.832 ± 0.018 | 0.691 ± 0.04 | 0.266 ± 0.016 | 0.503 ± 0.003 | 0.132 ± 0.005 | 0.471 ± 0.008 | 7.146 ± 0.192 | 10.727 ± 0.313 | 648.05 ± 33.963 | 3536.0 ± 444.638 | 11.062 ± 1.301 |
| CommonGen | | SPice | 50 | T5 | 0.329 ± 0.015 | 0.036 ± 0.008 | 0.247 ± 0.013 | 0.247 ± 0.013 | 0.137 ± 0.009 | 0.006 ± 0.002 | 0.817 ± 0.024 | 0.515 ± 0.033 | 0.323 ± 0.021 | 0.543 ± 0.023 | 0.174 ± 0.004 | 0.568 ± 0.026 | 7.176 ± 0.212 | 10.551 ± 0.216 | 479.45 ± 19.77 | 2065.8 ± 288.843 | 5.785 ± 0.431 |
| | | CiDer | 50 | T5 | 0.515 ± 0.006 | 0.143 ± 0.008 | 0.387 ± 0.006 | 0.308 ± 0.006 | 0.19 ± 0.001 | 0.019 ± 0.001 | 0.865 ± 0.015 | 0.726 ± 0.018 | 0.282 ± 0.009 | 0.55 ± 0.02 | 0.179 ± 0.005 | 0.576 ± 0.014 | 7.286 ± 0.125 | 10.812 ± 0.089 | 661.46 ± 21.776 | 2726.32 ± 71.253 | 7.13 ± 0.223 |
| | | SPider | 50 | T5 | 0.393 ± 0.008 | 0.086 ± 0.012 | 0.297 ± 0.007 | 0.297 ± 0.007 | 0.183 ± 0.007 | 0.02 ± 0.003 | 0.842 ± 0.019 | 0.717 ± 0.026 | 0.297 ± 0.019 | 0.525 ± 0.024 | 0.167 ± 0.009 | 0.537 ± 0.025 | 6.986 ± 0.262 | 10.451 ± 0.171 | 530.14 ± 16.805 | 2263.4 ± 166.221 | 6.687 ± 0.372 |
| | Supervised | | | T5 | 0.503 ± 0.001 | 0.175 ± 0.001 | 0.411 ± 0.001 | 0.411 ± 0.001 | 0.309 ± 0.001 | 0.069 ± 0.001 | 0.929 ± 0.000 | 1.381 ± 0.011 | 0.443 ± 0.001 | 0.509 ± 0.001 | 0.101 ± 0.001 | 0.339 ± 0.001 | 6.531 ± 0.006 | 10.079 ± 0.016 | 503.600 ± 6.530 | 2158.8 ± 24.514 | 10.934 ± 0.020 |
| | Supervised + PPO | Rouge-1 | 50 | T5 | 0.537 ± 0.004 | 0.198 ± 0.005 | 0.433 ± 0.002 | 0.433 ± 0.002 | 0.314 ± 0.003 | 0.070 ± 0.002 | 0.930 ± 0.001 | 1.426 ± 0.018 | 0.449 ± 0.001 | 0.527 ± 0.007 | 0.112 ± 0.001 | 0.393 ± 0.004 | 6.680 ± 0.044 | 10.289 ± 0.040 | 498.2 ± 8.931 | 2317.0 ± 22.609 | 9.667 ± 0.105 |
| | | Rouge-Avg | 50 | T5 | 0.536 ± 0.001 | 0.198 ± 0.002 | 0.433 ± 0.002 | 0.433 ± 0.002 | 0.311 ± 0.002 | 0.070 ± 0.002 | 0.929 ± 0.001 | 1.421 ± 0.028 | 0.446 ± 0.004 | 0.526 ± 0.004 | 0.114 ± 0.002 | 0.395 ± 0.005 | 6.682 ± 0.0297 | 10.274 ± 0.042 | 506.4 ± 6.829 | 2326.4 ± 41.778 | 9.614 ± 0.102 |
| | | Meteor | 50 | T5 | 0.540 ± 0.005 | 0.204 ± 0.005 | 0.436 ± 0.004 | 0.436 ± 0.004 | 0.329 ± 0.003 | 0.076 ± 0.003 | 0.930 ± 0.001 | 1.474 ± 0.022 | 0.447 ± 0.004 | 0.514 ± 0.004 | 0.105 ± 0.002 | 0.378 ± 0.008 | 6.631 ± 0.053 | 10.270 ± 0.064 | 507.0 ± 17.146 | 2424.6 ± 72.550 | 10.551 ± 0.271 |
| | | SPice | 50 | T5 | 0.532 ± 0.006 | 0.194 ± 0.007 | 0.430 ± 0.005 | 0.430 ± 0.005 | 0.311 ± 0.004 | 0.068 ± 0.003 | 0.929 ± 0.001 | 1.415 ± 0.029 | 0.458 ± 0.001 | 0.532 ± 0.008 | 0.113 ± 0.0038 | 0.392 ± 0.009 | 6.736 ± 0.058 | 10.338 ± 0.057 | 507.4 ± 14.319 | 2313.8 ± 27.694 | 9.742 ± 0.208 |
| | | CiDer | 50 | T5 | 0.530 ± 0.004 | 0.191 ± 0.003 | 0.427 ± 0.004 | 0.427 ± 0.004 | 0.309 ± 0.008 | 0.063 ± 0.002 | 0.928 ± 0.001 | 1.337 ± 0.040 | 0.444 ± 0.002 | 0.518 ± 0.009 | 0.110 ± 0.003 | 0.382 ± 0.006 | 6.614 ± 0.082 | 10.166 ± 0.053 | 490.4 ± 9.457 | 2295.4 ± 51.554 | 9.838 ± 0.265 |
| | | SpiDer | 50 | T5 | 0.536 ± 0.002 | 0.197 ± 0.002 | 0.430 ± 0.002 | 0.430 ± 0.002 | 0.313 ± 0.002 | 0.064 ± 0.002 | 0.928 ± 0.001 | 1.374 ± 0.018 | 0.445 ± 0.003 | 0.524 ± 0.007 | 0.112 ± 0.001 | 0.394 ± 0.004 | 6.673 ± 0.066 | 10.247 ± 0.066 | 504.8 ± 7.440 | 2361.8 ± 20.856 | 9.761 ± 0.121 |
| | Supervised + NLPO | Rouge-1 | 50 | T5 | 0.545 ± 0.002 | 0.197 ± 0.002 | 0.432 ± 0.001 | 0.432 ± 0.001 | 0.31 ± 0.002 | 0.068 ± 0.001 | 0.929 ± 0.0 | 1.41 ± 0.012 | 0.449 ± 0.001 | 0.529 ± 0.002 | 0.114 ± 0.002 | 0.399 ± 0.005 | 6.705 ± 0.018 | 10.301 ± 0.03 | 498.86 ± 8.594 | 2311.46 ± 33.451 | 9.463 ± 0.111 |
| | | Rouge-Avg | 50 | T5 | 0.541 ± 0.003 | 0.2 ± 0.003 | 0.435 ± 0.002 | 0.435 ± 0.002 | 0.313 ± 0.002 | 0.07 ± 0.002 | 0.93 ± 0.0 | 1.464 ± 0.025 | 0.448 ± 0.002 | 0.53 ± 0.006 | 0.113 ± 0.002 | 0.396 ± 0.008 | 6.708 ± 0.05 | 10.318 ± 0.074 | 493.64 ± 10.068 | 2319.42 ± 55.738 | 9.596 ± 0.123 |
| | | Meteor | 50 | T5 | 0.537 ± 0.003 | 0.201 ± 0.004 | 0.431 ± 0.002 | 0.431 ± 0.002 | 0.326 ± 0.002 | 0.074 ± 0.003 | 0.93 ± 0.0 | 1.464 ± 0.025 | 0.448 ± 0.002 | 0.516 ± 0.006 | 0.106 ± 0.002 | 0.377 ± 0.008 | 6.634 ± 0.044 | 10.26 ± 0.077 | 506.04 ± 3.502 | 2401.32 ± 38.569 | 10.453 ± 0.194 |
| | | SPice | 50 | T5 | 0.535 ± 0.007 | 0.193 ± 0.008 | 0.429 ± 0.005 | 0.429 ± 0.005 | 0.3 ± 0.003 | 0.064 ± 0.002 | 0.927 ± 0.001 | 1.333 ± 0.017 | 0.459 ± 0.003 | 0.553 ± 0.013 | 0.12 ± 0.004 | 0.415 ± 0.014 | 6.908 ± 0.118 | 10.445 ± 0.057 | 508.075 ± 4.669 | 2343.3 ± 53.274 | 9.249 ± 0.225 |
| | | CiDer | 50 | T5 | 0.533 ± 0.003 | 0.197 ± 0.004 | 0.43 ± 0.003 | 0.43 ± 0.004 | 0.314 ± 0.004 | 0.066 ± 0.001 | 0.929 ± 0.0 | 1.381 ± 0.014 | 0.446 ± 0.004 | 0.516 ± 0.009 | 0.108 ± 0.003 | 0.379 ± 0.01 | 6.583 ± 0.077 | 10.165 ± 0.084 | 490.78 ± 4.734 | 2304.52 ± 62.068 | 9.923 ± 0.213 |
| | | SPider | 50 | T5 | 0.532 ± 0.006 | 0.196 ± 0.006 | 0.431 ± 0.004 | 0.431 ± 0.004 | 0.314 ± 0.004 | 0.066 ± 0.002 | 0.929 ± 0.0 | 1.371 ± 0.011 | 0.448 ± 0.002 | 0.521 ± 0.005 | 0.109 ± 0.002 | 0.385 ± 0.005 | 6.623 ± 0.034 | 10.223 ± 0.049 | 485.325 ± 5.683 | 2297.575 ± 21.271 | 9.798 ± 0.179 |

Table 13: **CommonGen dev evaluation**: Table shows lexical, semantic and diversity metrics for best performing models found in each algorithm-reward function combinations along with best performing supervised baseline models. Generated text from these models are submitted to official CommonGen test evaluation to obtain test scores presented in Table 12

| Algorithm | Unique N | Coherence | | | Commonsense | | |
|---|---|---|---|---|---|---|---|
| | | Value | Alpha | Skew | Value | Alpha | Skew |
| PPO+Supervised | 25 | 4.14 | 0.073 | 4.137 | 4.03 | 0.137 | 4.023 |
| NLPO+Supervised | 26 | 4.25 | 0.036 | 4.253 | 4.16 | 0.002 | 4.163 |
| Zero Shot | 24 | 2.15 | 0.391 | 2.154 | 2.29 | 0.342 | 2.291 |
| PPO | 24 | 2.84 | 0.16 | 2.849 | 3.03 | 0.081 | 3.027 |
| Supervised | 23 | 4.39 | 0.159 | 4.387 | 4.21 | 0.225 | 4.209 |
| NLPO | 24 | 2 | 0.335 | 2.003 | 2.13 | 0.265 | 2.124 |

Table 14: Results of the human subject study showing the number of participants N, average Likert scale value for coherence and sentiment, Krippendorf's alpha showing inter-annotator agreement, and Skew. For each model a total of 100 samples were drawn randomly from the test set and rated by 3 annotators each, resulting in 300 data points per algorithm.

| Group 1 | Group 2 | Coherence | | Commonsense | |
|---|---|---|---|---|---|
| | | Diff (G2-G1) | p-values | Diff (G2-G1) | p-values |
| NLPO | PPO | **0.847** | **0.001** | **0.897** | **0.001** |
| NLPO | Supervised | **2.397** | **0.001** | **2.083** | **0.001** |
| NLPO | NLPO+Supervised | **2.257** | **0.001** | **2.033** | **0.001** |
| NLPO | PPO+Supervised | **2.143** | **0.001** | **1.897** | **0.001** |
| NLPO | Zero Shot | 0.153 | 0.515 | 0.157 | 0.624 |
| PPO | Supervised | **1.550** | **0.001** | **1.187** | **0.001** |
| PPO | NLPO+Supervised | **1.410** | **0.001** | **1.137** | **0.001** |
| PPO | PPO+Supervised | **1.297** | **0.001** | **1.000** | **0.001** |
| PPO | Zero Shot | **-0.693** | **0.001** | **-0.740** | **0.001** |
| Supervised | NLPO+Supervised | -0.140 | 0.601 | -0.050 | 0.900 |
| Supervised | PPO+Supervised | **-0.253** | **0.050** | **-0.187** | **0.045** |
| Supervised | Zero Shot | **-2.243** | **0.001** | **-1.927** | **0.001** |
| NLPO+Supervised | PPO+Supervised | **-0.113** | **0.008** | **-0.137** | **0.007** |
| NLPO+Supervised | Zero Shot | **-2.103** | **0.001** | **-1.877** | **0.001** |
| PPO+Supervised | Zero Shot | **-1.990** | **0.001** | **-1.740** | **0.001** |

Table 15: Results of an post-hoc Tukey HSD Test for difference in means between pairs of algorithms (Group 2 - Group 1) and corresponding $p$-values. Individually statistically significant results are bolded and are used to discuss results in the analysis. Overall $p$-values showing that there is a significant difference in means between the models via a one-way ANOVA test are significant with $p \ll 0.05$ for both coherence and sentiment.

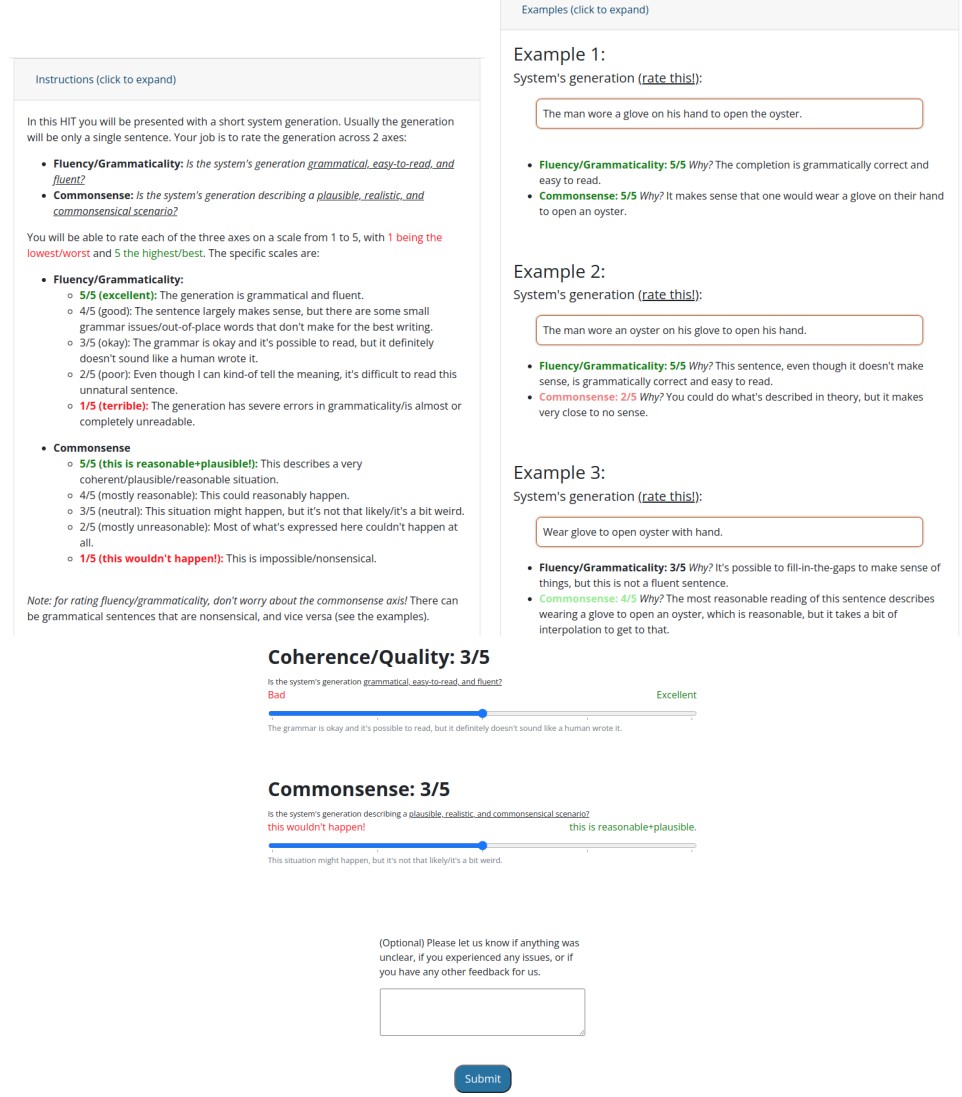

Figure 6: Instructions, examples, and interface for the Commongen task.

### B.4.3 HUMAN PARTICIPANT STUDY

Figure 6 shows the commongen instructions, examples, and interface used for the human evaluation experiments. Different from the other human evaluations, we didn't provide any prompt because knowing the set of words to be used isn't required for rating either of the axes. Tables 14, 15 show averaged results, annotator agreement, and the results of statistical significance tests to determine which models output better generations when rated by humans.

### B.4.4 HUMAN PREFERENCE LEARNING EXPERIMENTS

First, we randomly select prompts from the Commongen train dataset and sample a single completion from both the Supervised and Supervised+NLPO models. Next, we filter to prompts where both models at least attempted to use all input concepts. This filtration step was conducted because if a model fails to use all concepts, it may generate a more natural/fluent sentence, but, *a priori*, it shouldn't be preferred by crowdworkers; instead of training crowdworkers to prefer sentences with all concepts, we perform this filter. Figure 7 shows the task presented to the crowdworkers. We then present the prompt and the two completion candidates to 3 unique crowdworkers and ask them to

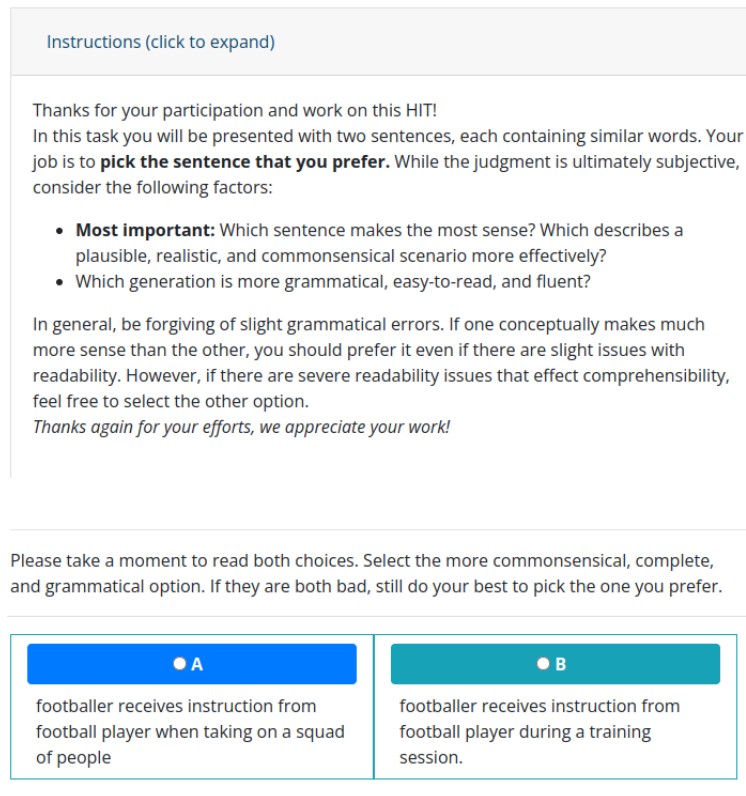

Figure 7: Instructions and interface for the pairwise Commongen HIT.

select which one they prefer with respect to commonsense/fluency; We gathered 3 annotations on 417 pairs (Krippendorf $\alpha = .28$), and split into 60/20/20 train/val/test split. We then trained a reward model, T5-11B Raffel et al. (2020), on the balanced binary classification task of predicting which of the pair was preferred by a majority of 3 annotators, conditioned on the prompt and completion. The resulting model achieved 69.5 test ROC AUC suggesting it indeed captures average human preferences. The model is then used as a reward function. We train Supervised+RL with a METEOR-only reward as a baseline, and compare it to a reward function that uses the fine-tuned T5-11B model. We design the reward function based on the preference model as $r = meteor + pref/(1 + |miss|)$ where $miss$ is a set of concepts not covered in the generated text, in an attempt to mimic the data collection process that humans are instructed to follow. This reward function accounts for both the task of using all concepts and also human's preferences for how a sentence should look within the constraints stipulated by the task. Finally, we rerun the same pairwise preference collection procedure—this time sampling from Commongen test—with human participants to compare the generations from a preference optimized RL policy to the previously best Supervised+NLPO policy. Comparing the METEOR-only to the preference model head-to-head, the generations produced by the human feedback model are preferred in 682 cases, compared to the METEOR-only model which is preferred in 587 cases ($p < 0.01$ the models are equally preferred).

### B.4.5 QUALITATIVE ANALYSIS

This section shows sample generations from different algorithms for three randomly picked prompts.

```
Sample 1
Prompt: generate a sentence with: apron cut hat kitchen sausage
Zero-Shot: generate a sentence with: apron cut hat kitchen sausage.
PPO: sausage in the kitchen on an apron.
NLPO: sausage cut hat cut hat cut hat cut apron cut hat
Supervised: A man is cutting sausage on an apron in a kitchen.
Supervised + PPO: A man cutting sausage from a hat on an apron in the kitchen.
Supervised + NLPO: A man cuts sausage in a hat on an apron in the kitchen.

Sample 2
```

**Prompt:** generate a sentence with: backhand competition player reach shuttlecock
**Zero-Shot:** backhand competition player reach shuttlecock.
**PPO:** the ball in the backhand and the competition player in the shuttlecock.
**NLPO:** backhanded competition player reaches shuttlecock.
**Supervised:** tennis player reaches for a shuttlecock during a competition.
**Supervised** + **PPO:** football player reaches for a shuttlecock in backhand during the competition.

**Supervised** + **NLPO:** football player reaches for a shuttlecock after hitting a backhand during the competition.

**Sample** 3
**Prompt:** generate a sentence with: bowl combine cucumber onion pasta
**Zero-Shot:** cucumber onion pasta.
**PPO:** the cucumber onion pasta in a small bowl.
**NLPO:** the in a bowl Combine cucumber onion pasta in a large bowl. Toss pasta
**Supervised:** A bowl containing pasta, tomatoes, cucumbers, and onions.
**Supervised** + **PPO:** A bowl containing pasta topped with cucumbers, onions, and peppers.
**Supervised** + **NLPO:** A bowl containing a mixture of pasta, cucumber, and onion.

## B.5 CNN DAILY MAIL

### B.5.1 SETUP

As a representative of the summarization task, we consider CNN/DM dataset consisting of long news articles and their highlights written by news authors. The dataset consists of 287k training, 13k validation and 11k test examples. We trained RL methods using 3 different automated metrics, namely Rouge-1, Rouge-avg and Meteor. We chose T5 as our base LM as it is pre-trained in a unified text-to-text framework and relishes Zero-Shot capabilities. For decoding, we use multinomial sampling with a temperature of 0.7 for all the models.

| Model Params | value |
|---|---|
| supervised | batch size: 16
epochs: 2
learning rate: 0.0001
learning rate scheduler: cosine
weight decay: 0.1 |
| ppo/ nlpo | steps per update: 5120
total number of steps: 512000
batch size: 64
epochs per update: 5
learning rate: 0.000002
entropy coefficient: 0.0
initial kl coeff: 0.001
target kl: 0.2
discount factor: 0.99
gae lambda: 0.95
clip ratio: 0.2
value function coeff: 0.5
rollouts top k: sweep of (50,100)
top mask ratio: 0.9
target update iterations: sweep of (10, 20, 30) |
| supervised+ppo/ nlpo | steps per update: 5120
total number of steps: 256000
batch size: 64
epochs per update: 5
learning rate: 0.000002
entropy coefficient: 0.0
initial kl coeff: 0.01
target kl: 0.2
discount factor: 0.99
gae lambda: 0.95
clip ratio: 0.2
value function coeff: 0.5
rollouts top k: sweep of (50,100)
top mask ratio: 0.9
target update iterations: sweep of (10, 20, 30) |
| decoding | sampling: True
temperature: 0.7
min length: 50
max new tokens: 100 |
| tokenizer | padding side: left
truncation side: right
max length: 512 |

Table 16: **CNN/DM Hyperparams**: Table shows a list of all hyper-parameters and their settings

### B.5.2 RESULTS AND DISCUSSION

Table 17 presents benchmarking results on test set reporting a wide range of metrics: lexical, semantic, factual correctness and diversity metrics. As baselines, we report lead-3 which selects first three sentences as the summary, Zero-Shot and a supervised model. PPO and NLPO models are on par with supervised performance on several metrics including Rouge-2, Rouge-L, and Bleu. On fine-tuning on top of supervised model, performance improves consistently on all metrics indicating that RL fine-tuning is beneficial. Another interesting finding is that, RL fine-tuned models are factually consistent as measured by SummaCZS metric. For ablations on PPO params, NLPO params, we refer to Tables 18,19.

| Tasks | Alg | Reward Function | LM | Rouge-1 | Rouge-2 | Rouge-L | Rouge-LSum | Meteor | BLEU | BertScore | SummaCZS | MSTTR | Distinct$_1$ | Distinct$_2$ | H$_1$ | H$_2$ | Unique$_1$ | Unique$_2$ | Mean Output Length |
|---|---|---|---|---|---|---|---|---|---|---|---|---|---|---|---|---|---|---|---|
| | | | | | | | **Lexical and Semantic Metrics** | | | | **Factual Consistency** | | | **Diversity Metrics** | | | | | |
| | Lead-3 | | | 0.401 | 0.175 | 0.250 | 0.363 | 0.333 | 0.099 | 0.874 | 0.993 | 0.750 | 0.0482 | 0.386 | 10.481 | 16.631 | 21465 | 273153 | 84 |
| | Zero-Shot | | T5 | 0.372 | 0.145 | 0.247 | 0.311 | 0.256 | 0.077 | 0.864 | 0.654 | 0.725 | 0.061 | 0.414 | 10.285 | 16.183 | 19113 | 193999 | 55 |
| | PPO | Rouge-1 | T5 | 0.410 | 0.182 | 0.283 | 0.349 | 0.276 | 0.095 | 0.876 | 0.622 | 0.760 | 0.068 | 0.464 | 10.661 | 16.437 | 18189 | 191383 | 47 |
| | | Rouge-Avg | T5 | 0.396 | 0.176 | 0.273 | 0.338 | 0.270 | 0.095 | 0.874 | 0.622 | 0.773 | 0.071 | 0.490 | 10.830 | 16.664 | 19478 | 209140 | 48 |
| | | Meteor | T5 | 0.408 | 0.178 | 0.276 | 0.342 | 0.301 | 0.109 | 0.873 | 0.527 | 0.765 | 0.060 | 0.447 | 10.699 | 16.688 | 20528 | 234386 | 61 |
| | NLPO | Rouge-1 | T5 | 0.404 | 0.180 | 0.278 | 0.344 | 0.275 | 0.096 | 0.875 | 0.636 | 0.771 | 0.069 | 0.480 | 10.789 | 16.618 | 18677 | 201971 | 48 |
| | | Rouge-Avg | T5 | 0.404 | 0.177 | 0.279 | 0.344 | 0.274 | 0.094 | 0.874 | 0.586 | 0.765 | 0.066 | 0.476 | 10.744 | 16.620 | 18179 | 206368 | 50 |
| | | Meteor | T5 | 0.405 | 0.180 | 0.277 | 0.343 | 0.292 | 0.108 | 0.872 | 0.578 | 0.772 | 0.064 | 0.471 | 10.802 | 16.766 | 20212 | 231038 | 56 |
| CNN/DM | Supervised | | T5 | 0.411 | 0.177 | 0.276 | 0.343 | 0.309 | 0.108 | 0.876 | 0.654 | 0.727 | 0.057 | 0.401 | 10.459 | 16.410 | 21096 | 230343 | 68 |
| | Supervised + PPO | Rouge-1 | T5 | 0.417 | 0.189 | 0.294 | 0.358 | 0.278 | 0.101 | 0.882 | 0.722 | 0.750 | 0.070 | 0.459 | 10.595 | 16.389 | 18184 | 184220 | 46 |
| | | Rouge-Avg | T5 | 0.425 | 0.194 | 0.297 | 0.363 | 0.296 | 0.114 | 0.882 | 0.728 | 0.747 | 0.066 | 0.445 | 10.589 | 16.458 | 18939 | 200617 | 52 |
| | | Meteor | T5 | 0.426 | 0.194 | 0.293 | 0.361 | 0.316 | 0.125 | 0.880 | 0.726 | 0.741 | 0.059 | 0.420 | 10.532 | 16.491 | 20395 | 224432 | 63 |
| | Supervised + NLPO | Rouge-1 | T5 | 0.421 | 0.193 | 0.297 | 0.361 | 0.287 | 0.108 | **0.882** | 0.740 | 0.748 | 0.067 | 0.446 | 10.528 | 16.313 | 18204 | 185561 | 48 |
| | | Rouge-Avg | T5 | 0.424 | 0.193 | 0.296 | 0.363 | 0.295 | 0.115 | 0.882 | 0.743 | 0.744 | 0.065 | 0.443 | 10.570 | 16.444 | 18747 | 201705 | 53 |
| | | Meteor | T5 | **0.429** | 0.194 | 0.293 | 0.361 | **0.319** | 0.124 | 0.880 | 0.743 | 0.745 | 0.059 | 0.422 | 10.574 | 16.516 | 20358 | 226801 | 63 |

Table 17: **CNN/Daily Mail test evaluation**: Table presents a wide range of metrics: lexical, semantic, factual correctness and diversity metrics on test set. As baselines, we report lead-3 which selects first three sentences as the summary, Zero-Shot and a supervised model. PPO and NLPO models are on par with supervised performance on several metrics including Rouge-2, Rouge-L, and Bleu. On fine-tuning on top of supervised model, performance improves consistently on all metrics indicating that RL fine-tuning is beneficial. Another interesting finding is that, RL fine-tuned models are factually consistent as measured by SummaCZS metric.

| Alg | Reward Function | Top k | Lexical and Semantic Metrics | | | | | | |
|---|---|---|---|---|---|---|---|---|---|
| | | | Rouge-1 | Rouge-2 | Rouge-L | Rouge-LSum | Meteor | BLEU | BertScore |
| PPO | Rouge-1 | 50 | 0.404 | 0.181 | 0.280 | 0.346 | 0.273 | **0.095** | 0.874 |
| | | 100 | **0.412** | **0.186** | **0.286** | **0.354** | **0.276** | 0.094 | **0.876** |
| | Rouge-Avg | 50 | **0.401** | 0.177 | **0.276** | 0.342 | **0.271** | 0.092 | 0.873 |
| | | 100 | 0.399 | **0.179** | 0.275 | **0.342** | 0.270 | **0.094** | **0.874** |
| | Meteor | 50 | **0.413** | **0.182** | **0.279** | **0.348** | **0.301** | **0.110** | **0.873** |
| | | 100 | 0.409 | 0.179 | 0.276 | 0.345 | 0.296 | 0.108 | 0.871 |
| Supervised+PPO | Rouge-1 | 50 | 0.414 | 0.190 | 0.293 | 0.358 | 0.272 | 0.097 | 0.881 |
| | | 100 | **0.420** | **0.193** | **0.295** | **0.362** | **0.277** | **0.100** | 0.881 |
| | Rouge-Avg | 50 | 0.426 | 0.196 | 0.298 | 0.366 | 0.294 | **0.114** | 0.881 |
| | | 100 | **0.427** | **0.196** | **0.298** | **0.366** | **0.294** | 0.113 | 0.881 |
| | Meteor | 50 | 0.429 | 0.197 | 0.297 | 0.367 | 0.306 | 0.122 | **0.881** |
| | | 100 | **0.432** | **0.199** | **0.297** | **0.367** | **0.317** | **0.131** | 0.879 |

Table 18: **PPO Ablation/Model Selection**: Evaluation of PPO models on validation set with different reward functions and top k values for rollouts. For each alg-reward combo, best model (top k ) is chosen.

| Alg | Reward Function | Top k (rollout) | Top p (Action mask) | target update $n_{i}ters$ | Lexical and Semantic Metrics | | | | | | |
|---|---|---|---|---|---|---|---|---|---|---|---|
| | | | | | Rouge-1 | Rouge-2 | Rouge-L | Rouge-LSum | Meteor | BLEU | BertScore |
| NLPO | Rouge-1 | 50 | 0.9 | 10 | 0.400 | 0.178 | 0.275 | 0.343 | 0.269 | 0.094 | 0.872 |
| | | | | 20 | 0.396 | 0.173 | 0.274 | 0.340 | 0.257 | 0.082 | 0.873 |
| | | | | 30 | 0.396 | 0.174 | 0.273 | 0.339 | 0.265 | 0.091 | 0.872 |
| | | 100 | 0.9 | 10 | **0.407** | 0.177 | 0.279 | 0.347 | 0.265 | 0.085 | **0.875** |
| | | | | 20 | 0.406 | **0.182** | **0.281** | **0.347** | **0.273** | **0.094** | 0.874 |
| | | | | 30 | 0.405 | 0.180 | 0.279 | 0.347 | 0.269 | 0.091 | 0.875 |
| | Rouge-Avg | 50 | 0.9 | 10 | 0.400 | **0.180** | 0.276 | 0.343 | 0.271 | 0.096 | 0.873 |
| | | | | 20 | 0.349 | 0.147 | 0.241 | 0.298 | 0.237 | 0.078 | 0.858 |
| | | | | 30 | 0.393 | 0.173 | 0.272 | 0.336 | 0.267 | 0.092 | 0.870 |
| | | 100 | 0.9 | 10 | 0.396 | 0.174 | 0.274 | 0.339 | 0.265 | 0.088 | 0.872 |
| | | | | 20 | **0.406** | 0.179 | **0.280** | **0.347** | **0.272** | **0.092** | **0.874** |
| | | | | 30 | 0.400 | 0.178 | 0.279 | 0.344 | 0.266 | 0.087 | 0.874 |
| | Meteor | 50 | 0.9 | 10 | 0.404 | 0.177 | 0.274 | 0.343 | 0.286 | 0.102 | 0.872 |
| | | | | 20 | 0.406 | 0.180 | 0.276 | 0.343 | 0.292 | 0.107 | 0.871 |
| | | | | 30 | 0.401 | 0.172 | 0.271 | 0.337 | 0.288 | 0.099 | 0.870 |
| | | 100 | 0.9 | 10 | 0.405 | 0.178 | 0.276 | 0.343 | **0.294** | 0.107 | 0.870 |
| | | | | 20 | 0.406 | 0.176 | 0.276 | 0.343 | 0.291 | 0.106 | 0.872 |
| | | | | 30 | **0.409** | **0.184** | **0.280** | **0.348** | 0.291 | **0.108** | **0.873** |
| Supervised + NLPO | Rouge-1 | 50 | 0.9 | 10 | **0.425** | 0.196 | **0.299** | **0.366** | 0.285 | 0.106 | **0.882** |
| | | | | 20 | 0.417 | 0.191 | 0.295 | 0.360 | 0.276 | 0.100 | 0.881 |
| | | | | 30 | 0.418 | 0.192 | 0.296 | 0.361 | 0.278 | 0.101 | 0.881 |
| | | 100 | 0.9 | 10 | 0.424 | 0.196 | 0.299 | 0.366 | 0.286 | 0.106 | 0.882 |
| | | | | 20 | 0.423 | 0.196 | 0.299 | 0.365 | **0.289** | **0.110** | 0.881 |
| | | | | 30 | 0.420 | 0.193 | 0.296 | 0.362 | 0.279 | 0.102 | 0.881 |
| | Rouge-Avg | 50 | 0.9 | 10 | 0.426 | 0.197 | 0.298 | 0.367 | 0.294 | 0.115 | 0.881 |
| | | | | 20 | 0.425 | 0.196 | 0.298 | 0.366 | 0.292 | 0.112 | 0.881 |
| | | | | 30 | 0.424 | 0.194 | 0.297 | 0.365 | 0.287 | 0.107 | 0.881 |
| | | 100 | 0.9 | 10 | 0.424 | 0.196 | 0.298 | 0.365 | 0.291 | 0.113 | 0.881 |
| | | | | 20 | 0.428 | 0.198 | 0.300 | 0.368 | 0.296 | 0.115 | 0.882 |
| | | | | 30 | **0.429** | **0.199** | **0.300** | **0.369** | 0.296 | **0.116** | **0.882** |
| | Meteor | 50 | 0.9 | 10 | 0.430 | 0.197 | 0.294 | 0.364 | 0.320 | 0.130 | 0.879 |
| | | | | 20 | 0.432 | 0.198 | 0.297 | 0.367 | 0.318 | 0.130 | 0.880 |
| | | | | 30 | 0.423 | 0.191 | 0.293 | 0.361 | 0.297 | 0.116 | 0.879 |
| | | 100 | 0.9 | 10 | **0.435** | **0.200** | **0.298** | **0.369** | **0.320** | 0.131 | **0.881** |
| | | | | 20 | 0.433 | 0.198 | 0.297 | 0.368 | 0.319 | 0.130 | 0.879 |
| | | | | 30 | 0.434 | 0.200 | 0.297 | 0.369 | 0.324 | **0.132** | 0.879 |

Table 19: **NLPO Ablation/Model Selection**: Evaluation of NLPO models on validation set with different reward functions, top k values for rollouts and target update iterations. For each alg-reward combo, best model is chosen

| Algorithm | Unique N | Coherence | | | Quality | | |
|---|---|---|---|---|---|---|---|
| | | Value | Alpha | Skew | Value | Alpha | Skew |
| PPO+Supervised | 22 | 4.21 | 0.198 | 4.224 | 3.97 | 0.256 | 3.98 |
| NLPO+Supervised | 19 | 4.3 | 0.26 | 4.308 | 3.98 | 0.089 | 4 |
| Zero Shot | 17 | 3.73 | 0.1 | 3.757 | 3.69 | 0.25 | 3.722 |
| Supervised | 19 | 4.25 | 0.116 | 4.241 | 3.99 | 0.2 | 3.986 |
| NLPO | 17 | 4.03 | 0.13 | 4.042 | 3.83 | 0.191 | 3.832 |
| PPO | 21 | 3.94 | 0.111 | 3.945 | 3.76 | 0.129 | 3.767 |
| Human | 19 | 3.89 | 0.277 | 3.902 | 3.77 | 0.029 | 3.769 |

Table 20: Results of the human subject study showing the number of participants N, average Likert scale value for coherence and sentiment, Krippendorf's alpha showing inter-annotator agreement, and Skew. For each model a total of 50 samples were drawn randomly from the test set and rated by 3 annotators each, each resulting in 150 data points per algorithm.

| Group 1 | Group 2 | Coherence | | Quality | |
|---|---|---|---|---|---|
| | | Diff (G2-G1) | *p-values* | Diff (G2-G1) | *p-values* |
| Human | NLPO | 0.147 | 0.755 | 0.060 | 0.900 |
| Human | NLPO+Supervised | **0.413** | **0.001** | **0.213** | **0.047** |
| Human | PPO | 0.053 | 0.900 | -0.007 | 0.900 |
| Human | PPO+Supervised | **0.327** | **0.024** | 0.200 | 0.544 |
| Human | Supervised | **0.360** | **0.008** | **0.220** | **0.043** |
| Human | Zero Shot | -0.160 | 0.679 | -0.080 | 0.900 |
| NLPO | NLPO+Supervised | **0.267** | **0.012** | **0.153** | **0.008** |
| NLPO | PPO | -0.093 | 0.900 | -0.067 | 0.900 |
| NLPO | PPO+Supervised | 0.180 | 0.564 | 0.140 | 0.860 |
| NLPO | Supervised | 0.213 | 0.361 | 0.160 | 0.754 |
| NLPO | Zero Shot | **-0.307** | **0.044** | -0.140 | 0.860 |
| NLPO+Supervised | PPO | **-0.360** | **0.008** | **-0.220** | **0.043** |
| NLPO+Supervised | PPO+Supervised | **-0.087** | **0.009** | **-0.013** | **0.009** |
| NLPO+Supervised | Supervised | **-0.053** | **0.009** | 0.007 | 0.900 |
| NLPO+Supervised | Zero Shot | **-0.573** | **0.001** | **-0.293** | **0.012** |
| PPO | PPO+Supervised | 0.273 | 0.106 | 0.207 | 0.508 |
| PPO | Supervised | 0.307 | 0.044 | 0.227 | 0.394 |
| PPO | Zero Shot | -0.213 | 0.361 | -0.073 | 0.900 |
| PPO+Supervised | Supervised | 0.033 | 0.900 | 0.020 | 0.900 |
| PPO+Supervised | Zero Shot | -0.487 | 0.001 | -0.280 | 0.155 |
| Supervised | Zero Shot | -0.520 | 0.001 | -0.300 | 0.101 |

Table 21: Results of an post-hoc Tukey HSD Test for difference in means between pairs of algorithms (Group 2 - Group 1) and corresponding $p$-values. Individually statistically significant results are bolded and are used to discuss results in the analysis. Overall $p$-values showing that there is a significant difference in means between the models via a one-way ANOVA test are significant with $p \ll 0.05$ for both coherence and sentiment.

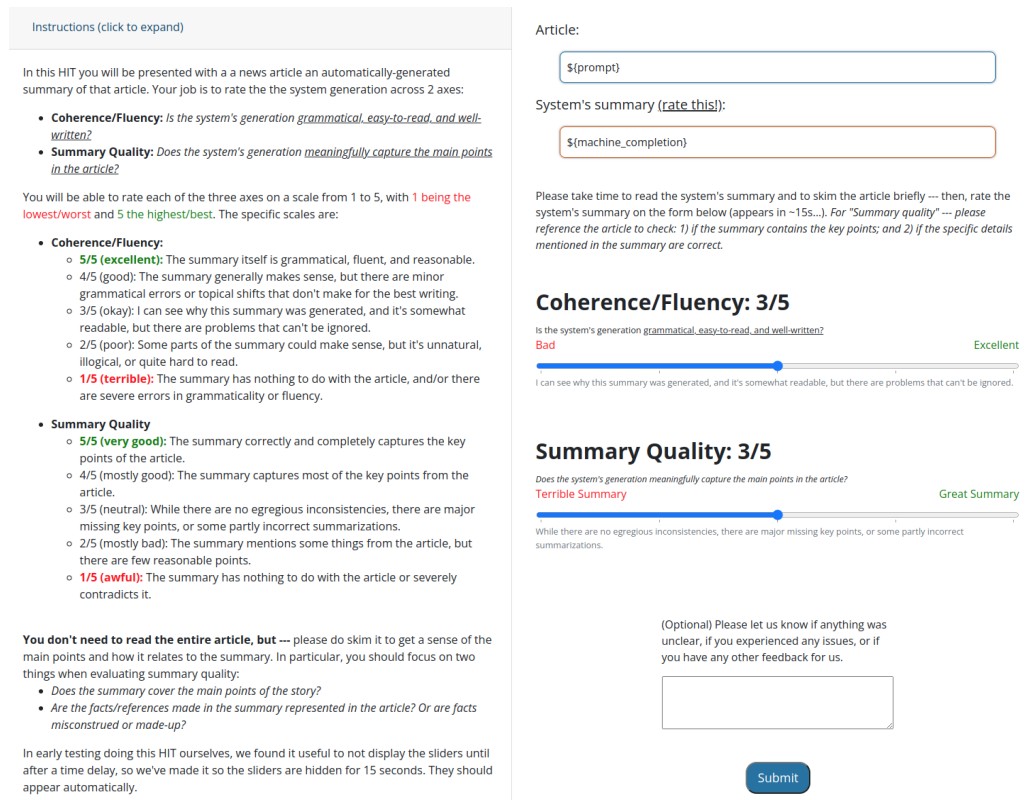

Figure 8: Instructions and interface for the summarization task.

### B.5.3 Human Participant Study

Figure 8 shows the summarization instructions and interface used for the human evaluation experiments. Participants weren't required to read the entire article, but to encourage some reading, a minimum time on the window of 15s was enforced via hiding the sliders. Tables 20, 21 show averaged results, annotator agreement, and the results of statistical significance tests to determine which models output better generations when rated by humans.

### B.5.4 Qualitative Analysis

We show sample generations from each of the algorithms for three randomly picked prompts below.

**Sample** 1

**Prompt:** Manchester City are confident UEFAâĂŹs punishment for breaching financial fairplay regulations will be lifted this summer which would allow them to bid for stellar names like Raheem Sterling, Gareth Bale, Kevin de Bruyne and Ross Barkley. City boss Manuel Pellegrini has been hampered over the past year by UEFA restricting them to a net transfer spend of 49million in each window and keeping the clubâĂŹs overall wage bill to its current level of 205million-a-year. UEFAâĂŹs settlement with City published in May stated those penalties would remain in place until the end of the 2015/16 season but the clubâĂŹs latest financial figures showed drastically-reduced losses of 23million which they feel proves they are now compliant with FPP regulations. Manuel Pellegrini is hoping that the financial restrictions imposed by UEFA for a breach of FFP rules will be lifted at the end of this season . Manchester City have been limited to a net spend of 49 million in the last two transfer windows – they spent 25m bringing Wilfried Bony in from Swansea in January . Ahead of Monday nightâĂŹs trip to Crystal Palace, Pellegrini was certainly talking like a man excited at the prospect of signing 'crack' players this summer. âĂŸI think that next season we donâĂŹt have any restrictions so we will be in the same position that all the other English clubs have,âĂŹ said Pellegrini. âĂŸItâĂŹs important. You have so many strong teams here in England and in Champions League, you can not allow them to keep the advantage every year; having less players to put in your squad or spending less money. We spend money, of course we always spend money, but they spent more.âĂŹ Manchester United, Barcelona, Liverpool and Arsenal have all paid more in transfer fees in the past 12 months than City who were traditionally EuropeâĂŹs biggest

spenders after the club was taken over by Abu Dhabi owners in 2008. Uefa also ordered City to play with a reduced squad from 25 players to 21 in the Champions League this season and while that restriction has now ended, any time reduction in the penalties on spending and wages is more controversial. Arsenal have paid more in transfer fees than City in the last 12 months, including 30m on Alexis Sanchez . The document published last May by UEFAâĂŹs Club Financial Control Body investigative chamber explicitly said CityâĂŹs financial penalties would run for two seasons at least and there has been no official deviation from that decision.

The published statement said at the time: âĂŸManchester City agrees to significantly limit spending in the transfer market for the seasons 2014/15 and 2015/16. It means City will have to argue their case with Uefa that as they have been financially compliant over the past year, they deserve to be free of restrictions moving forward. They have successfully argued their case with UEFA before. Last summer they persuaded the governing body to allow them to bypass the normal quota of eight homegrown players as their Champions League squad had been reduced. Eliaquim Mangala joined the champions from Porto for 32m last summer . The reigning Premier League champions have only paid money for Fernando, Willie Caballero, Eliaquim Mangala and Wilfried Bony in the last two transfer windows and that was part-paid by the sales of Javi Garcia and Jack Rodwell. Pellegrini admitted they werenâĂŹt in a position to compete with neighbours United for established world stars like Angel di Maria because of the FFP penalties. It has cost City on the pitch. Even if they win their remaining eight Premier League fixtures their points tally will be lower than last season and they crashed out once again at the last-16 stage of the Champions League. Pellegrini and director of football Txiki Begiristain both accept the squad needs replenishing and they want to be in the market for the top available players. Manchester City have been heavily linked with a move for former Chelsea midfielder Kevin de Bruyne . The club also want to be involved in any race for Liverpool contract rebel Raheem Sterling . They have been strongly linked with former Chelsea winger Kevin de Bruyne who has been one of the best players in the Bundesliga with Wolfsburg. And if Liverpool do allow Sterling to leave, City want to be involved in the chase for one of EnglandâĂŹs most outstanding talents to help fit in with quotas for homegrown players. âĂŸIf you are the manager or a player of a big tem, you are always open to criticism because everyone expects a high level of performance,âĂŽ he said. âĂŸThe major issue with our team always seems to be the money but this season I think we were the team that spent less than any other team. That is the weird thing. âĂŸFor this club trying to be a big team in so few years, maybe we are paying the cost for that.âĂŽ Since the transfer penalties were introduced, City have spent 91million on players in the last two windows ( Net spend 68million). That compares to Manchester United 145.5million, Liverpool 113 million, Arsenal 92.5million and Chelsea 82.6million. Over the same time period Barcelona spent 118.3million on players and Real Madrid 81.2million though they also broke the world transfer record for Gareth Bale the previous summer.

**Zero-Shot:** manuel Pellegrini hoping UEFAâĂŹs punishment for breaching financial fairplay regulations will be lifted this summer. city have been restricted to a net transfer spend of 49million in each window. a total of 23million has been lost in the last year.

**References:** Manchester City have been restricted to a net transfer spend of 49m .Club's also had to keep overall wage bill to its current level of 205m .Punishments imposed by UEFA for breaching Financial Fair Play rules .The spending restrictions were set for this season and the next one .But City are confident they will be lifted early after their compliance .

**PPO:** Manuel Pellegrini is hoping UEFAâĂŹs punishment for breaching financial fairplay rules will be lifted this summer. Manchester City have been limited to a net spend of 49million in each transfer window. the club also keep the club's overall wage bill to its current level of 205million-a-year.

**NLPO:** Manchester City are confident UEFAâĂŹs punishment for breaching financial fairplay regulations will be lifted this summer. that would allow them to bid for stellar names like Raheem Sterling, Kevin de Bruyne and Ross Barkley.

**Supervised:** Manchester City have been limited to a net transfer spend of 49million in each window. City's overall wage bill is 205m-a-year due to a breach of UEFA rules. City are confident the financial restrictions will be lifted at the end of the current season in order to bid for stars such as Raheem Sterling. Manuel Pellegrini is hoping that the financial restrictions will be lifted at the end of the 2015/16 season

**Supervised + PPO:** Manchester City have been restricted to a net spend of 49million in each window and kept their overall wage bill at 205m. Manuel Pellegrini believes these restrictions will be lifted at the end of the season. Arsenal have spent more in transfer fees than City in the last 12 months, including 30m on Alexis Sanchez.

**Supervised + NLPO:** Manchester City are confident UEFAâĂŹs punishment for a breach of financial fairplay regulations will be lifted this summer. The club have been restricted to a net transfer spend of 49million over the past two window. They aim to bid for stars like Raheem Sterling, Gareth Bale and Kevin de Bruyne. City lost 23m on transfers in the last 12 months due to financial restrictions imposed by UEFA for breach

**Sample** 2

**Prompt:** (CNN)Larry Johnson remembers the fear and feeling of helplessness from being on the SkyWest Airlines flight that made an emergency landing in Buffalo, New York. "I thought we were done," he said Thursday, one day later. "There was no doubt in my mind that we weren't going to land." Johnson was flying with his brother, his girlfriend and his 8-month-old son when he says a flight attendant came over the speaker asking for someone who was medically trained to help with a sick passenger. Minutes later, Johnson says, the attendant announced there was a pressurization problem and told passengers to prepare for the emergency landing. "You're going to feel dizzy and woozy and lightheaded, and you'

re not going to be able to breathe," Johnson recalled the flight attendant saying. The Federal Aviation Administration on Wednesday initially reported a pressurization problem with SkyWest Flight 5622, and said it would investigate. It later issued a statement that did not reference any pressurization issues. "It was like being trapped and you couldn't do anything. You have no control over it. Your body just felt pressured," Johnson said. Johnson said his girlfriend was seeing spots and that "when I was blinking, it was really bright. I could barely see." According to Marissa Snow, spokeswoman for SkyWest, three passengers reported a loss of consciousness while on the flight. Fifteen passengers were examined by medical personnel at the Buffalo airport, but no one was transported to the hospital, she said. The spokeswoman said that maintenance personnel found no indication of a pressurization problem with the aircraft, an Embraer E170, and that the airline continues to investigate the cause. An official with the National Transportation Safety Board told CNN that the agency is in communication with the FAA and SkyWest to gather information on the incident to better understand what took place.

Mary Cunningham was the nurse on board who attended to the first sick passenger. "There was a woman, very lethargic, very nauseous, turning gray, just not looking good," Cunningham said. After administering oxygen to the woman, Cunningham, a nurse at Yale-New Haven Hospital in Connecticut, returned to her seat. Soon after, Cunningham said, she was called up again by the flight attendant after another women near the first started to lose consciousness. "It was something that was affecting oxygen in that area," which was near the middle of the plane, she said. Cunningham said she had to return to her seat after feeling lightheaded. There were 84 passengers on board the plane, including nine crew members. Flight 5622 was originally scheduled to fly from Chicago to Hartford, Connecticut. The plane descended 28,000 feet in three minutes. "It would feel like a roller coaster -- when you're coming over the top and you're going down," CNN aviation analyst Mary Schiavo said, describing how such a descent would feel. "You know that these pilots knew they were in a very grave and very serious situation." Johnson, flying to visit his parents with their only grandchild, was glad that roller coaster ride ended safely. "It was a shaky ride. It wasn't anything I would like to experience again," he said. CNN's Jean Casarez, Josh Gaynor, Dana Ford and Stephanie Gallman contributed to this report.

**Zero-Shot:** "you're going to feel dizzy and woozy and lightheaded," Johnson says. "it was like being trapped and you couldn't do anything," Johnson says. "the plane was trying to get back on the ground," passenger says of fear.

**References:** Three passengers report a loss of consciousness on SkyWest flight .But officials say there is no evidence of a pressurization problem .

**PPO:** SkyWest flight 5312 made emergency landing in Buffalo, new york. passengers were told they were not going to be able to breathe, passenger recalls attendant. the faa initially reported a pressurization problem with the flight 5622.

**NLPO:** SkyWest flight 5622 made emergency landing in buffalo, new york.Larry Johnson says flight attendant announced there was a pressurization problem. three passengers reported loss of consciousness while on flight; no one transported to hospital.

**Supervised:** The FAA says it is investigating the cause of the emergency landing. SkyWest Flight 5622 made an emergency landing in Buffalo, New York. The flight was carrying a family of three people. The flight attendant told the passengers to prepare for the landing.

**Supervised + PPO:** "I thought we were done," the father of one passenger says. The flight attendant announced there was a pressurization problem. The flight made an emergency landing in Buffalo, New York. Three passengers reported a loss of consciousness while on the flight.

**Supervised + NLPO:** SkyWest Airlines flight made an emergency landing in Buffalo, New York, on Thursday. "There was no doubt in my mind that we weren't going to land," passenger says. "You're going to feel dizzy and woozy and lightheaded...you're not going to be able to breathe," he says."

**Sample** 3

**Prompt:** For Inverness the latest chapter in their remarkable story featured not one, but two, plot-changing twists. One featured penalty Celtic didn't get for 2-0 in which Josh Meekings handball should have also led to a sending off. The other the spot kick they did, followed by a red card for Craig Gordon. 'I've not seen it yet, but going by the reaction of the Celtic players we got away with a penalty and a sending off and that was probably the turning point in the game,' acknowledged Caley manager John Hughes after. Inverness's Josh Meekings appears to get away with a handball on the line in their win over Celtic . Caley boss John Hughes says the break, which could have meant a penalty and red card, was a turning point . 'I've not spoken to Josh. I haven't seen it - but going by the media it was definitely a hand ball. We look at the referee behind the line and all that and I know Ronny will feel aggrieved - because I certainly would. 'But it's part and parcel of football and you need a wee bit of luck to beat Celtic. 'This was their biggest game of the season because they will go on and win the league and if they had beaten us today there was a good chance they would have gone on and won the Scottish Cup. 'But when Marley Watkins was clipped by Craig Gordon and they were down to 10 men that was advantage Inverness. 'We weren't going to give Celtic the ball back, they had to come and get it and we had to be patient. 'When big Edward put us into the lead we thought it was going to be our day on the back of things that had happened. 'Celtic equalised with another free kick but it's typical of Inverness that we don't do anything easy. 'We do it the hard way and we came up with the winner through David Raven.' Hughes hauled Raven, his Scouse defender, from his backside as extra-time beckoned. Offended by the sight of one of his players resting he had a message to impart. Caley players celebrate after upsetting Celtic in a Scottish Cup semi-final 3-2 thriller . Celtic, depleted by games and absentees, were virtually on their knees after a relentless programme of midweek games. In last season's League Cup Final Inverness had been passive and unambitious prior

to losing on penalties. This was no time to repeat the mistake. 'I tried to emphasise to the players they would never have a better time to go on and beat Celtic, down to 10 men in the semi final of a cup. We needed to go for it,' Hughes said. 'Before Raven scored at the back post I was looking to change it.

I was going to bring on another winger, Aaron Doran, and put him in the full-back position over on the right, but more advanced so he could take their left back on. Thankfully I didn't do that and David Raven came up with the goal. Virgil Van Dijk (centre) fired Celtic into an early lead with a superb free-kick in the 18th minute . 'I didn't realise this is the first time the club have been in the final of the Scottish Cup and that's a remarkable achievement given it was only formed 20 years ago. 'It is a great story isn't it? It's an absolutely fantastic story. It is 20 odd years since the amalgamation. We are a small provincial club up there in the Highlands. 'We have lost a real inspirational skipper in Richie Foran right from the start of the season. He has never played. We have had to adjust to that. 'We had to sell Billy McKay, our top goalscorer, at Christmas. We have had to go again and adjust. I am a very humble guy and I am grateful and thankful that injuries have never caught up with us.' There is remarkable irony in the fact Falkirk will be the opponents for the final. A former Bairns captain, he was manager of the club in 2009 when they lost to Rangers at Hampden. Former Falkirk captain and manager John Hughes will take on his former club in the final . 'I had a lot of great times at Falkirk. So much so that it is possibly my favourite time in my playing career. I am still friendly with an awful lot of the characters who were in that dressing room. Neil Oliver is a good friend of mine from my Falkirk days. He comes along and sits on the bench and gives me a wee hand out. 'That is the spirit that we had at that club. I have met some great guys. Brian Rice, my ex-coach, Davie Weir, Ian McCall, the list is endless. I was just talking the other day about that day at Falkirk. There are times even now when I see. 'I have a real special, special feel for the place. I am not kidded Falkirk is a right good side, trust me on that. No side goes to Tynecastle and beats Hearts without being a good side. No side beats Hibs the way they were playing without being a good side. You might say we are favourites I say it is even stevens and I am just looking forward to when the days comes.' David Raven (second right) celebrates with his Inverness team-mates following his extra-time winner . An Inverness and Falkirk final would not have been the choice of sponsors William Hill, you suspect. Vast swathes of Hampden's west stand were empty, a 12.15 kick-off for the benefit of broadcasters proving unhelpful . There will be no ticket rows this time. Not over a final unlikely to raise a crowd much higher than 35,000 in a stadium of 52,000. The contract with Hampden plc is signed, with no question of moving it. If the bitter amalgamation of 21 years ago produced a fine football club it also disenfranchised many of those who followed the old ones. 'I hope this encourages the next generation to follow Inverness,' added Hughes. 'In the peer groups it is still Celtic and Rangers they follow but I hope the next ones will want to support their local club. 'If we can add another few hundred or thousand to the support then we are playing our part.

**Zero-Shot:** inverness beat Celtic 2-0 in their opening game of the season. Josh Meekings was the subject of a handball after the break. the break could have meant a penalty and a red card for Craig Gordon.

**References:** Josh Meekings handball should have been a penalty and a red card .If the penalty was awarded and converted Celtic would have gone to 2-0 .Caley manager John Hughes admitted his side were fortunate in win .Virgil van Dijk scored Celtic's opener with a superb free-kick .Celtic keeper Craig Gordon was sent off early in the second half .Greg Tansey, Edward Ofere and finally Daven Raven scored for Caley .

**PPO:** Josh Meekings handball could have meant sending off for Celtic in extra-time win. John Hughes says the break could have meant an penalty and red card. Caley boss also admits he would feel aggrieved had Celtic won the championship.

**NLPO:** Caley beat Celtic 2-0 in their Scottish Cup clash. Josh Meekings handball could have led to a penalty and red card. John Hughes says the break was probably the turning point in the game. defender does not believe defensive striker can get away with being shown the yellow card - despite playing in same half we never scored.

**Supervised:** Inverness beat Celtic 2-0 in their Scottish Cup clash. Josh Meekings appeared to get away with a handball on the line. Another player was sent off for a spot kick which ended in a red card. Caley manager John Hughes says the break could have meant a penalty. Hughes says he 'would feel aggrieved' after the break.

**Supervised + PPO:** Caley beat Celtic 1-0 in their Scottish Cup encounter at the Etihad. Josh Meekings' handball on the line should have also seen a sending off. But Craig Gordon got away with a spot kick and sending off. John Hughes said it was a turning point in the game for Inverness.

**Supervised + NLPO:** Josh Meekings appears to get away with a handball after Inverness's win. Inverness were down to 10 men when Craig Gordon was sent off. John Hughes says the break could have meant a penalty and red card. Hughes says Celtic will go on and win the Scottish Cup with win

## B.6    ToTTo

### B.6.1    Setup

ToTTo (Parikh et al., 2020) is a controlled table-to-text generation task in which the goal is to produce one-sentence description of highlighted table cells. For training RL methods, we consider 5 different reward functions: BLEU, SacreBLEU, METEOR, PARENT and a combination of Meteor and PARENT. We chose T5 as our base LM here too, as they are more suitable for structure to text tasks. For decoding, we use beam search during inference and for generating rollouts, we use top k sampling. Other implementation details are captured in Table 22.

| Model Params | value |
|---|---|
| supervised | batch size: 8
epochs: 4
learning rate: 0.0001
learning rate scheduler: constant with warm up
weight decay: 0.1 |
| ppo/nlpo | steps per update: 2560
total number of steps: 256000
batch size: 64
epochs per update: 5
learning rate: 0.000002
entropy coefficient: 0.0
initial kl coeff: 0.001
target kl: 2.0
discount factor: 0.99
gae lambda: 0.95
clip ratio: 0.2
rollouts top k : 0
value function coeff: 0.5
top mask ratio: 0.9
target update iterations: 20 |
| supervised+ ppo (or nlpo) | steps per update:2560
total number of steps: 256000
batch size: 64
epochs per update: 5
learning rate: 0.0000005
entropy coefficient: 0.0
initial kl coeff: 0.01
target kl: 0.2
discount factor: 0.99
gae lambda: 0.95
clip ratio: 0.2
rollouts top k : 50
value function coeff: 0.5
top mask ratio: 0.9
target update iterations: 20 |
| decoding | num beams: 5
min length: 10
max new tokens: 50 |
| tokenizer | padding side: left
truncation side: right
max length: 512 |

Table 22: **ToTTO Hyperparams**: Table shows a list of all hyper-parameters and their settings

### B.6.2    Results and Discussion

Tables 24, 23 presents our benchmarking results with 5 reward functions along with supervised baseline performances on dev and test sets respectively. Similar to other tasks, our main finding is that warm-started initial policies are crucial for learning to generate descriptions from highlighted cells. Without warm-start, policies suffer from reward hacking and resulting in sub-optimal solutions despite application of task-specific metrics such as PARENT etc. We find that Supervised+NLPO method outperforms all models on ToTTo leaderboard in terms of PARENT metric.

| Tasks | Alg | LM | Reward function | Lexical and Semantic Metrics | | | | | | Factual Consistency | | |
| | | | | SacreBleu | | | BLEURT | | | PARENT | | |
| | | | | Overall | Overlap | Non-Overlap | Overall | Overlap | Non-Overlap | Overall | Overlap | Non-Overlap |
|---|---|---|---|---|---|---|---|---|---|---|---|---|
| | Zero-Shot | T5 | | 0.036 | 0.040 | 0.032 | -1.392 | -1.387 | -1.397 | 0.116 | 0.119 | 0.112 |
| | PPO | T5 | bleu | 0.065 | 0.067 | 0.063 | -1.074 | -1.045 | -1.098 | 0.246 | 0.246 | 0.244 |
| | | T5 | sacrebleu | 0.086 | 0.090 | 0.083 | -0.979 | -0.955 | -1.003 | 0.293 | 0.292 | 0.294 |
| | | T5 | meteor | 0.144 | 0.155 | 0.132 | -0.769 | -0.713 | -0.826 | 0.356 | 0.361 | 0.351 |
| | | T5 | parent | 0.146 | 0.153 | 0.128 | -0.721 | -0.688 | -0.753 | 0.336 | 0.335 | 0.339 |
| | | T5 | meteor + parent | 0.161 | 0.169 | 0.152 | -0.891 | -0.861 | -0.922 | 0.345 | 0.342 | 0.348 |
| | NLPO | T5 | bleu | 0.062 | 0.065 | 0.059 | -1.077 | -1.057 | -1.097 | 0.235 | 0.236 | 0.233 |
| | | T5 | sacrebleu | 0.085 | 0.088 | 0.083 | -0.945 | -0.917 | -0.972 | 0.314 | 0.315 | 0.313 |
| | | T5 | meteor | 0.102 | 0.108 | 0.097 | -1.044 | -1.009 | -1.079 | 0.329 | 0.328 | 0.330 |
| | | T5 | parent | 0.159 | 0.166 | 0.152 | -0.710 | -0.675 | -0.745 | 0.357 | 0.351 | 0.363 |
| ToTTo | | T5 | meteor + parent | **0.166** | **0.175** | **0.158** | **-0.704** | **-0.668** | **-0.740** | **0.365** | **0.362** | **0.368** |
| | Supervised | T5 | | 0.457 | 0.535 | 0.377 | 0.204 | 0.327 | 0.081 | 0.583 | 0.631 | 0.534 |
| | Supervised + PPO | T5 | bleu | 0.473 | 0.548 | 0.395 | 0.200 | 0.323 | 0.078 | 0.590 | 0.638 | 0.542 |
| | | T5 | sacrebleu | 0.474 | 0.557 | 0.389 | **0.209** | **0.340** | 0.077 | 0.573 | 0.620 | 0.525 |
| | | T5 | meteor | 0.468 | 0.541 | 0.392 | 0.203 | 0.325 | **0.082** | 0.590 | 0.638 | 0.542 |
| | | T5 | parent | 0.469 | 0.547 | 0.388 | 0.175 | 0.300 | 0.050 | 0.595 | 0.641 | 0.549 |
| | | T5 | meteor + parent | 0.473 | 0.547 | 0.392 | 0.192 | 0.314 | 0.069 | 0.595 | 0.642 | 0.549 |
| | Supervised + NLPO | T5 | bleu | **0.475** | 0.548 | **0.399** | 0.208 | 0.330 | 0.085 | 0.593 | 0.639 | 0.546 |
| | | T5 | sacrebleu | 0.475 | 0.557 | 0.392 | 0.208 | 0.335 | 0.081 | 0.577 | 0.625 | 0.529 |
| | | T5 | meteor | 0.468 | 0.541 | 0.392 | 0.201 | 0.322 | 0.079 | 0.594 | 0.641 | 0.546 |
| | | T5 | parent | 0.474 | **0.550** | 0.392 | 0.192 | 0.315 | 0.068 | **0.596** | **0.643** | **0.550** |
| | | T5 | meteor + parent | 0.471 | 0.546 | 0.393 | 0.204 | 0.326 | 0.081 | 0.592 | 0.640 | 0.544 |

Table 23: **ToTTo test evaluation**: Table shows lexical, semantic and factual correctness metric scores of algorithms with different reward functions on hold-out test set. Without supervised pre-training, both PPO and NLPO results in sub-optimal solutions, with NLPO better than PPO. With supervised pre-training, PPO and NLPO achieve better scores across all metrics showing RL fine-tuning is beneficial. Most importantly, RL fine-tuned models produce more factually consistent text as seen in higher PARENT scores. Another observation, fine-tuning with a task-specific metric PARENT is better than training on task-agnostic lexical rewards

| Tasks | Alg | LM | Reward function | Rouge-1 | Rouge-2 | Rouge-L | Rouge-LSum | Meteor | BertScore | SacreBleu Overall | SacreBleu Overlap | SacreBleu Non-Overlap | PARENT Overall | PARENT Overlap | PARENT Non-Overlap | MSTTR | Distinct$_1$ | Distinct$_2$ | H$_1$ | H$_2$ | Unique$_1$ | Unique$_2$ | Mean Output Length |
|---|---|---|---|---|---|---|---|---|---|---|---|---|---|---|---|---|---|---|---|---|---|---|---|
| | Zero-Shot | T5 | | 0.131 | 0.055 | 0.127 | 0.127 | 0.057 | 0.805 | 0.038 | 0.042 | 0.034 | 0.118 | 0.119 | 0.116 | 0.428 | 0.084 | 0.238 | 6.703 | 9.933 | 8387 | 26490 | 19.964 |
| | Supervised | T5 | | 0.410 | 0.279 | 0.388 | 0.388 | 0.223 | 0.953 | 0.458 | 0.533 | 0.387 | 0.586 | 0.633 | 0.540 | 0.715 | 0.162 | 0.511 | 9.995 | 14.468 | 15168 | 54706 | 17.791 |
| | PPO | T5 | bleu | 0.274 | 0.138 | 0.249 | 0.249 | 0.139 | 0.844 | 0.068 | 0.071 | 0.066 | 0.251 | 0.250 | 0.251 | 0.403 | 0.091 | 0.308 | 10.659 | 14.511 | 7536 | 34232 | 28.545 |
| | | T5 | sacrebleu | 0.341 | 0.166 | 0.300 | 0.300 | 0.165 | 0.858 | 0.09 | 0.094 | 0.086 | 0.300 | 0.299 | 0.300 | 0.469 | 0.121 | 0.407 | 11.071 | 14.880 | 10138 | 48195 | 26.612 |
| | | T5 | meteor | 0.322 | 0.157 | 0.286 | 0.286 | 0.173 | 0.888 | 0.147 | 0.163 | 0.133 | 0.358 | 0.367 | 0.350 | 0.625 | 0.136 | 0.482 | 10.189 | 14.910 | 12346 | 54925 | 21.484 |
| | | T5 | parent | 0.268 | 0.125 | 0.251 | 0.251 | 0.119 | 0.890 | 0.150 | 0.158 | 0.143 | 0.337 | 0.332 | 0.342 | 0.764 | 0.202 | 0.646 | 11.068 | 14.988 | 13068 | 50313 | 13.035 |
| | | T5 | meteor + parent | 0.266 | 0.128 | 0.251 | 0.251 | 0.130 | 0.886 | 0.165 | 0.175 | 0.155 | 0.348 | 0.346 | 0.350 | 0.702 | 0.181 | 0.594 | 10.096 | 14.432 | 14422 | 55770 | 15.354 |
| | NLPO | T5 | bleu | 0.267 | 0.134 | 0.24 | 0.24 | 0.137 | 0.84 | 0.068 | 0.071 | 0.065 | 0.238 | 0.239 | 0.237 | 0.448 | 0.1 | 0.359 | 11.259 | 14.623 | 9029 | 47209 | 28.472 |
| | | T5 | sacrebleu | 0.341 | 0.168 | 0.297 | 0.297 | 0.183 | 0.863 | 0.089 | 0.093 | 0.085 | 0.32 | 0.324 | 0.317 | 0.494 | 0.111 | 0.373 | 11.007 | 15.032 | 9455 | 43379 | 27.977 |
| ToTTo | | T5 | meteor | 0.322 | 0.157 | 0.286 | 0.286 | 0.173 | 0.888 | 0.147 | 0.163 | 0.133 | 0.358 | 0.367 | 0.350 | 0.625 | 0.136 | 0.482 | 10.189 | 14.910 | 12346 | 54925 | 21.484 |
| | | T5 | parent | 0.283 | 0.132 | 0.264 | 0.264 | 0.133 | 0.894 | 0.163 | 0.174 | 0.153 | 0.36 | 0.357 | 0.364 | 0.824 | 0.223 | 0.691 | 11.493 | 15.127 | 14344 | 55542 | 14.204 |
| | | T5 | meteor + parent | 0.299 | 0.14 | 0.276 | 0.276 | 0.142 | 0.896 | 0.171 | 0.181 | 0.161 | 0.369 | 0.365 | 0.372 | 0.779 | 0.214 | 0.674 | 11.072 | 15.275 | 14939 | 58737 | 15.141 |
| | Supervised + PPO | T5 | bleu | 0.408 | 0.283 | 0.388 | 0.388 | 0.222 | 0.954 | 0.477 | 0.549 | 0.405 | 0.596 | 0.644 | 0.550 | 0.722 | 0.167 | 0.525 | 10.080 | 14.524 | 15203 | 54724 | 17.296 |
| | | T5 | sacrebleu | 0.395 | 0.275 | 0.378 | 0.378 | 0.211 | 0.955 | 0.477 | 0.554 | 0.401 | 0.577 | 0.621 | 0.535 | 0.728 | 0.174 | 0.539 | 10.086 | 14.518 | 14846 | 52327 | 16.063 |
| | | T5 | meteor | 0.410 | 0.282 | 0.389 | 0.389 | 0.223 | 0.954 | 0.469 | 0.540 | 0.398 | 0.593 | 0.642 | 0.547 | 0.718 | 0.165 | 0.516 | 10.037 | 14.467 | 15182.0 | 54446 | 17.542 |
| | | T5 | parent | 0.401 | 0.277 | 0.382 | 0.382 | 0.215 | 0.953 | 0.470 | 0.543 | 0.394 | 0.598 | 0.647 | 0.550 | 0.732 | 0.174 | 0.545 | 10.209 | 14.660 | 15379.0 | 55421 | 16.826 |
| | | T5 | meteor + parent | 0.406 | 0.281 | 0.386 | 0.387 | 0.220 | 0.954 | 0.473 | 0.544 | 0.399 | 0.600 | 0.648 | 0.553 | 0.727 | 0.170 | 0.532 | 10.143 | 14.586 | 15330 | 55211 | 17.185 |
| | Supervised + NLPO | T5 | bleu | 0.410 | 0.283 | 0.388 | 0.388 | 0.222 | 0.954 | 0.476 | 0.548 | 0.404 | 0.597 | 0.644 | 0.552 | 0.721 | 0.167 | 0.524 | 10.077 | 14.532 | 15213 | 54948 | 17.408 |
| | | T5 | sacrebleu | 0.397 | 0.276 | 0.38 | 0.38 | 0.214 | 0.955 | 0.477 | 0.555 | 0.401 | 0.581 | 0.628 | 0.535 | 0.729 | 0.174 | 0.54 | 10.124 | 14.544 | 14940 | 52986 | 16.334 |
| | | T5 | meteor | 0.411 | 0.283 | 0.389 | 0.39 | 0.224 | 0.954 | 0.474 | 0.547 | 0.403 | 0.6 | 0.649 | 0.554 | 0.727 | 0.171 | 0.536 | 10.156 | 14.612 | 15341 | 55292 | 17.637 |
| | | T5 | parent | 0.405 | 0.28 | 0.386 | 0.386 | 0.219 | 0.954 | 0.469 | 0.541 | 0.398 | 0.598 | 0.645 | 0.552 | 0.716 | 0.165 | 0.519 | 10.019 | 14.5 | 15218 | 54793 | 17.095 |
| | | T5 | meteor + parent | 0.405 | 0.28 | 0.386 | 0.386 | 0.219 | 0.954 | 0.474 | 0.547 | 0.398 | 0.598 | 0.646 | 0.552 | 0.727 | 0.171 | 0.536 | 10.156 | 14.612 | 15341 | 55292 | 17.095 |

Table 24: **ToTTo dev evaluation**: Table shows lexical, semantic and factual correctness metric scores of algorithms with different reward functions on dev set. Without supervised pre-training, both PPO and NLPO results in sub-optimal solutions, with NLPO better than PPO. With supervised pre-training, PPO and NLPO achieve better scores across all metrics showing RL fine-tuning is beneficial. Most importantly, RL fine-tuned models produce more factually correct text as seen in higher PARENT scores. Another observation, fine-tuning with a task-specific metric PARENT is better than training just on task-agnostic lexical metrics

| Algorithm | Unique N | Coherence | | | Correctness | | |
|---|---|---|---|---|---|---|---|
| | | **Value** | **Alpha** | **Skew** | **Value** | **Alpha** | **Skew** |
| Zero Shot | 25 | 1.63 | 0.718 | 1.642 | 1.93 | 0.503 | 1.946 |
| PPO+Supervised | 24 | 4.57 | 0.221 | 4.579 | 4.48 | 0.098 | 4.483 |
| PPO | 26 | 2.75 | 0.427 | 2.753 | 3.23 | 0.214 | 3.227 |
| NLPO | 28 | 2.25 | 0.401 | 2.247 | 2.61 | 0.419 | 2.613 |
| Supervised | 24 | **4.59** | 0.173 | 4.592 | 4.54 | 0.189 | 4.537 |
| NLPO+Supervised | 26 | **4.58** | 0.244 | 4.601 | **4.57** | 0.144 | 4.581 |

Table 25: Results of the human subject study showing the number of participants N, average Likert scale value for coherence and sentiment, Krippendorf's alpha showing inter-annotator agreement, and Skew. For each model a total of 50 samples were drawn randomly from the test set and rated by 3 annotators each, resulting in 150 data points per algorithm.

| Group 1 | Group 2 | Coherence | | Correctness | |
|---|---|---|---|---|---|
| | | **Diff (G2-G1)** | **p-values** | **Diff (G2-G1)** | **p-values** |
| PPO | NLPO | **-0.507** | **0.001** | **-0.613** | **0.001** |
| PPO | NLPO+Supervised | **1.827** | **0.001** | **1.340** | **0.001** |
| PPO | Supervised | **1.833** | **0.001** | **1.313** | **0.001** |
| PPO | PPO+Supervised | **1.813** | **0.001** | **1.253** | **0.001** |
| PPO | Zero Shot | **-1.120** | **0.001** | **-1.293** | **0.001** |
| NLPO | NLPO+Supervised | **2.333** | **0.001** | **1.953** | **0.001** |
| NLPO | Supervised | **2.340** | **0.001** | **1.927** | **0.001** |
| NLPO | PPO+Supervised | **2.320** | **0.001** | **1.867** | **0.001** |
| NLPO | Zero Shot | **-0.613** | **0.001** | **-0.680** | **0.001** |
| NLPO+Supervised | Supervised | 0.007 | 0.9 | **-0.027** | **0.009** |
| NLPO+Supervised | PPO+Supervised | **-0.013** | **0.009** | **-0.087** | **0.009** |
| NLPO+Supervised | Zero Shot | **-2.947** | **0.001** | **-2.633** | **0.001** |
| Supervised | PPO+Supervised | **-0.020** | **0.009** | **-0.060** | **0.009** |
| Supervised | Zero Shot | **-2.953** | **0.001** | **-2.607** | **0.001** |
| PPO+Supervised | Zero Shot | **-2.933** | **0.001** | **-2.547** | **0.001** |

Table 26: Results of an post-hoc Tukey HSD Test for difference in means between pairs of algorithms (Group 2 - Group 1) and corresponding $p$-values. Individually statistically significant results are bolded and are used to discuss results in the analysis. Overall $p$-values showing that there is a significant difference in means between the models via a one-way ANOVA test are significant with $p \ll 0.05$ for both coherence and sentiment.

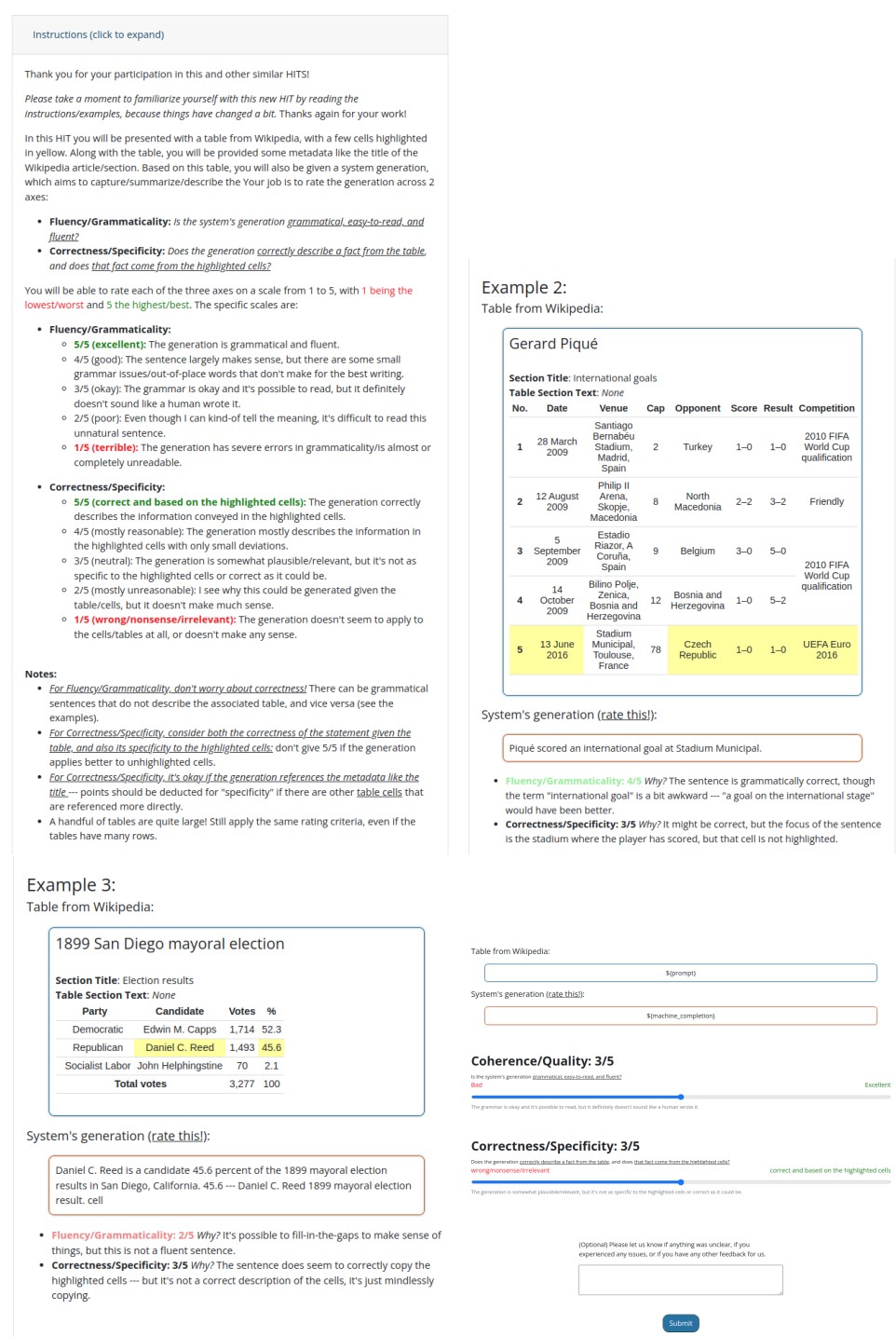

Figure 9: Instructions, two examples, and interface for the ToTTo table description task.

### B.6.3 Human Participant Study

Figure 9 shows the ToTTo instructions, example, and interface used for the human evaluation experiments. We made small modifications to the original code release's HTML renderer to make the tables display in our HITs. Tables 25, 26 show averaged results, annotator agreement, and the results of statistical significance tests to determine which models output better generations when rated by humans.

### B.6.4 QUALITATIVE ANALYSIS

We show sample generations from each of the algorithms for three randomly picked prompts below.

**Sample** 1
**Prompt:** `<table> <cell> August 1, 1992  Date   North`
`America   Australasia   Asia </`
`col_header>  North America  </cell> <cell> Allentown `
`City   North America   Australasia </`
`col_header>  Asia   North America  </`
`cell> <cell> Allentown Fairgrounds Grandstand  Country  <`
`col_header> North America   Australasia  <`
`col_header> Asia   North America  </cell> </table><`
`page_title> Under My Spell Tour </page_title> <section_title> Tour dates </section_title>`
`<table> <cell> August 1, 1992  Date   North America`
`  Australasia   Asia  <`
`col_header> North America  </cell> <cell> Allentown  City </`
`col_header>  North America   Australasia </`
`col_header>  Asia   North America  </`
`cell> <cell> Allentown Fairgrounds Grandstand  Country  <`
`col_header> North America   Australasia  <`
`col_header> Asia   North America  </cell> </table>`
**Zero-Shot:** `table> cell> Under My Spell Tour /section_title> Tour dates /cell> cell> August 1,`
`1992 col_header> Date /col_header>`
**PPO:** `Under My Spell Tour, August 1, 1992. August 1, 1992.`
**NLPO:** `Under My Spell Tour dates Under My Spell Tour dates Allentown Fairgrounds Grandstand`
`August 1, 1992. Under My Spell Tour dates Under My Spell Tour dates Allentown Fairgrounds`
`Grandstand August 1, 1992 table>`
**Supervised:** `The Under My Spell Tour began on August 1, 1992 at the Allentown Fairgrounds`
`Grandstand in Allentown.`
**Supervised + PPO:** `On August 1, 1992, Under My Spell Tour was held at the Allentown Fairgrounds`
`Grandstand in Allentown.`
**Supervised + NLPO:** `On August 1, 1992, Under My Spell Tour was held at the Allentown`
`Fairgrounds Grandstand in Allentown.`

**Sample** 2
**Prompt:** `<table> <cell> Philip IV the Fair, the Iron King (Philippe)  Name </`
`col_header> </cell> <cell> 5 October 1285  King from  </cell> <`
`cell> King of France and of Navarre (Roi de France et de Navarre)  Title </`
`col_header> </cell> </table><page_title> List of French monarchs </page_title> <`
`section_title> House of Capet (987â\u0102\u015f1328) </section_title> <table> <cell> Philip IV the`
`Fair, the Iron King (Philippe)  Name  </cell> <cell> 5 October`
`1285  King from  </cell> <cell> King of France and of Navarre (`
`Roi de France et de Navarre)  Title  </cell> </table>`
**Zero-Shot:** `> </table><page_title> List of French monarchs <`
**PPO:** `Philip IV the Fair, the Iron King (Philippe) 5 October 1285. Title: King of France and of`
`Navarre (Roi de France et de Navarre`
**NLPO:** `King (Philippe) Philip IV the Fair, the Iron King (Philippe) 5 October 1285`
**Supervised:** `Philip IV the Fair, the Iron King (Philippe) was King of France and Navarre from`
`1285.`
**Supervised + PPO:** `Philip IV the Fair, the Iron King was King of France and Navarre from 1285.`
**Supervised + NLPO:** `Philip IV the Fair, the Iron King (Philippe) was King of France and Navarre`
`(Roi de France et de Navarre) in 1285.`

**Sample** 3
**Prompt:** `<table> <cell> Benjamin A. Smith II (D)  Vacator  </cell> <`
`cell> Successor elected November 6, 1962.  Reason for change  </`
`cell> <cell> Ted Kennedy (D)  Successor  </cell> <cell> November`
`7, 1962  Date of successor's formal installation  </cell> </`
`table><page_title> 87th United States Congress </page_title> <section_title> Senate </`
`section_title> <table> <cell> Benjamin A. Smith II (D)  Vacator `
`</cell> <cell> Successor elected November 6, 1962.  Reason for change </`
`col_header> </cell> <cell> Ted Kennedy (D)  Successor  </cell> <`
`cell> November 7, 1962  Date of successor's formal installation `
`</cell> </table>`
**Zero-Shot:** `cell> Ted Kennedy (D) col_header> Vacator /col_header> /cell> cell> November 7,`
`1962 col_header> Reason for change`
**PPO:** `Ted Kennedy (D) succeeding November 6, 1962 as Vacator of the 87th United States Congress.`

**NLPO:** `Ted Kennedy (D) Successor elected November 6, 1962. table> cell Benjamin A. Smith II (D)`
`col_header Reason for change: Date of successor's formal installation November 7, 1962.`
**Supervised:** `Benjamin A. Smith II (D) served as senate until November 6, 1962 which was later`
`served by Ted Kennedy (D) from November 7, 1962.`
**Supervised + PPO:** `Benjamin A. Smith II (D) served until November 6, 1962 and Ted Kennedy (D)`
`succeeded him from November 7, 1962.`
**Supervised + NLPO:** `Benjamin A. Smith II (D) served until November 6, 1962 and Ted Kennedy (D)`
`succeeded him from November 7, 1962.`

## B.7 NARRATIVE QA

### B.7.1 SETUP

NarrativeQA (Kočiskỳ et al., 2018) deals with task of generating answers to questions about a given story. For training RL methods, we consider 2 traditional lexical rewards namely Rouge Combined and Rouge-L-Max. We chose T5-base as the base LM since it has been shown to do well at question answering in prior work (Khashabi et al., 2020). We note that the supervised models we use are trained on the UnifiedQA dataset, which contains other QA datasets, and is shown by Khashabi et al. (2020) to outperform supervised fine-tuning only on NarrativeQA. Hyperparams for our models can be found in Table 27.

| Model Params | value |
|---|---|
| ppo/nlpo | steps per update: 5120
total number of steps: 512000
batch size: 64
epochs per update: 5
learning rate: 0.000002
entropy coefficient: 0.0
initial kl coeff: 0.001
target kl: 1.0
discount factor: 0.99
gae lambda: 0.95
clip ratio: 0.2
rollouts top k : 50
value function coeff: 0.5
top mask ratio: 0.9
target update iterations: 20 |
| supervised+ ppo (or nlpo) | steps per update:2560
total number of steps: 512000
batch size: 64
epochs per update: 5
learning rate: 0.0000005
entropy coefficient: 0.0
initial kl coeff: 0.001
target kl: 0.2
discount factor: 0.99
gae lambda: 0.95
clip ratio: 0.2
rollouts top k : 50
value function coeff: 0.5
top mask ratio: 0.9
target update iterations: 20 |
| decoding | num beams: 4
max new tokens: 50 |
| tokenizer | padding side: left
truncation side: right
max length: 512 |

Table 27: **NarQA Hyperparams**: Table shows a list of all hyper-parameters and their settings

| Tasks | Alg | Reward Function | LM | Lexical and Semantic Metrics | | | | | | | | Diversity Metrics | | | | | | | |
|---|---|---|---|---|---|---|---|---|---|---|---|---|---|---|---|---|---|---|---|
| | | | | Rouge-1 | Rouge-2 | Rouge-L | Rouge-LSum | Rouge-LMax | Meteor | BLEU | BertScore | MSTTR | Distinct$_1$ | Distinct$_2$ | H$_1$ | H$_2$ | Unique$_1$ | Unique$_2$ | Mean Output Length |
| NarQA | Zero Shot | | T5 | 0.095 | 0.022 | 0.084 | 0.084 | 0.117 | 0.095 | 0.009 | 0.835 | 0.415 | 0.026 | 0.097 | 9.641 | 13.468 | 1880 | 11495 | 31.688 |
| | PPO | Rouge Combined | T5 | 0.101 | 0.025 | 0.088 | 0.088 | 0.122 | 0.099 | 0.01 | 0.837 | 0.462 | 0.03 | 0.125 | 9.759 | 13.789 | 2522 | 17806 | 32.352 |
| | | Rouge-L Max | T5 | 0.099 | 0.025 | 0.087 | 0.087 | 0.122 | 0.099 | 0.01 | 0.835 | 0.439 | 0.029 | 0.119 | 9.653 | 13.618 | 2292 | 15816 | 31.479 |
| | NLPO | Rouge Combined | T5 | 0.097 | 0.023 | 0.085 | 0.085 | 0.118 | 0.098 | 0.009 | 0.836 | 0.418 | 0.025 | 0.096 | 9.652 | 13.528 | 1816 | 10980 | 32.117 |
| | | Rouge-L Max | T5 | 0.102 | 0.026 | 0.089 | 0.089 | 0.124 | 0.1 | 0.01 | 0.837 | 0.445 | 0.029 | 0.118 | 9.776 | 13.75 | 2181 | 14569 | 31.555 |
| | Supervised | | T5 | 0.378 | 0.190 | 0.367 | 0.367 | 0.581 | 0.099 | 0.209 | 0.931 | 0.609 | 0.156 | 0.534 | 9.807 | 13.657 | 3250 | 14995 | 4.923 |
| | Supervised + PPO | Rouge Combined | T5 | 0.38 | 0.177 | 0.371 | 0.371 | 0.585 | 0.09 | 0.229 | 0.931 | 0.64 | 0.174 | 0.559 | 10.132 | 13.547 | 3326 | 13785 | 4.353 |
| | | Rouge-L Max | T5 | 0.368 | 0.18 | 0.36 | 0.36 | 0.585 | 0.083 | 0.239 | 0.931 | 0.641 | 0.187 | 0.576 | 10.201 | 13.452 | 3287 | 12436 | 3.913 |
| | Supervised + NLPO | Rouge Combined | T5 | 0.398 | 0.21 | 0.393 | 0.373 | 0.589 | 0.096 | 0.24 | 0.971 | 0.679 | 0.185 | 0.595 | 10.304 | 13.694 | 3371 | 15067 | 4.728 |
| | | Rouge-L Max | T5 | 0.381 | 0.194 | 0.383 | 0.383 | 0.588 | 0.093 | 0.243 | 0.932 | 0.645 | 0.187 | 0.59 | 10.2 | 13.397 | 3287 | 12171 | 3.889 |

Table 28: **Evaluation of NarrativeQA**: Reference Metrics, supervised is based on UnifiedQA (Khashabi et al., 2020).

### B.7.2 RESULTS AND DISCUSSION

Table 28 presents our benchmarking results with 2 reward functions along with supervised baseline performances on the NarrativeQA test set. Similar to other methods, our main finding is that warm-started initial policies are crucial for learning to generate answers that successfully use the input context.

### B.7.3 QUALITATIVE RESULTS

We show sample generations from each of the algorithms for three randomly picked prompts below.

```
Sample 1
Prompt: who is mark hunter? mark hunter (slater), a high school student in a sleepy suburb of
    phoenix, arizona, starts an fm pirate radio station that broadcasts from the basement of
    his parents' house. mark is a loner, an outsider, whose only outlet for his teenage angst
     and aggression is his unauthorized radio station. his pirate station's theme song is "
    everybody knows" by leonard cohen and there are glimpses of cassettes by such alternative
     musicians as the jesus and mary chain, camper van beethoven, primal scream, soundgarden,
     ice-t, bad brains, concrete blonde, henry rollins, and the pixies. by day, mark is seen
    as a loner, hardly talking to anyone around him; by night, he expresses his outsider
    views about what is wrong with american society. when he speaks his mind about what is
    going on at his school and in the community, more and more of his fellow students tune in
     to hear his show.nobody knows the true identity of "hard harry" or "happy harry hard-on
    ," as mark refers to himself, until nora diniro (mathis), a fellow student, tracks him
    down and confronts him the day after a student named malcolm commits suicide after harry
    attempts to reason with him. the radio show becomes increasingly popular and influential
    after harry confronts the suicide head-on, exhorting his listeners to do something about
    their problems instead of surrendering to them through suicideÃ¢ÂÂÂÏat the crescendo of
    his yelled speech, an overachieving student named paige woodward (who has been a constant
     listener) jams her various medals and accolades into a microwave and turns it on. she
    then sits, watching the awards cook until the microwave explodes, injuring her. while
    this is happening, other students act out in cathartic release.eventually, the radio show
     causes so much trouble in the community that the fcc is called in to investigate. during
     the fracas, it is revealed that the school's principal (annie ross) has been expelling "
    problem students," namely, students with below-average standardized test scores, in an
    effort to boost the district's test scores while still keeping their names on the rolls (
    a criminal offense) in order to retain government funding.realizing he has started
    something huge, mark decides it is up to him to end it. he dismantles his radio station
    and attaches it to his mother's old jeep, creating a mobile transmitter so his position
    can't be triangulated. pursued by the police and the fcc, nora drives the jeep around
    while mark broadcasts. the harmonizer he uses to disguise his voice breaks, and with no
    time left to fix it, mark decides to broadcast his final message as himself. they finally
     drive up to the crowd of protesting students, and mark tells them that the world belongs
     to them and that they should make their own future. the police step in and arrest mark
    and nora. as they are taken away, mark reminds the students to "talk hard." as the film
    ends, the voices of other students (and even one of the teachers) speak as intros for
    their own independent stations, which can be heard broadcasting across the country.
Zero-Shot: hunter (slater) is a high school student in a sleepy suburb of phoenix, arizona. he
    starts an fm pirate radio station that broadcasts from the basement of his
References: he is a high school student in phoenix 2) loner and outsider student with a radio
    station
PPO: hunter (slater), a high school student in a sleepy suburb of phoenix, arizona, starts an
    fm pirate radio station that broadcasts from the basement of his parents' house
NLPO: hunter (slater), a high school student in a sleepy suburb of phoenix, arizona, starts an
     fm pirate radio station that broadcasts from the basement of his parents' house
Supervised: a high school student
Supervised + PPO: a high school student
Supervised + NLPO: a high school student

Sample 2
Prompt: what was the principle doing with the problem students? mark hunter (slater), a high
    school student in a sleepy suburb of phoenix, arizona, starts an fm pirate radio station
    that broadcasts from the basement of his parents' house. mark is a loner, an outsider,
    whose only outlet for his teenage angst and aggression is his unauthorized radio station.
     his pirate station's theme song is "everybody knows" by leonard cohen and there are
    glimpses of cassettes by such alternative musicians as the jesus and mary chain, camper
    van beethoven, primal scream, soundgarden, ice-t, bad brains, concrete blonde, henry
    rollins, and the pixies. by day, mark is seen as a loner, hardly talking to anyone around
     him; by night, he expresses his outsider views about what is wrong with american society.
     when he speaks his mind about what is going on at his school and in the community, more
    and more of his fellow students tune in to hear his show.nobody knows the true identity
    of "hard harry" or "happy harry hard-on," as mark refers to himself, until nora diniro (
    mathis), a fellow student, tracks him down and confronts him the day after a student
    named malcolm commits suicide after harry attempts to reason with him. the radio show
    becomes increasingly popular and influential after harry confronts the suicide head-on,
    exhorting his listeners to do something about their problems instead of surrendering to
    them through suicideÃÂÂÏat the crescendo of his yelled speech, an overachieving student
    named paige woodward (who has been a constant listener) jams her various medals and
```

accolades into a microwave and turns it on. she then sits, watching the awards cook until the microwave explodes, injuring her. while this is happening, other students act out in cathartic release.eventually, the radio show causes so much trouble in the community that the fcc is called in to investigate. during the fracas, it is revealed that the school's principal (annie ross) has been expelling "problem students," namely, students with below-average standardized test scores, in an effort to boost the district's test scores while still keeping their names on the rolls (a criminal offense) in order to retain government funding.realizing he has started something huge, mark decides it is up to him to end it. he dismantles his radio station and attaches it to his mother's old jeep, creating a mobile transmitter so his position can't be triangulated. pursued by the police and the fcc, nora drives the jeep around while mark broadcasts. the harmonizer he uses to disguise his voice breaks, and with no time left to fix it, mark decides to broadcast his final message as himself. they finally drive up to the crowd of protesting students, and mark tells them that the world belongs to them and that they should make their own future. the police step in and arrest mark and nora. as they are taken away, mark reminds the students to "talk hard." as the film ends, the voices of other students (and even one of the teachers) speak as intros for their own independent stations, which can be heard broadcasting across the country.

**Zero-Shot:** mark hunter (slater) starts an fm pirate radio station that broadcasts from the basement of his parents' house. by day, mark is seen as a loner, an outsider, whose only outlet

**References:** 1) expelling them. 2) expelling the students

**PPO:** mark hunter (slater) starts an fm pirate radio station that broadcasts from the basement of his parents' house. mark is a loner, an outsider, whose only outlet for his teenage ang

**NLPO:** mark hunter (slater) starts an fm pirate radio station that broadcasts from the basement of his parents' house. mark is a loner, an outsider, whose only outlet for his teenage ang

**Supervised:** he was taking them to a psychiatric hospital.

**Supervised** + **PPO:** addressing them

**Supervised** + **NLPO:** he was expelling them

**Sample** 3

**Prompt:** what happens to the people maskull meets in his travels to various lands? maskull, a man longing for adventures, accepts an invitation from krag, an acquaintance of his friend nightspore, to travel to tormance after a seance. the three set off in a crystal ship from an abandoned observatory in scotland but maskull awakens to find himself alone on tormance. in every land he passes through he usually meets only one or two persons; these meetings often (though not always) end in the death of those he meets, either at his own hand or by that of another. he learns of his own impending death, meets krag again, and dies shortly after learning that he is in fact nightspore himself. the book concludes with a final revelation from krag (who claims to be known on earth as "pain") to nightspore about the origin of the universe. the author turns out to support a variation of the doctrine of the demiurge, somewhat similar to that propounded by some gnostics.all of the characters and lands are types used to convey the author's critique of several philosophical systems. on tormance, most such viewpoints or ways of life are accompanied by corresponding new bodily sense organs or modifications of the same, thus each distinct weltanschauung landscape has its corresponding sensorium.

**Zero-Shot:** what happens to the people maskull meets in his travels to various lands? n maskull, a man longing for adventures, accepts an invitation from krag to travel to tormance after

**References:** 1) they usually die by his hand or anothers. 2) they die

**PPO:** what happens to the people maskull meets in his travels to various lands? n maskull, a man longing for adventures, accepts an invitation from krag to travel to tormance after

**NLPO:** maskull meets krag, an acquaintance of nightspore, to travel to tormance after a seance. maskull awakens to find himself alone on tormance. in every land

**Supervised:** they die either from his own hand or from another person's death.

**Supervised** + **PPO:** they end up dying either at his own hand or by another's.

**Supervised** + **NLPO:** they end up dying either at his own hand or by another's

## B.8 MACHINE TRANSLATION

### B.8.1 SETUP

WMT-16 We pick two languages, English and German, and frame this task similarly to other machine translation tasks—requiring the models to translate from English to German. We train models on 4 rewards: SacreBLEU, chRF, TER, and BertScore.

| Model Params | value |
|---|---|
| supervised | batch size: 64
epochs: 5
learning rate: 0.00001
learning rate scheduler: constant
weight decay: 0.1 |
| ppo/nlpo | steps per update: 5120
total number of steps: 256000
batch size: 64
epochs per update: 5
learning rate: 0.0.000001
entropy coefficient: 0.0
initial kl coeff: 0.001
target kl: 0.2
discount factor: 0.99
gae lambda: 0.95
clip ratio: 0.2
rollouts top k : 10
value function coeff: 0.5
top mask ratio: 0.5
target update iterations: 20 |
| supervised+ ppo (or nlpo) | steps per update:2560
total number of steps: 256000
batch size: 64
epochs per update: 5
learning rate: 0.0000005
entropy coefficient: 0.0
initial kl coeff: 0.001
target kl: 0.2
discount factor: 0.99
gae lambda: 0.95
clip ratio: 0.2
rollouts top k : 10
value function coeff: 0.5
top mask ratio: 0.5
target update iterations: 20 |
| decoding | num beams: 4
length penalty: 0.6
max new tokens: 128 |
| tokenizer | padding side: left
truncation side: right
max length: 128 |

Table 29: **NMT Hyperparams**: Table shows a list of all hyper-parameters and their settings

### B.8.2 RESULTS AND DISCUSSION

Tables 30, 31 presents our benchmarking results with 4 reward functions along with supervised baseline performances on test set. Our main finding is that NLPO + Supervised performs better than PPO and supervised models.

| Datasets | Alg | LM | Reward Function | Lexical and Semantic Metrics | | | | | | | | | |
|---|---|---|---|---|---|---|---|---|---|---|---|---|---|
| | | | | Rouge-1 | Rouge-2 | Rouge-L | Rouge-LSum | Meteor | BLEU | SacreBLEU | chRf | TER | BertScore |
| WMT16 | Zero-Shot | T5 | | 0.635 | 0.414 | 0.591 | 0.591 | 0.483 | 0.294 | 0.348 | 0.613 | 0.543 | 0.882 |
| | PPO | T5 | SacreBLEU | 0.636 | 0.415 | 0.591 | 0.591 | 0.482 | 0.294 | 0.348 | 0.614 | 0.539 | 0.882 |
| | | T5 | chRF | 0.635 | 0.414 | 0.591 | 0.591 | 0.481 | 0.291 | 0.346 | 0.612 | 0.540 | 0.882 |
| | | T5 | TER | 0.638 | 0.416 | 0.595 | 0.594 | 0.484 | 0.294 | 0.350 | 0.616 | 0.534 | 0.883 |
| | | T5 | BertScore | 0.637 | 0.417 | 0.593 | 0.593 | 0.479 | 0.294 | 0.347 | 0.613 | 0.534 | 0.882 |
| | NLPO | T5 | SacreBLEU | 0.635 | 0.415 | 0.592 | 0.592 | 0.484 | 0.297 | 0.352 | 0.615 | 0.542 | 0.882 |
| | | T5 | chRF | 0.634 | 0.413 | 0.59 | 0.59 | 0.481 | 0.291 | 0.345 | 0.612 | 0.540 | 0.882 |
| | | T5 | TER | 0.633 | 0.412 | 0.59 | 0.59 | 0.477 | 0.286 | 0.341 | 0.608 | 0.540 | 0.881 |
| | | T5 | BertScore | 0.622 | 0.397 | 0.58 | 0.581 | 0.458 | 0.269 | 0.323 | 0.591 | 0.546 | 0.876 |
| | Supervised | T5 | | 0.635 | 0.411 | 0.590 | 0.590 | 0.482 | 0.294 | 0.350 | 0.617 | 0.540 | 0.882 |
| | Supervised + PPO | T5 | SacreBLEU | 0.640 | 0.416 | 0.595 | 0.595 | 0.487 | **0.298** | **0.355** | 0.620 | 0.533 | 0.883 |
| | | T5 | chRF | 0.640 | 0.416 | 0.596 | 0.596 | 0.486 | 0.298 | 0.354 | **0.621** | 0.532 | 0.883 |
| | | T5 | TER | 0.637 | 0.414 | 0.594 | 0.594 | 0.483 | 0.295 | 0.352 | 0.618 | 0.533 | 0.882 |
| | | T5 | BertScore | 0.637 | 0.413 | 0.593 | 0.594 | 0.482 | 0.294 | 0.350 | 0.616 | 0.533 | 0.882 |
| | Supervised + NLPO | T5 | SacreBLEU | **0.642** | **0.419** | 0.596 | 0.596 | 0.497 | 0.297 | **0.355** | **0.621** | 0.533 | **0.888** |
| | | T5 | chRF | 0.636 | 0.412 | 0.592 | 0.592 | 0.492 | 0.293 | 0.349 | 0.617 | 0.534 | 0.886 |
| | | T5 | TER | 0.637 | 0.414 | 0.594 | 0.594 | 0.491 | 0.292 | 0.349 | 0.615 | **0.531** | 0.886 |
| | | T5 | BertScore | 0.64 | 0.417 | **0.598** | **0.598** | **0.499** | 0.287 | 0.349 | 0.62 | 0.538 | 0.887 |
| IWSLT2017 | Zero-Shot | T5 | | 0.619 | 0.386 | 0.588 | 0.587 | 0.445 | 0.254 | 0.308 | 0.577 | 0.573 | 0.870 |
| | PPO | T5 | SacreBLEU | 0.621 | 0.383 | 0.587 | 0.587 | 0.448 | 0.243 | 0.296 | 0.575 | 0.583 | 0.869 |
| | | T5 | chRF | 0.622 | 0.385 | 0.590 | 0.590 | 0.448 | 0.248 | 0.301 | 0.578 | 0.575 | 0.870 |
| | | T5 | TER | 0.623 | 0.384 | 0.591 | 0.591 | 0.443 | 0.246 | 0.303 | 0.572 | 0.568 | 0.869 |
| | | T5 | BertScore | 0.533 | 0.326 | 0.507 | 0.507 | 0.321 | 0.143 | 0.174 | 0.406 | 0.573 | 0.839 |
| | NLPO | T5 | SacreBLEU | 0.624 | 0.385 | 0.59 | 0.59 | 0.45 | 0.245 | 0.299 | 0.578 | 0.578 | 0.87 |
| | | T5 | chRF | 0.624 | 0.386 | 0.59 | 0.59 | 0.451 | 0.248 | 0.302 | 0.581 | 0.576 | 0.87 |
| | | T5 | TER | 0.622 | 0.384 | 0.59 | 0.59 | 0.443 | 0.246 | 0.303 | 0.573 | 0.57 | 0.869 |
| | | T5 | BertScore | 0.611 | 0.377 | 0.58 | 0.58 | 0.425 | 0.239 | 0.291 | 0.555 | 0.573 | 0.866 |
| | Supervised | T5 | | 0.638 | 0.400 | 0.610 | 0.609 | 0.461 | 0.280 | 0.337 | 0.593 | 0.538 | **0.878** |
| | Supervised + PPO | T5 | SacreBLEU | 0.640 | 0.407 | 0.610 | 0.610 | 0.465 | 0.277 | 0.332 | 0.596 | 0.542 | 0.877 |
| | | T5 | chRF | 0.639 | 0.406 | 0.609 | 0.609 | 0.464 | 0.277 | 0.331 | 0.596 | 0.543 | 0.877 |
| | | T5 | TER | 0.637 | 0.406 | 0.609 | 0.609 | 0.457 | 0.274 | 0.331 | 0.589 | 0.535 | 0.876 |
| | | T5 | BertScore | 0.612 | 0.381 | 0.585 | 0.585 | 0.418 | 0.240 | 0.291 | 0.548 | 0.559 | 0.867 |
| | Supervised + NLPO | T5 | SacreBLEU | **0.641** | 0.418 | 0.614 | 0.614 | **0.474** | 0.289 | 0.343 | **0.597** | 0.535 | 0.877 |
| | | T5 | chRF | 0.643 | 0.418 | 0.621 | 0.621 | 0.464 | **0.291** | 0.345 | 0.596 | 0.539 | 0.877 |
| | | T5 | TER | 0.639 | **0.419** | **0.621** | **0.621** | 0.471 | 0.289 | **0.346** | 0.593 | **0.535** | 0.877 |
| | | T5 | BertScore | 0.633 | 0.401 | 0.606 | 0.606 | 0.448 | 0.267 | 0.323 | 0.580 | 0.537 | 0.875 |

Table 30: **WMT-16 and IWSLT test evaluation - lexical and semantic**: Table shows lexical, semantic metrics for RL algorithms with different reward functions bench-marked against supervised baseline models

| Tasks | | | | **Diversity Metrics** | | | | | | | |
|---|---|---|---|---|---|---|---|---|---|---|---|
| | Alg | Reward Function | LM | MSTTR | Distinct$_1$ | Distinct$_2$ | H$_1$ | H$_2$ | Unique$_1$ | Unique$_2$ | Mean Output Length |
| WMT16 | Zero-Shot | T5 | | 0.732 | 0.193 | 0.675 | 10.100 | 14.561 | 7290 | 33691 | 20.533 |
| | PPO | T5 | SacreBLEU | 0.738 | 0.198 | 0.687 | 10.166 | 14.613 | 7503 | 34140 | 20.375 |
| | | T5 | chRF | 0.738 | 0.196 | 0.687 | 10.175 | 14.611 | 7376 | 34116 | 20.337 |
| | | T5 | TER | 0.736 | 0.196 | 0.683 | 10.132 | 14.588 | 7447 | 33977 | 20.356 |
| | | T5 | BertScore | 0.736 | 0.195 | 0.685 | 10.129 | 14.574 | 7272 | 33477 | 20.035 |
| | NLPO | T5 | SacreBLEU | 0.735 | 0.193 | 0.68 | 10.125 | 14.592 | 7395 | 34276 | 20.672 |
| | | T5 | chRF | 0.738 | 0.196 | 0.686 | 10.164 | 14.606 | 7399 | 34056 | 20.351 |
| | | T5 | TER | 0.74 | 0.2 | 0.694 | 10.204 | 14.63 | 7522 | 34234 | 20.151 |
| | | T5 | BertScore | 0.739 | 0.2 | 0.698 | 10.194 | 14.608 | 7203 | 33169 | 19.482 |
| | Supervised | T5 | | 0.729 | 0.190 | 0.669 | 10.048 | 14.530 | 7205 | 33430 | 20.622 |
| | Supervised + PPO | T5 | SacreBLEU | 0.732 | 0.191 | 0.674 | 10.080 | 14.552 | 7222 | 33723 | 20.605 |
| | | T5 | chRF | 0.735 | 0.192 | 0.677 | 10.093 | 14.569 | 7319 | 33923 | 20.586 |
| | | T5 | TER | 0.732 | 0.192 | 0.676 | 10.079 | 14.553 | 7265 | 33635 | 20.441 |
| | | T5 | BertScore | 0.732 | 0.192 | 0.677 | 10.082 | 14.550 | 7187 | 33385 | 20.305 |
| | Supervised + NLPO | T5 | SacreBLEU | 0.734 | 0.191 | 0.675 | 10.089 | 14.568 | 7308 | 33941 | 20.686 |
| | | T5 | chRF | 0.735 | 0.194 | 0.681 | 10.112 | 14.571 | 7372 | 33814 | 20.348 |
| | | T5 | TER | 0.737 | 0.194 | 0.682 | 10.105 | 14.566 | 7243 | 33482 | 20.159 |
| | | T5 | BertScore | 0.737 | 0.227 | 0.742 | 10.042 | 14.179 | 5438 | 22574 | 12.63 |
| IWSLT2017 | Zero-Shot | T5 | | 0.662 | 0.097 | 0.4700 | 9.276 | 14.526 | 8312 | 52947 | 18.739 |
| | PPO | T5 | SacreBLEU | 0.657 | 0.095 | 0.464 | 9.230 | 14.498 | 8285 | 53000 | 19.069 |
| | | T5 | chRF | 0.660 | 0.096 | 0.468 | 9.253 | 14.526 | 8243 | 53142 | 18.912 |
| | | T5 | TER | 0.659 | 0.097 | 0.474 | 9.244 | 14.536 | 8129 | 51914 | 18.268 |
| | | T5 | BertScore | 0.673 | 0.120 | 0.541 | 9.288 | 14.388 | 6642 | 37267 | 11.602 |
| | NLPO | T5 | SacreBLEU | 0.656 | 0.094 | 0.463 | 9.207 | 14.483 | 8240 | 52822 | 19.043 |
| | | T5 | chRF | 0.658 | 0.095 | 0.464 | 9.233 | 14.502 | 8230 | 53167 | 19.073 |
| | | T5 | TER | 0.661 | 0.098 | 0.476 | 9.271 | 14.552 | 8223 | 52438 | 18.344 |
| | | T5 | BertScore | 0.667 | 0.102 | 0.491 | 9.31 | 14.576 | 8134 | 50740 | 17.162 |
| | Supervised | T5 | | 0.655 | 0.095 | 0.467 | 9.210 | 14.492 | 7970 | 51430 | 18.440 |
| | Supervised + PPO | T5 | SacreBLEU | 0.654 | 0.094 | 0.461 | 9.176 | 14.467 | 8061 | 51840 | 18.803 |
| | | T5 | chRF | 0.656 | 0.094 | 0.464 | 9.202 | 14.497 | 8054 | 52198 | 18.794 |
| | | T5 | TER | 0.658 | 0.097 | 0.475 | 9.239 | 14.529 | 7969 | 51255 | 18.048 |
| | | T5 | BertScore | 0.665 | 0.102 | 0.495 | 9.270 | 14.524 | 7495 | 47629 | 16.051 |
| | Supervised + NLPO | T5 | SacreBLEU | 0.655 | 0.094 | 0.461 | 9.182 | 14.469 | 8073 | 51797 | 18.777 |
| | | T5 | chRF | 0.657 | 0.095 | 0.466 | 9.212 | 14.503 | 8074 | 52056 | 18.642 |
| | | T5 | TER | 0.658 | 0.096 | 0.473 | 9.231 | 14.521 | 8025 | 51389 | 18.133 |
| | | T5 | BertScore | 0.661 | 0.098 | 0.482 | 9.26 | 14.547 | 7852 | 50672 | 17.496 |

Table 31: **WMT-16 and IWSLT test evaluation - diversity metrics** Table shows diversity metrics for RL algorithms with different reward functions bench-marked against supervised baseline models

### B.8.3 QUALITATIVE RESULTS

We show sample generations from each of the algorithms for three randomly picked prompts from IWSLT below.

**Sample** 1
**Prompt:** translate English to German: Several years ago here at TED, Peter Skillman introduced a design challenge called the marshmallow challenge.
**Zero-Shot:** Vor einigen Jahren hat Peter Skillman hier bei TED eine Design-Herausforderung namens Marshmallow Challenge vorgestellt.
**PPO:** Vor einigen Jahren, hier bei TED, stellte Peter Skillman einen Design-Wettbewerb namens " Die Marshmallow-Herausforderung" vor
**NLPO:** Vor einigen Jahren hier bei TED fÃijhrte Peter Skillman eine Design-Herausforderung namens Marshmallow Challenge ein.
**Supervised:** Vor einigen Jahren hier bei TED fÃijfchrte Peter Skillman eine Design-Herausforderung namens Marshmallow Challenge ein.
**Supervised + PPO:** Vor einigen Jahren hier bei TED, fÃijhrte Peter Skillman eine Design-Herausforderung namens Marshmallow-Herausforderung ein.
**Supervised + NLPO:** Vor einigen Jahren hier bei TED fÃijchrte Peter Skillman eine Design-Herausforderung ein, die Marshmallow-Herausforderung.

**Sample** 2
**Prompt:** translate English to German: They spend the majority of their time assembling the sticks into ever-growing structures.
**Zero-Shot:** Sie verbringen den grÃűÃ§ten Teil ihrer Zeit damit, die StÃűcke in immer wachsende Strukturen zu montieren.
**References:** Sie investieren einen GroÃ§teil ihrer Zeit in den Zusammenbau stÃd'ndig wachsender Strukturen
**PPO:** Sie verbringen den GroÃ§teil ihrer Zeit damit, die StÃűcke in immer grÃűÃ§er werdende Strukturen zu montieren.
**NLPO:** Sie verbringen den GroÃ§teil ihrer Zeit, die StÃűcke in immer wachsende Strukturen zu montieren.
**Supervised:** Sie verbringen den GroÃ§teil ihrer Zeit damit, die StÃűcke in immer wachsende Strukturen zu montieren.
**Supervised + PPO:** Sie verbringen den GroÃ§teil ihrer Zeit damit, die StÃűcke in immer wachsende Strukturen zu montieren.
**Supervised + NLPO:** Sie verbringen den GroÃ§teil ihrer Zeit damit, die StÃűcke in immer wachsende Strukturen zu montieren.

**Sample** 3
**Prompt:** translate English to German: And of course there are teams that have a lot more "ta-da " structures, and among the best are recent graduates of kindergarten.
**Zero-Shot:** Und natÃijrlich gibt es Teams, die viel mehr "ta-da" Strukturen haben, und zu den besten gehÃűren jÃijngste Absolventen des Kindergartens.
**References:** Und natÃijrlich gibt es auch Teams die viel mehr "ta-da"-Gebilde haben und unter den Besten sind frische Absolventen des Kindergartens.
**PPO:** Und natÃijrlich gibt es Teams, die viel mehr "ta-da" Strukturen, und unter den besten sind jÃijngste Absolventen des Kindergartens.
**NLPO:** Und natÃijrlich gibt es Teams, die viel mehr "t
a-da" Strukturen haben, und unter den besten sind jÃijngste Absolventen des Kindergartens
**Supervised:** Und natÃijrlich gibt es Teams, die viel mehr "ta-da"-Strukturen haben, und unter den besten sind jÃijngste Absolventen des Kindergartens.
**Supervised + PPO:** Und natÃijrlich gibt es Teams, die viel mehr "ta-da"-Strukturen haben, und unter den besten sind jÃijngste Absolventen des Kindergartens.
**Supervised + NLPO:** Und natÃijlich gibt es Teams, die viel mehr "ta-da"-Strukturen haben, und unter den besten sind jÃijngste Absolventen des Kindergartens.

## B.9 DAILY DIALOG

### B.9.1 SETUP

We consider DailyDialog (Li et al., 2017) as the test bed for the dialogue generation task. The dataset includes conversations written by human on various topics. In addition, each utterance contains labels of intent and emotional information. For simplicity, we focus only on generating the next utterance, given the dialogue context. We chose a context window of size 5, which results in 35k training, 3k and 3k utterances. The input to the model is dialogue history in which utterances are concatenated using a <EOU> token. We picked GPT-2 as the LM as they are more suited for text continuation than encoder-decoder LMs. For a fair comparison, we use top-k sampling with $k = 20$ as the decoding method for all methods. For RL methods, we use a linear combination of meteor score and intent match score (whether the generated text's intent matches with the reference's intent) as the reward function. The coefficients for meteor and intent are chosen based on both lexical scores and intent accuracy on the validation set. For this purpose, we trained an intent classifier (fine-tuned RoBERTa (Liu et al., 2019)) that classifies given text into intent categories such as *inform*, *question*, *directive* and *commisive*, etc. Table 32 provides a summary of hyperparameters and implementation details.

| Model Params | value |
|---|---|
| ppo/nlpo | steps per update: 1280 |
| | total number of steps: 128000 |
| | batch size: 64 |
| | epochs per update: 5 |
| | learning rate: 0.000001 |
| | entropy coefficient: 0.0 |
| | initial kl coeff: 0.2 |
| | target kl: 0.5 |
| | discount factor: 0.99 |
| | gae lambda: 0.95 |
| | clip ratio: 0.2 |
| | rollouts top k : 20 |
| | value function coeff: 0.5 |
| | meteor coeff: 0.25 |
| | intent coeff: 0.75 |
| | top mask ratio: 0.9 |
| | target update iterations: 20 |
| supervised+ ppo (or nlpo) | steps per update:1280 |
| | total number of steps: 64000 |
| | batch size: 64 |
| | epochs per update: 5 |
| | learning rate: 0.000001 |
| | entropy coefficient: 0.0 |
| | initial kl coeff: 0.2 |
| | target kl: 0.5 |
| | discount factor: 0.99 |
| | gae lambda: 0.95 |
| | clip ratio: 0.2 |
| | rollouts top k : 20 |
| | value function coeff: 0.5 |
| | meteor coeff: 0.5 0.25 |
| | intent coeff: 0.5 0.75 |
| | top mask ratio: 0.9 |
| | target update iterations: 20 |
| decoding | top k: 20 |
| | min length: 2 |
| | max new tokens: 50 |
| tokenizer | padding side: left |
| | truncation side: right |
| | max length: 128 |

Table 32: **DailyDialog Hyperparams**: Table shows a list of all hyper-parameters and their settings

| Tasks | Alg | Reward Function | LM | Lexical and Semantic Metrics | | | | | | | | Diversity Metrics | | | | | | | |
|---|---|---|---|---|---|---|---|---|---|---|---|---|---|---|---|---|---|---|---|
| | | | | Rouge-1 | Rouge-2 | Rouge-L | Rouge-LSum | Meteor | SacreBLEU | BertScore | Intent Accuracy | MSTTR | Distinct$_1$ | Distinct$_2$ | H$_1$ | H$_2$ | Unique$_1$ | Unique$_2$ | Mean Output Length |
| Dialog | Zero Shot | | GPT-2 | 0.157 | 0.012 | 0.131 | 0.131 | 0.191 | 0.066 | 0.854 | 0.427 | 0.608 | 0.055 | 0.316 | 7.787 | 11.831 | 1574 | 12327 | 18.685 |
| | Supervised | | GPT-2 | 0.162 | 0.020 | 0.138 | 0.138 | 0.186 | 0.064 | 0.855 | 0.437 | 0.635 | 0.065 | 0.342 | 8.051 | 12.119 | 1925 | 13952 | 18.919 |
| | PPO | Meteor + Intent | GPT-2 | 0.168 | 0.012 | 0.142 | 0.142 | 0.221 | 0.085 | 0.861 | 0.474 | 0.581 | 0.058 | 0.310 | 7.653 | 11.437 | 1719 | 12156 | 18.538 |
| | NLPO | Meteor + Intent | GPT-2 | 0.169 | 0.013 | 0.142 | 0.142 | 0.221 | 0.087 | 0.860 | 0.490 | 0.568 | 0.059 | 0.309 | 7.630 | 11.351 | 1718 | 11946 | 18.397 |
| | Supervised + PPO | Meteor + Intent | GPT-2 | 0.169 | 0.021 | 0.144 | 0.144 | 0.198 | 0.071 | 0.857 | 0.455 | 0.626 | 0.068 | 0.348 | 8.056 | 12.015 | 1983 | 14170 | 18.829 |
| | Supervised + NLPO | Meteor + Intent | GPT-2 | 0.171 | 0.020 | 0.146 | 0.146 | 0.205 | 0.074 | 0.858 | 0.454 | 0.624 | 0.070 | 0.349 | 8.044 | 11.990 | 2051 | 14213 | 18.763 |

Table 33: **Evaluation of Daily Dialog**: Table shows lexical, semantic metrics for RL algorithms bench-marked against supervised baseline models

| Algorithm | Unique N | Coherence | | | Quality | | |
|---|---|---|---|---|---|---|---|
| | | Value | Alpha | Skew | Value | Alpha | Skew |
| Zeroshot | 31 | 3.84 | 0.225 | 4.181 | 3.2 | 0.125 | 3.352 |
| NLPO | 30 | **4.18** | 0.114 | 4.17 | 3.35 | 0.159 | 3.318 |
| PPO | 32 | **4.18** | 0.112 | 4.032 | 3.32 | 0.163 | 3.478 |
| Supervised+PPO | 31 | 3.99 | 0.148 | 4.133 | 3.48 | 0.166 | 3.58 |
| Supervised+NLPO | 31 | 4.13 | 0.186 | 3.953 | **3.58** | 0.178 | 3.597 |
| Supervised | 31 | 3.96 | 0.249 | 3.834 | **3.59** | 0.236 | 3.196 |

Table 34: Results of the human subject study showing the number of participants N, average Likert scale value for coherence and sentiment, Krippendorf's alpha showing inter-annotator agreement, and Skew. For each model a total of 100 samples were drawn randomly from the test set and rated by 3 annotators each, each resulting in 300 data points per algorithm.

### B.9.2 Results and Discussion

Table 33 presents our benchmarking results of RL methods along with supervised baseline performances on test sets. Our main finding is that RL methods generally achieve better intent accuracy and automatic metric scores, in particular NLPO variants perform better than all other methods.

### B.9.3 Human Participant Study

Figure 10 shows the Daily Dialogue instructions and interface used for the human evaluation experiments. Tables 34, 35 show averaged results, annotator agreement, and the results of statistical significance tests to determine which models output better generations when rated by humans.

| Group 1 | Group 2 | Coherence | | Quality | |
|---|---|---|---|---|---|
| | | Diff (G2-G1) | *p-values* | Diff (G2-G1) | *p-values* |
| NLPO | PPO | -0.003 | 0.900 | -0.030 | 0.900 |
| NLPO | Supervised | **-0.227** | **0.043** | **0.238** | **0.020** |
| NLPO | Supervised+NLPO | -0.050 | 0.900 | **0.234** | **0.022** |
| NLPO | Supervised+PPO | **-0.194** | **0.013** | 0.127 | 0.803 |
| NLPO | Zero Shot | **-0.345** | **0.001** | -0.154 | 0.655 |
| PPO | Supervised | **-0.224** | **0.049** | **0.268** | **0.010** |
| PPO | Supervised+NLPO | -0.047 | 0.900 | **0.264** | **0.011** |
| PPO | Supervised+PPO | -0.191 | 0.144 | 0.157 | 0.636 |
| PPO | Zero Shot | **-0.341** | **0.001** | -0.124 | 0.822 |
| Supervised | Supervised+NLPO | **0.177** | **0.021** | -0.003 | 0.900 |
| Supervised | Supervised+PPO | 0.033 | 0.900 | -0.110 | 0.896 |
| Supervised | Zero Shot | -0.117 | 0.645 | **-0.391** | **0.002** |
| Supervised+NLPO | Supervised+PPO | -0.144 | 0.444 | **-0.107** | **0.009** |
| Supervised+NLPO | Zero Shot | **-0.294** | **0.002** | **-0.388** | **0.003** |
| Supervised+PPO | Zero Shot | -0.151 | 0.390 | **-0.281** | **0.008** |

Table 35: Results of an post-hoc Tukey HSD Test for difference in means between pairs of algorithms (Group 2 - Group 1) and corresponding $p$-values. Individually statistically significant results are bolded and are used to discuss results in the analysis. Overall $p$-values showing that there is a significant difference in means between the models via a one-way ANOVA test are significant with $p \ll 0.05$ for both coherence and sentiment.

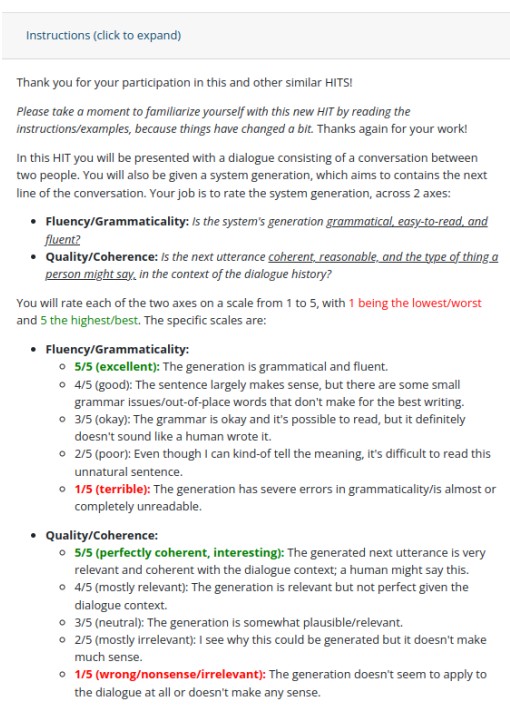

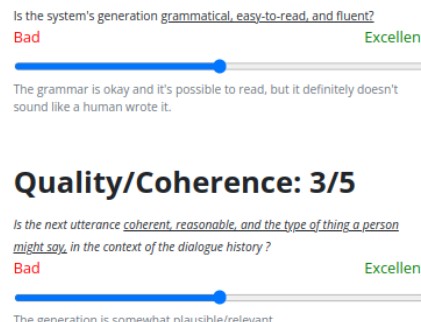

Figure 10: Instructions and interface for the Daily Dialogue task.

### B.9.4 Qualitative Analysis

We show sample generations from each of the algorithms for three randomly picked prompts below.

**Sample** 1
**Prompt:** Right . And China is the homeland of tea . <EOU> Yes , Chinese people love drinking
    tea so much . Some even claim they can't live without tea . <EOU> Do you know there are
    several catagories of Chinese tea ? <EOU> Yes , I believe there are green teas , black
    teas and scented teas . Any Others ? <EOU> Well , have you ever heard of Oulong tea and
    compressed tea ? <EOU>
**References:** Oh , yeah . Oulong tea is good for one's health . isn't it ?
**Zero-Shot:** I know that Oulong tea can be enjoyed with very few meals, and there are very few
**PPO:** I have, but it's not as strong as Oulong tea.
**NLPO:** I haven't. I don't have tea.
**Supervised:** Yes. I hear they do good. And I heard this Taijiquan Tea is very good
**Supervised** + **PPO:** Actually, I have had Oulong tea, but I haven't had any for awhile now.
**Supervised** + **NLPO:** No, I've never heard of that.

**Sample** 2

**Prompt:** I think I might have a dahl . <EOU> what's in a dahl ? <EOU> it's got chickpeas and
vegetables in a spicy curry sauce with rice . <EOU> that sounds nice . Do you want to
share some kebabs as well ? <EOU> ok . How about some lamb kebabs ? <EOU>
**References:** that's my favourite . Do you want to have some wine or beer ?
**Zero-Shot:** they're very salty
**PPO:** well you might want to take a peek.
**NLPO:** well I like that.
**Supervised:** ok, here they are. I'm so thirsty. I can't wait until the next bus to
**Supervised + PPO:** I've tried that a few times, but they're too dry and I don't want any of
**Supervised + NLPO:** oh, ok. That's a good choice. I'll take it.

**Sample** 3
**Prompt:** Really ? Have you had an auto-reverse break before ? <EOU> Yes , twice now . I think
it's too high-tech , and so it's the first part that breaks . <EOU> Hmm . You know , the
problem might be dust . Did you clean the unit often ? <EOU> Well . No , I didn't clean
it.But still , I only had the last one two months , and it broke . I don't trust auto-
reverse . <EOU> Alright . Well . We have many units without auto-reverse.Here is a good
unit . Very reliable.It has an AM / FM band , built-in microphone , recording capability ,
and no auto-reverse . <EOU>
**References:** How much is it ?
**Zero-Shot:** Oh. So you don't have the AM / FM band?
**PPO:** You said AM / FM. Are you sure about that?
**NLPO:** Well. Well. I had no idea when it was, and didn't care.
**Supervised:** Good.Well, what else I need?
**Supervised + PPO:** But I think it lacks the CD player or its CD player accessories.
**Supervised + NLPO:** Cool! So what's the trouble with it?

