# OpenReview forum: "Is Reinforcement Learning (Not) for Natural Language Processing: Benchmarks, Baselines, and Building Blocks for Natural Language Policy Optimization"
_ICLR.cc/2023/Conference — ICLR 2023 notable top 25%_

### Official Review · Reviewer_a1G9 · 2022-10-23

**Confidence:** 4
**Correctness:** 3
**Technical Novelty And Significance:** 3
**Empirical Novelty And Significance:** 3
**Recommendation:** 6

**Clarity, Quality, Novelty And Reproducibility:**

Neutral comment: The Figure 2 plots initially looked weird to me (e.g., why would there be a line connecting two unrelated tasks), but later I realized it’s a clear way to demonstrate the aggregate performance (because we can look at the area).


**Strength And Weaknesses:**

I think the toolkit will be a nice contribution to the research community. It will also encourage more people to dig deeper into the RL for NLP field.

It’s great that the software contains multiple RL algorithms.

The paper is clearly written – it’s effective in getting points across to readers.

Human annotation is involved, which is great.

Concerns below:

The selection of tasks isn’t totally satisfactory to me. Generating a positive continuation of movie reviews is too artificial and unchallenging. I think the benchmark (given there’s the word “general” in it) should consider real use cases of RL for NLP; for example: generating better dialogue responses where the reward is a human-preference-based metric or a toxicity-based metric. Moreover, IWSLT17 English-German is also quite artificial and IWSLT-trained systems will never be put into production or real use. Additionally, most of the tasks have a correct answer (or the space of correct generations is quite small), but there are many text generation tasks where the correct space of generations is very large, like in dialogue or story generation. I think these tasks should be taken into consideration as well.

One concern I have about the NLPO algorithm is that if the masked policy is an old copy, then the algorithm is not on-policy anymore. Then, the policy gradient derivation (e.g., see the Berkeley deep RL course policy gradient derivation, or any other policy gradient derivation) would not hold anymore. How do the authors justify the fact that we’re not able to do the policy gradient proof anymore (please correct me if I am wrong)? Or why does this periodic synchronization trick work? I would hope for a summary of either a theoretical analysis or empirical analysis.


**Summary Of The Paper:**

The paper introduces a library (open-sourced) called RL4LMs, which is for optimizing generation models using RL. The library is compatible with HuggingFace. Second, the paper comes up with a GRUE (general reinforced-language understanding evaluation) benchmark, which consists of six language generation tasks (under the RL framework) containing IMDB text continuation (reward is related to sentiment), a CommonGEN generative common sense task, CNN/DailyMail summarization, ToTTo (data-to-text generation), a machine translation task using a small IWSLT dataset, and NarrativeQA (question answering). The paper also proposes the NLPO algorithm (natural language policy optimization); the difference between NLPO and PPO is in green font in Algorithm 1. Essentially, the policy which we collect trajectories from is an old copy of the current policy. Moreover, initialization is different from PPO.


**Summary Of The Review:**

Toolkit -- nice contribution. Selection of task for the general benchmark -- not satisfactory to me. The NLPO algorithm is clever, but it needs more justification. In general, I'm leaning toward accepting because of the software and the encouraging empirical results.

---

> ### Author Response · Authors · 2022-11-18
> **Response to Reviewer**
>
> We thank the reviewer for the time and effort and will make some clarifications below.
>
> - The selection of tasks isn’t totally satisfactory to me…generating better dialogue responses where the reward is a human-preference-based metric or a toxicity-based metric.
>     - We have added a 7th task to the dataset, DailyDialog, which is a difficult chit-chat dataset with conversations from a range of everyday topics. Here we train an intent classifier designed to determine if a dialogue response generated by an agent is on topic and best matches the overall flow of the current conversation. The gist of our results here is that NLPO performs best on all metrics in both human participant studies as well as automated metrics. This is discussed in Section 5 and Appendix B.9
>     - We have also added in a pipeline to collect preferences, train rewards, and learn from the preferences as a step toward continual learning of preferences. We perform experiments on Commongen to validate it, finding that the pipeline does improve alignment to human preferences. This is discussed in Section 5.2 and Appendix B.4.4
> - Moreover, IWSLT17 English-German is also quite artificial and IWSLT-trained systems will never be put into production or real use.
>     - We have changed IWSLT-17 to the English-German subset of the more widely used WMT16 dataset (Bojar et al. 2016), finding that all the major trends discussed for IWSLT17 originally also apply to WMT16 (i.e. Supervised+NLPO performing best)
> - One concern I have about the NLPO algorithm is that if the masked policy is an old copy, then the algorithm is not on-policy anymore. Then, the policy gradient derivation (e.g., see the Berkeley deep RL course policy gradient derivation, or any other policy gradient derivation) would not hold anymore. How do the authors justify the fact that we’re not able to do the policy gradient proof anymore (please correct me if I am wrong)? Or why does this periodic synchronization trick work? I would hope for a summary of either a theoretical analysis or empirical analysis.
>     - Table 8 shows a empirical analysis by varying the hyperparameter of how often the masking policy update is applied, finding that in many cases unless the update interval is very small, the training performance remains approximately the same - robust to this hyperparameter.
>     - Huang and Ontanon, 2022 https://arxiv.org/abs/2006.14171 have a more theoretical analysis. Masking can be viewed as action elimination, the masking policy pre decides the action space and it can be thought of as part of the env - the policy is still online. WIth that formulation the standard policy gradient derivation still follows easily (the linked paper shows how standard action elimination doesn't violate the standard policy gradient derivation and results in an optimal gradient for the policy).

---

### Official Review · Reviewer_pVj7 · 2022-10-24

**Confidence:** 3
**Correctness:** 4
**Technical Novelty And Significance:** 2
**Empirical Novelty And Significance:** 1
**Recommendation:** 6

**Clarity, Quality, Novelty And Reproducibility:**

This paper has a clear writing and builds a library for benchmarking on a range of tasks. There is no concern on clarity.

**Strength And Weaknesses:**

Strength:
• This paper builds a cross-task diverse evaluation benchmark containing 6 tasks.
• For the proposed NLPO, human evaluation and automatic evaluation show it outperforms the PPO baseline


Weakness:
• With the mask prediction policy update, the gain based on human evaluation seems to be marginal except on IMDB task metrics, and CNN naturalness metric
• The proposed NLPO in this paper seems to be similar to [1] ?
• In experiments, there no comparison to traditional structure learning methods like [2]

[1] Donati, A.M., Quispe, G., Ollion, C., Corff, S.L., Strub, F., & Pietquin, O. (2021). Learning Natural Language Generation from Scratch. ArXiv, abs/2109.09371.

[2] Shu, R., Yoo, K., & Ha, J. (2021). Reward Optimization for Neural Machine Translation with Learned Metrics. ArXiv, abs/2104.07541.

**Summary Of The Paper:**

In this paper, the authors introduce a new library for benchmarking RL methods on NLP tasks, along with a modified version of PPO for on-policy optimization. Results demonstrate that the proposed NLPO algorithm outperforms the PPO baseline according to human assessment.

**Summary Of The Review:**

I would like to evaluate the contribution of this paper focused on the proposed NLPO method. I will lean towards an acceptation if the proposed method is indeed novel, results in state-of-the-art performance and the improvement is significant enough comparing to PPO.

---

> ### Author Response · Authors · 2022-11-18
> **Response to Reviewer**
>
> We would first like to thank the reviewer for their thoughtful comments and time. We will clarify our claims below.
>
> - The proposed NLPO in this paper seems to be similar to https://arxiv.org/abs/2109.09371?
>     - TrufLL also eliminates actions via tokens but the technique of action elimination is already well known. What is important is how those actions are eliminated. TrufLL uses a frozen, standard generic LM to eliminate actions to train LMs from scratch with RL. It is thus a competitior to supervised and self supervised methods of training LMs. In contrast, NLPO is a generalization of the method that can be used with other methods, where a KL penalty is used from a generic LM but the actual action elimination happens via copies of the current policy from previous iterations, making action elimination more task specific and effective overall. An ablation study showing that performance significantly decreases in equivalent settings to TrufLL (action elimination based on a generic frozen LM) is found in Appendix B.3 Table 8.
> - With the mask prediction policy update, the gain based on human evaluation seems to be marginal except on IMDB task metrics, and CNN naturalness metric.
>     - We note that all the results are significant and that NLPO shines especially in areas where the reward function is more prone to noise or in cases where the zero-shot performance of the model is not low (especially in the IMDB task or the newly added DailyDialog task as noted in the general reviewer response). NLPO in general achieves a better balance in terms of improving both task and naturalness metrics.

---

### Official Review · Reviewer_H8br · 2022-10-25

**Confidence:** 3
**Correctness:** 4
**Technical Novelty And Significance:** 3
**Empirical Novelty And Significance:** 3
**Recommendation:** 8

**Clarity, Quality, Novelty And Reproducibility:**

Clarity: For the most part, the writing in this paper is very clear.

Quality: Although RL is outside my area of expertise, the work appears to be high-quality. It implements 15 metrics/rewards and compares against 6 datasets, which is difficult to engineer.

Novelty: Based on a wide survey of RL-NLP lit, the authors "hypothesize that the size of the action space is a core cause of instability when training LMs with existing RL methods." Based on this insight, they develop a new learning algorithm (NLPO) which improves upon this shortcoming and outperforms the existing standard (PPO, as well as presumably REINFORCE, which it claims is strictly inferior to PPO based on previous work)

Reproducibility: By making their code (and datasets) into a public library, this work is very reproducible. Train/test splits of the datasets included in GRUE follow the splits of the original papers.

**Strength And Weaknesses:**

STRENGTHS
- This work is very comprehensive. It makes public a large library of many tasks, rewards/metrics, and learning algorithms.
- The learning algorithm (NLPO) outperforms the previous state of the art (PPO)
- In addition to the above work, further validating 6 of the datasets with MTurk is impressively thorough

WEAKNESSES
- The work could be more clear on pages 4 and 5 when describing on-policy actor-critic algorithms (3.3) and NLPO (4).

MINOR
- There appears to be a typo on page 8 "2 out of 4 tasks tasks". The word "tasks" is repeated.

**Summary Of The Paper:**

This work aims to explore whether reinforcement learning (RL) can be a useful part of NLP generative models. Although RL methods have been used fairly often, there have been disagreements among experts about whether it provides meaningful gains, especially in light of training difficulties (eg enormous action spaces of 30k+). In this work, the authors develop an open source library (6 tasks, 15 metrics, 3 learning algorithms), create a new state-of-the-art RL learning algorithm, create the first RL-NLP leaderboard, and use MTurk to annotate 4 of those 6 datasets with human-validated labels. It finds that tasks involving heavy novelty (ie "zero-shot performance") do seem to benefit from training with RL.

**Summary Of The Review:**

This work is very comprehensive: the authors develop an open source library (6 tasks, 15 metrics, 3 learning algorithms), create a new state-of-the-art RL learning algorithm, create the first RL-NLP leaderboard, and use MTurk to annotate 4 of those 6 datasets with human-validated labels. It finds that tasks involving heavy novelty (ie "zero-shot performance") do seem to benefit from training with RL. By making their code into a public benchmark, they provide a very helpful starting point for the community to develop & compare future work against. Finally, they provide a good lit review of many (though not every) instance of RL in NLP, commenting on its shortcomings to date.

---

> ### Author Response · Authors · 2022-11-18
> **Response to Reviewer**
>
> We thank the reviewer for their insightful comments and are encouraged by them. Some concerns are addressed below.
>
> - We would first like to point to our general response to all reviewers highlighting our three major changes adding to the comprehensiveness of the library: adding an additional Machine Translation dataset WMT-16 (Bojar et al. 2016), a new dialogue task in the form of DailyDialog (Li et al. 2017), and a human preference learning pipeline involving collecting data from humans for each task.
> - Regarding your concerns about the clarity of the Actor Critic algorithm’s descriptions, we have attempted to rephrase some of our wording in the text in Sections 3.3 and 4, and in the interest of space had moved some more detailed discussion on the workings of both general Actor Critic methods and NLPO to Appendix A. We hope these changes bring sufficient clarity on these topics.

---

### Official Review · Reviewer_zDTr · 2022-10-27

**Confidence:** 4
**Correctness:** 4
**Technical Novelty And Significance:** 3
**Empirical Novelty And Significance:** 4
**Recommendation:** 8

**Clarity, Quality, Novelty And Reproducibility:**


All of these are strong, particularly reproducibility. Novelty is a bit lower, but that is less important for a paper like this.


Questions:
- I'm very curious why the PPO + NLPO models do so badly without starting from the supervised init. Any ideas? (Never mind, I see this is addressed in 5.2).



**Strength And Weaknesses:**

Strengths:
+ The lack of open-source benchmarks and tasks for RL on language models is indeed a huge problem. This paper does the community a huge service by providing this library + benchmarks.
+ The paper is clearly written and easy to follow.
+ The paper is quite thorough. I appreciate the many ablations, and the study of human agreement with automatic metrics.
+ The NLPO algorithm is simple, clearly explained, and does seem to provide an improvement over PPO. It remains to be seen if this will transfer to other, more complicated tasks + larger scale. (it does involve an extra copy of the policy weights, which has memory downsides, especially at large scale, so it's unclear if it will be feasible / worth the tradeoff). Though overall I think this is a small part of the contribution.
+ The paper has a useful section on important implementation details.

Weaknesses:
- personal nit: I don't think CNN/DM is a very good summarization dataset (though it is very standard).
- I think the biggest weakness of this benchmark / tasks is that it relies on simple automatic metrics (eg BLEU, sentiment score). This makes sense of course, since it makes it tractable to evaluate. But I suspect that, once people start optimizing these metrics directly, whatever metric correlation with human judgments exists will disappear. I think it's important to consider how to incorporate human data collection into this process. (This would obviously be a very big endeavor, so I don't think it's within scope for this paper, but I think is a useful future direction.)  One option is to hold regular competitions (eg at a conference) where you collect human data which you use to evaluate models, and then open-source that data for further training (similar things have been done in the context of dialog models). Basically, I think it will require quite a bit of effort to ensure this benchmark remains useful over time, and doesn't saturate like every other NLP benchmark.

Small notes:
- "We find that using RL to learn from scalar reward feedback is can be more"
> remove "is"?

- "high-quality estimates.."
> two periods



**Summary Of The Paper:**

The paper studies RL on language models, and makes 3 contributions:
1) RL4LMs, a modular library for optimizing language generators with RL.
2) GRUE, a benchmark of 6 language generation tasks with reward functions.
3) NLPO, a new algorithm which improves on PPO for RL on LMs.

**Summary Of The Review:**

My personal belief is that fine-tuning language models to optimize specific metrics that are not perplexity (particularly human preferences) is going to be a huge field, perhaps one of the most impactful in ML. This paper clearly advances the accessibility of research on this problem. Thus, I think this paper may become an enormous contribution to the field. I'd be excited for the datasets and tasks to be updated regularly.

---

> ### Author Response · Authors · 2022-11-18
> **Response to Reviewer**
>
> We thank the reviewer for their time and effort and are encouraged by their analysis of this work's significance. We will attempt to address the weakness pointed out.
>
> - I think the biggest weakness of this benchmark / tasks is that it relies on simple automatic metrics (eg BLEU, sentiment score). This makes sense of course, since it makes it tractable to evaluate. Basically, I think it will require quite a bit of effort to ensure this benchmark remains useful over time, and doesn't saturate like every other NLP benchmark.
>     - We agree with the reviewer that GRUE is meant to be an ever-evolving benchmark and is not restricted to the original tasks. The framework allows for a wide range of tasks to be explored, we explore one such possibility during this discussion period by adding and experimenting on the DailyDialog chit-chat dialogue dataset, showing the usefulness of NLPO in particular here. Beyond just the results, we believe this showcases the flexibility and ease of extensibility of the library as we were able to perform a full set of experiments for a brand new task within the span of just over a week.
>     - Regarding the automated metrics, we further piloted a pipeline that collects human preference data, learns reward models on it and then optimizes LMs with RL on that as a way of testing the efficacy of such algorithms on more complex rewards. We find that we indeed able to align to human preferences more effectively using such rewards with RL.
>     - The details of the dialogue experiments are found in Section 5 and Appendix B.9 and the human preference learning experiments are found in Section 5.2 and Appendix B.4.4

---

### Author Response · Authors · 2022-11-18
**General Response to Reviewers and List of Salient Changes in Rebuttal Revision**

We thank the reviewers for their thoughtful comments and the time spent on constructing the reviews. We have made many significant improvements to our work in response to these comments. A salient changelist is found here:

1. (as suggested by Reviewer a1G9) Changed the IWSLT dataset to the more difficult WMT-16 (Bojar et al. 2016) dataset for Machine Translation.
   - We find that that general trends discussed do not change (Supervised+NLPO performs best)
2. (suggested by Reviewer a19G) Added a 7th task to the GRUE benchmark, DailyDialog (Li et al. 2017), as an example of a well-cited, more difficult task and a corresponding human evaluation study. These results are discussed in Section 5 and Appendix B.9.
   - Here we find that NLPO performs best among all other options when looking at both task and naturalness metrics.
3. (suggested by Reviewer zDTr, a19G) Added ablations experiments for *Human Preference Learning* in a pipeline form of collecting preference feedback data from humans, training reward models, and optimizing policies using these rewards with RL. This will serve as a step towards continual learning of such models. This is discussed in detail in Section 5.2 and Appendix B.4.4
   - Our findings suggest that after 2 rounds of deployment and preference collection, preference-reward based models are preferred in 682 cases compared to non-preference tuned models in 587 cases when presented head-to-head to human evaluators.

We have also clarified individual questions raised by each of the reviewers in separate responses.

We hope that the reviewers will consider raising their scores given these findings and additional experiment results that we have provided.

---

### Decision · Program_Chairs · 2023-01-20

**Decision:**

Accept: notable-top-25%

**Justification For Why Not Higher Score:**

Concerns on the selection of tasks/metrics in the benchmark.

**Justification For Why Not Lower Score:**

A number of contributions including a benchmark, an open-source library, and a new algorithm with improved performance.

**Metareview: Summary, Strengths And Weaknesses:**

The paper studies Reinforcement Learning (RL) for natural language generation. In particular, the paper develops an open-source library that implements several on-policy RL algorithms, a benchmark consisting of 6 language generation tasks, as well as new algorithm that improves over PPO for language generation tasks. Overall the paper makes a number of contributions that would facilitate the RL research for NLG/NLP. Reviewers had concerns on the selected tasks and metrics in the benchmark, e.g., CNN/DM is not an ideal corpus for studying summarization, and some of the tasks tend to be artificial. The authors have included a new task on dialog that partially addressed the concerns.

A side note: as a paper aiming to present a benchmark for RL-for-NLG research, it'd be nice to have a more comprehensive discussion (e.g., in related work) on the different existing RL algorithms applied to NLG in previous work. In particular, though claiming "RL" for NLG, the work seems having only discussed "on-policy" "policy-based" RL, without more discussion/mentioning on other categories of RL methods, such as "off-policy"/"value-based"/"offline", etc, that have been studied in NLG context.


**Note From Pc:**

if the above contains the word "oral" or "spotlight" please see: "oral" presentation means -> notable-top-5% and "spotlight" means -> notable-top-25%. As stated in our emails, we are disassociating presentation type from AC recommendations